# Hyperbolic Associative Memory Networks

**Boliang Hao**[1] **Bailing Zhang**[2] **Fangyu Wu**[3]

## Abstract

Modern Hopfield Networks (MHNs) have achieved widespread success across various domains but are confined to Euclidean/Hilbert spaces, failing to preserve the hierarchical structure of data due to geometric constraints—arbitrary tree structures cannot be embedded with low distortion, while hyperbolic spaces can naturally accommodate hierarchical structures through exponential volume growth. To address this issue, we propose Hyperbolic Associative Memory Networks (HAMNs), the first framework to embed modern associative memory into hyperbolic space: we map query and memory vectors from Euclidean space to a constant negative curvature manifold via exponential maps, define a regularized energy function based on the Minkowski inner product, and adopt curvature-aware Riemannian optimization combined with exponential map updates to achieve stable on-manifold retrieval. We put forward a hierarchy-sensitivity hypothesis—HAMNs outperform Euclidean MHNs on data with deep hierarchies but exhibit comparable performance on data with weak or shallow hierarchies, which is validated by depth-controlled experiments and cross-level metrics. As a plug-and-play, model-agnostic module, HAMNs are suitable for the storage and retrieval of representations in task architectures requiring hierarchical understanding, instantiated with the Poincaré ball in experiments, and also applicable to any hyperbolic model with constant negative curvature. The code is available at https://github.com/hbl66/HAMNs.

## 1. Introduction

Associative memory models have played a crucial role in enabling neural systems to retrieve stored patterns from partial or noisy inputs. In this domain, classical Hopfield network models (Hopfield, 1982; Amari, 1972) store memories as fixed-point attractor states in an energy landscape, leveraging Hebbian learning to recall full patterns from partial input cues through a recurrent architecture. More recently, modern Hopfield-type networks (Krotov & Hopfield, 2016) generalized the original Hopfield energy and motivated capacity analyses showing exponential storage under suitable settings (Demircigil et al., 2017). Ramsauer et al. (Ramsauer et al., 2021) further linked continuous-state modern Hopfield updates to Transformer softmax attention (Vaswani et al., 2017). Recent research has significantly extended the MHN framework in several directions. For example, kernelized MHNs (Wu et al., 2024) introduce learnable kernels that map raw memories into a Hilbert feature space, inducing a new distance structure to capture similarity between queries and memories, thereby improving memory capacity and robustness. Sparse and structured MHNs (Hu et al., 2023; Santos et al., 2024) introduce sparse entropic regularizers or Fenchel–Young losses to enhance efficiency and scalability. Latent-structured MHNs (Li et al., 2026) embed continuous attractor dynamics into autoencoder latent spaces to improve semantic association and episodic memory. However, our focus is not on kernel functions or regularization mechanisms, which are still carried out in Euclidean geometry with zero curvature. Modern Hopfield networks are typically used in deep learning as energy-based retrieval layers or attention-like modules, and we follow this line of work: we provide a geometric extension of existing MHN methods by moving the Hopfield energy and dynamics from Euclidean or Hilbert spaces to negatively curved manifolds and studying how this geometric change improves hierarchical retrieval. At the same time, our goal is to design a differentiable associative memory module for machine learning systems, rather than to emphasize biological interpretability.

Although representing data in Euclidean space $\mathbb{R}^d$ has traditionally been the standard choice because of its computational convenience, recent studies have shown that this approach faces fundamental limitations when applied to complex data types (Ganea et al., 2018). Many real-world datasets, particularly those involving graphs, taxonomies,

[1]Zhejiang Sci-Tech University, Hangzhou, Zhejiang, China [2]NingboTech University, Ningbo, Zhejiang, China [3]Xi'an Jiaotong-Liverpool University, Suzhou, Jiangsu, China. Correspondence to: Fangyu Wu <Fangyu.wu02@xjtlu.edu.cn>.

*Proceedings of the 43rd International Conference on Machine Learning*, Seoul, South Korea. PMLR 306, 2026. Copyright 2026 by the author(s).

or hierarchical relationships, exhibit an inherently non-Euclidean latent structure (Bronstein et al., 2017). In such cases, Euclidean embeddings often struggle to faithfully preserve semantic proximity and hierarchical organization (Gromov, 1987). For example, arbitrary tree structures cannot be embedded with arbitrarily low distortion even in high-dimensional Euclidean spaces (Linial et al., 1995), whereas hyperbolic spaces, owing to their exponential growth of volume, can naturally accommodate such structures even in low dimensions (Krioukov et al., 2010; Nickel & Kiela, 2017). Thus, in tasks of this kind (e.g. hierarchical classification, hierarchical clustering, knowledge graph completion, and graph/image/text classification or retrieval with hierarchical labels), applying associative memory mechanisms purely within Euclidean geometry may distort the underlying structural information during memory retrieval. These observations motivate us to embed the associative memory process into hyperbolic space, changing the geometry of the associative memory layer so as to represent hierarchical information.

To address these limitations, we introduce Hyperbolic Associative Memory Networks (HAMNs), the first framework that embeds modern associative memory into hyperbolic space. Specifically, we first apply exponential maps to transform query and memory vectors from Euclidean space to hyperbolic space (a constant–negative–curvature manifold), thereby leveraging the natural capacity of hyperbolic geometry to model hierarchical structures. On top of these mapped representations, we define a principled energy function using the Minkowski inner product to capture similarity relations in hyperbolic geometry. During memory retrieval, we incorporate curvature-aware Riemannian optimization (Bonnabel, 2013) with exponential-map updates to ensure that each update step follows the tangent direction of the hyperbolic manifold and remains strictly within hyperbolic space. On the theoretical side, we prove that, under fixed state dimension and minimum geodesic separation, formulating the Hopfield energy on a negatively curved manifold allows the number of well-separated recall wells to grow exponentially with the effective hyperbolic radius (i.e., hierarchy depth), whereas in the Euclidean case the analogous packing bound grows only polynomially with the radius. In our experiments, we instantiate the method with the Poincaré ball (Nickel & Kiela, 2017) due to implementation maturity, while the derivations apply equally to other hyperbolic models.

With this design, we propose a hierarchy-sensitivity hypothesis that does not presuppose pronounced hierarchical structure in all tasks or datasets; when hierarchical/tree structure does exist and is sufficiently deep, HAMNs demonstrate a stronger ability to understand, preserve, and retrieve hierarchical relations, whereas when the hierarchy is weak or essentially absent, their performance is largely on par with

Euclidean MHNs. To validate this hypothesis, we conduct a systematic evaluation by controlling hierarchy depth and reporting metrics such as cross-level consistency, and the empirical results are consistent with the hypothesis. Our main contributions are summarized as follows:

- We design Hyperbolic Associative Memory Networks (HAMNs), a plug-and-play, model-agnostic associative memory module operating in hyperbolic space that can be dropped into architectures requiring hierarchical understanding to store and retrieve raw inputs, intermediate representations, or learned prototypes, explicitly leveraging hierarchical structure.

- We design a principled energy function and optimization mechanism based on hyperbolic geometry, ensuring a stable and efficient memory update process.

- With hierarchy depth controlled and cross-level consistency measured, our method achieves clear benefits on hierarchical data and competitive flat/shallow results, outperforming Euclidean Hopfield networks at representing complex structures.

## 2. Preliminaries

### 2.1. Modern Hopfield Networks

Modern Hopfield Networks (MHNs) (Ramsauer et al., 2021) extend classical associative memory models by introducing continuous state representations and modifying the energy function landscape. These modifications significantly enhance the storage capacity and enable the network to retrieve stored patterns through continuous optimization dynamics.

Given a set of $N$ memory patterns $\{\xi_i \in \mathbb{R}^d\}_{i=1}^N$, organized as a memory matrix $\Xi \in \mathbb{R}^{N \times d}$, and a query state vector $s \in \mathbb{R}^d$, the energy function of MHNs is formulated as:

$$E(s, \Xi; \theta) = F_\theta\left(f_{\text{sim}}(\{\xi_i\}, s)\right) + \frac{1}{2}s^\top s \qquad (1)$$

where the similarity is defined as $f_{\text{sim}}(\{\xi_i\}, s) = \{\langle \xi_i, s \rangle\}_{i=1}^N$ (dot product between $s$ and each memory), and $F_\theta(\cdot)$ is the log-sum-exponential (LSE) function: $F_\theta(z) = -\frac{1}{\theta}\log\left(\sum_{i=1}^N \exp(\theta z_i)\right)$, with $\theta > 0$ controlling the sharpness. The associative retrieval process minimizes the energy iteratively as:

$$
\begin{aligned}
s^{(t+1)} &= \Xi^\top \operatorname{softmax}\left(\theta \Xi s^{(t)}\right) \\
&= \sum_{i=1}^N \xi_i \frac{\exp\left(\theta \xi_i^\top s^{(t)}\right)}{\sum_{j=1}^N \exp\left(\theta \xi_j^\top s^{(t)}\right)}
\end{aligned}
\qquad (2)
$$

Under mild conditions, the update rule monotonically decreases the system energy and converges to a (meta-)stable

fixed point (Ramsauer et al., 2021). This framework thus enables efficient pattern retrieval even from noisy or partial cues. Eq. (2) is equivalent to the readout of single-head attention, with keys=values $= \Xi$ and query $s^{(t)}$, hence an MHN can be viewed as an energy-based realization of attention.

## 2.2. Hyperbolic Geometry Preliminaries

A **hyperbolic manifold** is a Riemannian manifold $(\mathcal{M}, g)$ with constant negative curvature $-c < 0$ (Cannon et al., 1997). It has diverging geodesics, triangle angle sums smaller than $\pi$, and exponential volume growth with radius, which naturally matches the branching structure of trees and hierarchies (Gromov, 1987; Krioukov et al., 2010; Nickel & Kiela, 2017). Hyperbolic space admits several mutually isometric models, such as the Poincaré ball, Klein, and Lorentz models, which differ only by parametrization. In this paper, we rely only on three standard Riemannian primitives.

### 2.2.1. PRIMITIVE 1: EXP/LOG MAPS

For any $p \in \mathcal{M}$, the exponential and logarithmic maps

$$\exp_p^c : T_p\mathcal{M} \to \mathcal{M}, \qquad \log_p^c : \mathcal{M} \to T_p\mathcal{M}$$

move along geodesics and back: given a tangent vector $v \in T_p\mathcal{M}$, $\exp_p^c(v)$ is the point reached at unit time on the geodesic starting from $p$ with initial velocity $v$, while $\log_p^c$ is its local inverse. They satisfy $\exp_p^c(0) = p$ and $\mathrm{d}(\exp_p^c)_0 = \mathrm{id}$, so for small $v$ we have $\exp_p^c(v) \approx p + v$ in local coordinates. In practice, these maps form the bridge between Euclidean vector spaces and the manifold: we encode queries and memories in $T_p\mathcal{M}$, map them to $\mathcal{M}$ with $\exp_p$, and during retrieval compute a descent direction in $T_{\xi^{(t)}}\mathcal{M}$ (via $\log_{\xi^{(t)}}^c$ and the Riemannian gradient) before exponentiating it back onto the manifold.

### 2.2.2. PRIMITIVE 2: GEODESIC DISTANCE

**Definition.** $d_{\mathcal{M}}(x, y)$ is the length of the shortest geodesic between $x$ and $y$ induced by $g$.

**Hierarchy intuition.** Radial distance grows roughly linearly with radius, but near the boundary any fixed Euclidean displacement is exponentially magnified, naturally separating differences in hierarchical depth (see the toy example C.2).

### 2.2.3. PRIMITIVE 3: HYPERBOLIC SIMILARITY

We use $\langle x, y \rangle_M := -\cosh\big(d_{\mathcal{M}}(x, y)\big)$ as the similarity in hyperbolic space. It has two useful properties: *(i) Monotonicity* — it is decreasing in $d_{\mathcal{M}}(x, y)$, equals $-1$ at $x = y$, and tends to $-\infty$ as distance increases; and *(ii) Equivalence* — in the Lorentz model this similarity is a monotone func-

tion of the Minkowski bilinear form, coinciding numerically with it when the curvature is $-1$, so the construction is model-agnostic and invariant under hyperbolic isometries.

All derivations below are written in terms of these model-agnostic primitives, so they apply to any constant-curvature hyperbolic realization. In experiments we instantiate the method on the Poincaré ball because of its implementation convenience (Nickel & Kiela, 2017). More detailed hyperbolic-geometry background, including isometry invariance, common model formulas, and a toy hierarchy example, is deferred to Appendix C.

## 3. Methodology

Our proposed Hyperbolic Associative Memory Networks (HAMNs) use hyperbolic geometry to store and retrieve patterns. This section introduces the core components of HAMNs.

### 3.1. Memory Encoding in Hyperbolic Space

We first map all memories and the query onto a common hyperbolic manifold. Let $x_i^R \in \mathbb{R}^d$ ($i = 1, \ldots, N$) denote the $N$ stored patterns in Euclidean space (these can be regarded as the keys in memory), and let $\xi^R \in \mathbb{R}^d$ be the query pattern (the cue or initial state). Let $(\mathcal{M}, g)$ be a complete, simply connected Riemannian manifold with constant negative curvature $-c < 0$. Choose a reference point $p \in \mathcal{M}$ and fix an orthonormal frame on its tangent space $T_p\mathcal{M}$, thereby identifying $T_p\mathcal{M} \cong \mathbb{R}^d$ via an isometric isomorphism $\iota_p : \mathbb{R}^d \to T_p\mathcal{M}$. We encode using the exponential map at $p$:

$$\begin{aligned} v_i &= \iota_p(x_i^R), & v_\xi &= \iota_p(\xi^R), \\ x_i &= \exp_p^c\big(\Pi(v_i)\big), & \xi &= \exp_p^c\big(\Pi(v_\xi)\big), \end{aligned} \tag{3}$$

To avoid, in some models, mapped points becoming too close to the boundary (which may lead to numerical instability and gradient explosion), we may perform norm clipping in the tangent space *before* the exponential map. Given a clipping threshold $\texttt{clip}_{\text{tan}} > 0$, $\Pi(\cdot)$ denotes tangent-space norm clipping:

$$\Pi(v) = v \cdot \min\left(1, \frac{\texttt{clip}_{\text{tan}}}{\|v\| + \varepsilon}\right), \qquad \varepsilon > 0. \tag{4}$$

Here $\varepsilon$ is a small constant for numerical stability (e.g. $10^{-5}$). After this encoding step, all memory points $x_i$ and the query point $\xi$ lie on the manifold $\mathcal{M}$.

### 3.2. Energy Function Design

On a hyperbolic manifold, we use an energy function $E(\xi)$ to measure how well the current retrieval state $\xi$ matches the stored patterns $\{x_i\}_{i=1}^N$: the energy should be low when

$\xi$ is close to some memory $x_i$, and high otherwise. To this end, we replace the Euclidean inner product by a *hyperbolic similarity*:

$$\langle x, y \rangle_M := -\cosh\big(d_{\mathcal{M}}(x,y)\big),$$

where $d_{\mathcal{M}}$ is the geodesic distance induced by the metric $g$. This similarity is identical across hyperbolic models; in particular, in the Lorentz (hyperboloid) model $\langle x, y \rangle_M$ coincides with the classical Minkowski inner product, while in other models it can be computed directly from $d_{\mathcal{M}}$ without explicitly mapping between models. See Appendix A.1 for a short derivation and geometric intuition.

Accordingly, for any $\xi \in \mathcal{M}$ we define the energy as

$$E(\xi) \;=\; -\frac{1}{\theta} \log\left( \sum_{i=1}^{N} \exp\big(\theta \langle x_i, \xi \rangle_M\big) \right) + \frac{1}{2} d_{\mathcal{M}}(\xi, p)^2, \tag{5}$$

where $\theta > 0$ is a temperature parameter and $p \in \mathcal{M}$ is a fixed reference point (e.g., the origin in Poincaré coordinates). We use the *intrinsic* squared geodesic regularizer $\frac{1}{2} d_{\mathcal{M}}(\xi, p)^2$, which is geodesically convex in hyperbolic space and penalizes deviations from $p$.

The first term in (5) is a smooth approximation to the "maximum similarity": when $\theta$ is large, $-\frac{1}{\theta} \log \sum_i \exp(\theta \langle x_i, \xi \rangle_M) \approx -\max_i \langle x_i, \xi \rangle_M$, so it is minimized when $\xi$ is close to one of the memories $x_i$. The second term penalizes large geodesic deviations from $p$, suppressing excursions toward the boundary and stabilizing the optimization trajectory.

Together, these two terms yield energy minima around stored memories. When $\xi = x_k$, we have $d_{\mathcal{M}}(x_k, \xi) = 0$ and $\langle x_k, \xi \rangle_M = -1$, leading to a low energy; conversely, when $\xi$ is far from all memories, the energy becomes large. Further discussion of energy bounds is provided in Appendix A.3. A simple inequality relating geodesic margin, softmax mass, and a sphere-packing style capacity bound is given in Appendix B (Propositions B.1, B.3, and B.4). In addition, Appendix B.4 provides a random-shell capacity theorem, showing that randomly sampled hyperbolic shell memories yield retrieval-stable basins with high probability and imply an $e^{\Omega(d)}$ lower bound on random-memory capacity.

### 3.3. Memory Retrieval and Optimization

We optimize the retrieval energy using a *Riemannian* version of the Concave–Convex Procedure (CCCP) on a Hadamard manifold (Yuille & Rangarajan, 2001). A detailed derivation and convergence discussion for our setting are provided in Appendix A.4, and optional neural-computation analogies for the softmax weights, Riemannian gradient, and the explicit base-point tangent CCCP step are given in Appendix A.6.

**CCCP decomposition** We decompose $E(\xi)$ in (5) into a geodesically convex term and a smooth log-sum-exp term:

$$\begin{aligned} E(\xi) &= E_{\text{cvx}}(\xi) + E_{\text{cave}}(\xi), \\ E_{\text{cvx}}(\xi) &= \tfrac{1}{2} d_{\mathcal{M}}(\xi, p)^2, \quad p \in \mathcal{M} \\ E_{\text{cave}}(\xi) &= -\frac{1}{\theta} \log\Big( \sum_{i=1}^{N} e^{\theta \langle x_i, \xi \rangle_M} \Big), \end{aligned} \tag{6}$$

where $\langle x, \xi \rangle_M := -\cosh(d_{\mathcal{M}}(x, \xi))$ denotes the hyperbolic similarity. The squared distance is geodesically convex on Hadamard manifolds.

**Softmax weights** At iteration $t$, define

$$w_i^{(t)} \;=\; \frac{\exp\big(\theta \langle x_i, \xi^{(t)} \rangle_M\big)}{\sum_{j=1}^{N} \exp\big(\theta \langle x_j, \xi^{(t)} \rangle_M\big)}. \tag{7}$$

**Riemannian linearization and surrogate** Let $a^{(t)} := \text{grad}\, E_{\text{cave}}(\xi^{(t)})$ be the *Riemannian* gradient at $\xi^{(t)}$. When $E_{\text{cave}}$ is geodesically concave on a geodesically convex neighborhood $\mathcal{R}$ containing $\xi^{(t)}$ (e.g., under the non-degenerate variance condition in App.A.4.1), the Riemannian first-order expansion gives an upper bound: for all $\xi \in \mathcal{R}$,

$$E_{\text{cave}}(\xi) \;\leq\; E_{\text{cave}}(\xi^{(t)}) \;+\; \big\langle a^{(t)}, \log_{\xi^{(t)}}(\xi) \big\rangle_{\xi^{(t)}}.$$

so the CCCP/MM surrogate on $\mathcal{R}$ reads

$$\begin{aligned} Q\big(\xi \mid \xi^{(t)}\big) \;=\; &\tfrac{1}{2} d_{\mathcal{M}}(\xi, p)^2 + \\ &\big\langle a^{(t)}, \log_{\xi^{(t)}}(\xi) \big\rangle_{\xi^{(t)}} \quad \text{(constants dropped).} \end{aligned} \tag{8}$$

**Explicit CCCP/MM step via a base-point tangent proxy** We obtain an explicit update by minimizing a tangent proxy of (8) in $T_p\mathcal{M}$: let $u = \log_p(\xi)$ and $b^{(t)} = \text{PT}_{\xi^{(t)} \to p}(a^{(t)})$, and minimize $\frac{1}{2} \|u\|_p^2 + \langle b^{(t)}, u \rangle_p$,

$$\xi^{(t+1)} \;=\; \exp_p\Big( -\,\text{PT}_{\xi^{(t)} \to p}\big(a^{(t)}\big)\Big), \tag{9}$$

where $\text{PT}_{\xi^{(t)} \to p}$ denotes parallel transport along the unique geodesic from $\xi^{(t)}$ to $p$. Equivalently, introducing $v^{(t)} := -\text{PT}_{\xi^{(t)} \to p}\big(a^{(t)}\big)$ and a damping step size $\eta \in (0, 1]$, we use the stable update

$$\xi^{(t+1)} \;=\; \exp_p\big( \eta\, v^{(t)} \big), \tag{10}$$

where $\eta = 1$ yields (9).

**Intrinsic gradient** Using (7), the Riemannian gradient of the log-sum-exp term can be written as

$$\begin{aligned} a^{(t)} \;=\; &\text{grad}\, E_{\text{cave}}(\xi^{(t)}) \\ =\; &-\sum_{i=1}^{N} w_i^{(t)} \,\text{grad}_\xi \langle x_i, \xi \rangle_M \Big|_{\xi = \xi^{(t)}}. \end{aligned} \tag{11}$$

---

**Algorithm 1** HypHopfield retrieval on the Poincaré ball $\mathbb{D}_c^d$

---

**Require:** Memories $Y = \{Y_i\}_{i=1}^N \subset \mathbb{D}_c^d$, queries $R^{(0)} = \{R_b^{(0)}\}_{b=1}^B \subset \mathbb{D}_c^d$, curvature $c > 0$, temperature $\theta > 0$, stepsize $\eta \in (0, 1]$, base point $p = 0$, max iters $T_{\max}$, tolerance $\varepsilon$

1: **for** $t = 0, 1, \ldots, T_{\max} - 1$ **do**
2:    **Hyperbolic similarities:**  $S_{b,i} \leftarrow -\cosh\big(d_{\mathbb{D}_c}(Y_i, R_b^{(t)})\big)$
3:    **Soft weights:**  $W_{b,i} \leftarrow \exp(\theta S_{b,i}) \big/ \sum_j \exp(\theta S_{b,j})$
4:    **Riemannian gradient of concave term at $R_b^{(t)}$:**

$$a_b \leftarrow -\sum_{i=1}^N W_{b,i} \ \sinh\big(d_{\mathbb{D}_c}(Y_i, R_b^{(t)})\big) \ \frac{\log_{R_b^{(t)}}(Y_i)}{\|\log_{R_b^{(t)}}(Y_i)\|_g}$$

5:    **Parallel transport to $p = 0$:**  $v_b \leftarrow -\mathrm{PT}_{R_b^{(t)} \to 0}(a_b)$
6:    **Base-point update (CCCP with damping):**  $R_b^{(t+1)} \leftarrow \exp_0^c(\eta\, v_b)$,   project back to $\mathbb{D}_c^d$ if needed
7:    **Stopping:**  **if** $d_{\mathbb{D}_c}(R_b^{(t+1)}, R_b^{(t)}) < \varepsilon$ for all $b$ **then break**
8: **end for**
9: **Output:**  $Z = \{R_b^{(t+1)}\}_{b=1}^B$

---

**Convergence note** Monotonicity holds when the local surrogate (8) upper-bounds $E$ and is minimized (App. A.4). This is a local, region-dependent condition rather than a global guarantee; in particular, it may fail near sharply peaked single-memory attractors. An empirical local-concavity probe is provided in Appendix A.4.2.

### 3.4. Hyperbolic Hopfield Modules and Implementation

Sec. 3.3 defines the model-agnostic HAMN retrieval update. We now describe how this update is instantiated as plug-and-play neural modules. In all experiments, we implement HAMNs on the Poincaré ball $\mathbb{D}_c^d$ and use the origin $p = 0$ as the reference point. The three modules share the same retrieval core but differ in how queries and memories are provided.

**HypHopfield.** This is the basic query-memory association module. It takes queries $R \in \mathbb{R}^{B \times d}$ and memories $Y \in \mathbb{R}^{N \times d}$, maps them to the Poincaré ball, performs hyperbolic associative retrieval, and returns retrieved states $Z \in \mathbb{R}^{B \times d}$.

**HypPooling.** This module uses a small number of learnable query vectors to aggregate a variable-size set of instance embeddings. The instance embeddings serve as memories, while the learnable queries retrieve a fixed-size set of hyperbolic summary vectors. It is used for set- or bag-level aggregation, such as multi-instance learning.

**HypLayer.** This layer propagates input queries through a learnable memory matrix $Y \in \mathbb{R}^{N \times d}$. The memory matrix can be initialized randomly, from class prototypes, or from training-set embeddings, and is optimized together with the downstream model. It supports prototype retrieval, similarity-based matching, and pattern aggregation.

**Computational pipeline.** Given Euclidean features from an upstream encoder, the overall pipeline is:

$$\text{Euc. features} \to T_p\mathcal{M} \to \mathbb{D}_c^d \to \text{HAMNs retrieval}$$
$$\to T_p\mathcal{M} \to \text{task head.}$$

Concretely, features are first mapped to the tangent space at the reference point and then to the Poincaré ball using $\exp_p^c$, optionally with tangent-space norm clipping for numerical stability. Retrieval is performed on the ball. When a Euclidean output is required, the retrieved state is mapped back to the tangent space using $\log_p^c$ and then passed to task-specific prediction heads.

**Poincaré-ball implementation.** In the Poincaré-ball instantiation, the generic update in Eq. (10) is implemented by computing geodesic-distance similarities $S_{b,i} = -\cosh(d_{\mathbb{D}_c}(Y_i, R_b))$, forming the corresponding soft weights, transporting the Riemannian gradient of the concave log-sum-exp term to the base point $p = 0$, and applying the base-point exponential map. For $p = 0$, the Poincaré-ball transport is implemented by the conformal rescaling

$$v_b = -\mathrm{PT}_{R_b \to 0}(a_b) = -\frac{\lambda(R_b)}{2} a_b, \qquad \lambda(x) = \frac{2}{1 - c\|x\|^2}.$$

The explicit Poincaré-ball update is therefore

$$R_b^+ = \exp_0^c(\eta v_b) = \tanh\big(\sqrt{c}\,\|\eta v_b\|\big) \frac{\eta v_b}{\sqrt{c}\,\|\eta v_b\|}.$$

Alg. 1 summarizes the full retrieval procedure.

The same algorithmic core is used by HypHopfield, Hyp-Pooling, and HypLayer; the difference lies only in whether the queries are input-dependent, learnable pooling vectors, or states propagated through a learnable memory matrix. Additional implementation details, including the explicit Poincaré-ball formulas, parallel-transport choices, and task-specific usage of the three modules, are provided in Appendix D.

## 3.5. Relation to Hyperbolic Attention

Although HAMNs and hyperbolic attention both operate on a hyperbolic manifold, they implement different retrieval mechanisms. Hyperbolic attention (Gülçehre et al., 2019) is a one-shot geometric attention layer: it computes weights from geodesic distances and returns a Fréchet mean or weighted manifold aggregation for each query, without defining an explicit global energy or iterative retrieval dynamics. By contrast, HAMNs define the Hopfield-style energy in Eq. (5) and retrieve memories by approximately minimizing this energy through the CCCP update in Eq. (10). Therefore, HAMNs are not merely hyperbolic attention with a different similarity function; they move both the energy landscape and the retrieval dynamics of MHNs to a negatively curved manifold. A more detailed comparison is provided in Appendix D.4.

## 4. Experiments

*Instantiation.* All experiments instantiate HAMNs on the Poincaré ball model (constant negative curvature $-c$); the model-agnostic derivations hold for any hyperbolic realization, and concrete formulas for instantiations on common hyperbolic models are provided in Appendix C.3.

**Overview**  We evaluate HAMNs around the "hierarchy-sensitivity hypothesis" and their practical usefulness through five groups of experiments:

1. **CIFAR-100 hierarchical classification**: On our 2/3/4-layer label trees, **HAMNs** deliver the strongest cross-level consistency and competitive accuracy across levels, clearly outperforming **Euclidean MHNs** and the kernelized **U-Hop**; the consistency gap widens as the hierarchy deepens. **HypAttn** is strongest for shallow/mid-level retrieval, while **HypNN** excels at fine-grained recognition.

2. **Weak/shallow hierarchical tasks**: On classical MIL multi-instance learning and MoleculeNet molecular property prediction (where hierarchy is weak or only shallow), HAMNs perform on par with Euclidean MHNs overall, with slight advantages on a few datasets.

3. **Real-world taxonomy and knowledge-graph tasks on WordNet**: On WordNet hypernym prediction, we further test a shared ontology-embedding front-end and find that ontology embeddings and hyperbolic retrieval are complementary. On WN18RR, we keep the encoder fixed and swap only the decoder to test whether HAMNs can serve as a plug-and-play hyperbolic retrieval module.

4. **Computation/performance comparison**: Theoretically fewer FLOPs and parameters, but due to hyperbolic operations and memory-access overhead, the current GPU implementation exhibits longer runtime and higher peak memory.

5. **Ablation studies**: Performance is best when the curvature $c$ lies in a moderate range (approximately 0.7–2.0); using too many stored patterns slightly degrades top-level accuracy, though the method is overall robust to this hyperparameter.

## 4.1. Hierarchical classification on CIFAR-100

To demonstrate that HAMNs can understand and exploit multi-level structure, we conduct hierarchical classification experiments on CIFAR-100 (Krizhevsky et al., 2009). CIFAR-100 groups 100 *fine* classes into 20 *coarse* classes, yielding a balanced two-level hierarchy. Without modifying the original samples, we further cluster the 20 coarse classes into 7 "super" classes (e.g., large terrestrial vertebrates, plants, vehicles), and then group these 7 super classes into 3 "top" classes (e.g., animals, plants & natural scenes), thereby forming three- and four-level hierarchies.

On the model side, we adopt a ResNet-18 (He et al., 2016) backbone with the final fully connected layer removed, and insert one of five memory/retrieval modules: (i) **HAMNs** (Using our **HypLayer**; see Appendix D.), (ii) a hyperbolic attention (**HypAttn**; (Gülçehre et al., 2019)), (iii) a lightweight hyperbolic neural block (**HypNN**; (Ganea et al., 2018)), and (iv) Euclidean-space modern Hopfield networks (**MHNs**; (Ramsauer et al., 2021)). (v) a kernelized Euclidean Hopfield (**U-Hop**; (Wu et al., 2024)). The retrieved representations are then fed into level-specific classification heads.

*Coarse–Fine Coherence Correlation* (coph_corr) measures the consistency between the model's "coarse" predictions and the "coarse" predictions obtained by aggregating its "fine" outputs.

From Table 1 we observe a clear pattern across depths. Euclidean MHNs remain reasonably strong but are never dominant: they reach the best hierarchy-consistency score (coph_corr) on the 3-layer tree, yet on the deepest 4-layer setting they trail all hyperbolic variants in both high-level accuracy and cross-level coherence, and also under-

*Table 1.* Hierarchical classification on CIFAR-100 results.

| MODEL | TOP_ACC | SUPER_ACC | COARSE_ACC | FINE_ACC | COPH_CORR |
|---|---|---|---|---|---|
| **CIFAR-100-2-layer** | | | | | |
| BACKBONE ONLY | — | — | $64.20 \pm 0.91$ | $51.00 \pm 1.28$ | $0.6652 \pm 0.0163$ |
| HYPATTN | — | — | $\mathbf{70.67 \pm 0.56}$ | $\mathbf{58.19 \pm 0.38}$ | $0.6740 \pm 0.0142$ |
| HYPNN | — | — | $69.02 \pm 0.47$ | $56.82 \pm 0.41$ | $0.5938 \pm 0.0202$ |
| MHNS | — | — | $65.34 \pm 0.86$ | $49.86 \pm 0.63$ | $0.6295 \pm 0.0143$ |
| U-HOP | — | — | $62.14 \pm 0.46$ | $45.74 \pm 0.44$ | $\mathbf{0.7195 \pm 0.0993}$ |
| **HAMNS (OURS)** | — | — | $70.12 \pm 0.57$ | $56.00 \pm 0.64$ | $0.6778 \pm 0.0193$ |
| **CIFAR-100-3-layer** | | | | | |
| BACKBONE ONLY | — | $72.75 \pm 1.89$ | $62.58 \pm 1.35$ | $50.79 \pm 0.82$ | $0.7023 \pm 0.0164$ |
| HYPATTN | — | $79.33 \pm 0.66$ | $68.68 \pm 0.78$ | $54.01 \pm 0.97$ | $0.6902 \pm 0.0240$ |
| HYPNN | — | $79.40 \pm 0.66$ | $\mathbf{68.84 \pm 0.95}$ | $54.09 \pm 1.05$ | $0.7123 \pm 0.0211$ |
| MHNS | — | $79.17 \pm 0.59$ | $68.08 \pm 0.84$ | $52.89 \pm 0.97$ | $\mathbf{0.7152 \pm 0.0256}$ |
| U-HOP | — | $75.11 \pm 0.51$ | $62.85 \pm 0.55$ | $45.93 \pm 0.83$ | $0.6996 \pm 0.0717$ |
| **HAMNS (OURS)** | — | $\mathbf{79.70 \pm 0.29}$ | $68.81 \pm 0.59$ | $\mathbf{54.27 \pm 0.47}$ | $0.7017 \pm 0.0658$ |
| **CIFAR-100-4-layer** | | | | | |
| BACKBONE ONLY | $87.51 \pm 0.73$ | $72.68 \pm 1.85$ | $60.02 \pm 1.02$ | $47.23 \pm 0.77$ | $0.7180 \pm 0.0230$ |
| HYPATTN | $90.13 \pm 0.48$ | $78.23 \pm 0.48$ | $67.74 \pm 0.93$ | $54.50 \pm 0.78$ | $0.6795 \pm 0.0143$ |
| HYPNN | $90.30 \pm 0.35$ | $78.72 \pm 0.59$ | $68.29 \pm 0.80$ | $\mathbf{55.93 \pm 0.88}$ | $0.6046 \pm 0.0149$ |
| MHNS | $89.39 \pm 0.29$ | $76.97 \pm 0.44$ | $65.56 \pm 0.42$ | $49.37 \pm 0.57$ | $0.5902 \pm 0.0218$ |
| U-HOP | $88.62 \pm 0.57$ | $75.46 \pm 0.58$ | $62.46 \pm 0.97$ | $45.44 \pm 0.94$ | $0.7154 \pm 0.0680$ |
| **HAMNS (OURS)** | $\mathbf{90.98 \pm 0.39}$ | $\mathbf{79.48 \pm 0.57}$ | $\mathbf{68.51 \pm 0.84}$ | $53.49 \pm 1.05$ | $\mathbf{0.7184 \pm 0.0254}$ |

perform them at the fine-grained level. U-Hop exhibits a trade-off: it attains relatively high `coph_corr` but at a clear cost in flat accuracy; on deeper hierarchies it still lags markedly behind hyperbolic decoders. We conjecture this depth-insensitive trend is tied to its two-stage protocol: U-Hop first learns a dataset-level kernel (independent of hierarchy depth), then trains Hopfield retrieval dynamics on top of the fixed kernel. Among the two hyperbolic baselines, HypAttn is particularly well suited to shallow hierarchies, giving the strongest performance when only a coarse–fine split is present, whereas HypNN gradually becomes the best fine-grained recognizer as the tree deepens, at the cost of weaker alignment between its predictions across levels. HAMNs (ours) offer the most balanced behaviour: already on the 3-layer hierarchy they match or exceed the baselines at all levels, and on the 4-layer hierarchy they simultaneously achieve the strongest high-level performance and the highest hierarchy-consistency, while keeping fine-grained accuracy competitive and only slightly below HypNN.

**Takeaway.** On deeper label trees, negatively curved retrieval provides a better accuracy–coherence trade-off than Euclidean Hopfield baselines (including kernelized U-Hop): Euclidean methods can improve cross-level coherence but do not match hyperbolic decoders in deep-hierarchy performance, while **HAMNs** achieve the best overall balance on the deepest hierarchy.

Beyond these main results, we provide two additional analyses in Appendix E.1. First, we show that our conclusions are robust under an alternative, feature-driven hierarchy

on CIFAR-100 (Appendix E.1.1). Second, we inspect the learned hyperbolic memories and find that radial coordinates and geodesic distances correlate well with the ground-truth tree structure (Appendix E.1.2).

### 4.2. Real-world hierarchical tasks: WordNet taxonomy and WN18RR

While CIFAR-100 provides a benchmark for synthetic label trees, many applications come with a fixed ontology. These WordNet-based experiments are intended to test the practical applicability of our method, rather than to serve as an exhaustive benchmark suite. We adopt relatively simple validation protocols. Nevertheless, the results show that HAMNs can be used as a plug-and-play retrieval module for realistic hierarchical tasks, while Euclidean MHN-style retrieval still clearly lags behind hyperbolic retrieval.

On **WordNet hypernym prediction**, inputs are SBERT-encoded synsets and the goal is to predict their immediate hypernyms. As shown in Table 2, under plain SBERT features, the Euclidean Hopfield baselines (MHN_Euc and U-Hop) perform much worse than the hyperbolic decoders: their mean accuracies are only $20.03\%$–$24.98\%$, whereas HypAttn, HypNN, and HAMNs reach $48.09\%$, $43.97\%$, and $47.25\%$, respectively. Their average graph distances are also much larger ($4.40$–$4.87$ vs. about $2.61$–$2.99$), indicating that a flat Euclidean energy struggles to exploit the multi-level taxonomy. Both hyperbolic attention (HypAttn) and our HAMNs achieve substantially higher accuracy and much smaller hierarchical error, with HypAttn

*Table 2.* Hypernym prediction on WordNet (nouns). We report test accuracy (%) and the average shortest-path distance.

| MODEL | ACC (%) | AVG. HIER. DIST. |
|---|---|---|
| MHN_EUC | 24.98 ± 1.89 | 4.40 ± 0.14 |
| U-HOP | 20.03 ± 8.24 | 4.87 ± 0.90 |
| HYPATTN | **48.09 ± 0.86** | **2.61 ± 0.08** |
| HYPNN | 43.97 ± 1.55 | 2.99 ± 0.11 |
| HAMNS | 47.25 ± 1.06 | 2.66 ± 0.06 |

*Table 3.* Hypernym prediction on WordNet (nouns) with an additional ontology embedding front-end (**+OntEuc**).

| MODEL | ACC (%) | AVG. HIER. DIST. |
|---|---|---|
| MHN_EUC + ONTEUC | 88.69 ± 1.28 | 0.473 ± 0.076 |
| U-HOP+ONTEUC | 87.40 ± 0.64 | 0.691 ± 0.076 |
| HYPATTN + ONTEUC | 96.58 ± 0.81 | 0.040 ± 0.008 |
| HYPNN + ONTEUC | 93.50 ± 1.52 | 0.082 ± 0.032 |
| HAMNS + ONTEUC | **96.97 ± 0.41** | **0.037 ± 0.005** |

slightly ahead and HAMNs remaining competitive.

When we equip *all* decoders with the same ontology encoder (**OntEuc**), trained as a lightweight MLP on the hypernym graph, every model improves dramatically (Table 3). Nevertheless, HypAttn and HAMNs still clearly outperform MHN_Euc and U-Hop, with HAMNs achieving the best accuracy (96.97%) and the smallest average hierarchical distance (0.037). This shows that ontology embeddings and hyperbolic retrieval are complementary rather than interchangeable. Detailed settings and additional results are provided in Appendix E.4.

On **WN18RR link prediction**, a knowledge-graph completion benchmark built from the same WordNet ontology, we reuse a single encoder for entities and relations and only swap the decoder. All three hyperbolic decoders (HypAttn, HypNN, HAMNs) substantially outperform Euclidean MHN_Euc on MRR and Hits@1/3/10, confirming that hyperbolic geometry is advantageous on strongly hierarchical graphs. Among them, HypNN obtains the best MRR and Hits@1/3, while HAMNs remains very close and achieves the best Hits@10. This indicates that the proposed CCCP-based hyperbolic energy minimization is practically effective as a competitive drop-in decoder. Detailed settings and additional results are provided in Appendix E.5.

**Takeaway.** Across these two WordNet-based tasks, HAMNs act as a robust hyperbolic retrieval module that can be plugged into standard pipelines, clearly outperform Euclidean MHN-style retrieval on deep hierarchical structures, and remain competitive with other hyperbolic decoders.

### 4.3. Weak/shallow hierarchy tasks: MIL and molecular property prediction

**Multi–Instance Learning (MIL)** We evaluate on three classical MIL datasets—**Tiger**, **Elephant**, and **Fox**—to probe the bag–instance regime without instance-level labels (Dietterich et al., 1997), using the standard splits introduced by (Ilse et al., 2018; Küçükaşcı & Baydoğan, 2018; Carbonneau et al., 2018). We plug our **HypPooling** into the MIL pipeline: embedded instances serve as stored memories ($Y$), while a fixed set of learnable query vectors acts as state (query) patterns ($R$) on the same Poincaré ball; retrieval is performed via hyperbolic attention and on-manifold updates. See Appendix D for the layer design and Appendix E.2 for training protocol and hyperparameters. We compare against representative MIL baselines (e.g., attention-MIL (Ilse et al., 2018), mi-Net variants (Carbonneau et al., 2018)). Results show *competitive* overall performance and new SOTA on **Fox**; elsewhere gaps to MHNs are modest (Table 8).

**Molecular property prediction** Experiments on four MoleculeNet datasets—**HIV**, **BACE** (Subramanian et al., 2016), **BBBP** (Martins et al., 2012), and **SIDER** (Kuhn et al., 2016)—probe the weak/shallow–hierarchy regime. The proposed **HypLayer** is inserted into standard pipelines: training samples serve as stored memories ($Y$), inputs as queries ($R$), followed by hyperbolic embedding and retrieval (exact layer design, training protocol, and hyperparameters are detailed in Appendix D). Comparisons cover representative baselines (classical ML, GNNs, and MHNs). This approach yields *competitive* performance and sets SOTA on **BBBP** and **SIDER** (Appendix E.3); margins over MHNs are small, consistent with the weak-hierarchy hypothesis.

### 4.4. Computational cost and performance

Using PyTorch's profiler on a single NVIDIA RTX 4090 GPU, we compare MHNs with the three hyperbolic decoders (HypAttn, HypNN, HAMNs) under the CIFAR-100 hierarchical classification and WN18RR link-prediction setups. For a fixed backbone and hidden size, HAMNs typically use fewer parameters and FLOPs than Euclidean MHNs, but incur higher runtime and peak memory because of hyperbolic operations (Möbius addition, exponential/logarithmic maps, Riemannian gradient transforms) and the associated memory-access overhead. Among the hyperbolic variants, HypAttn is the most expensive due to repeated geodesic distance and Fréchet-mean computations, HypNN is the most lightweight, and HAMNs lies in between. (See Appendix E.6)

### 4.5. Ablations: curvature and number of stored patterns

**Summary.** A *moderate curvature* provides the best trade-off; extremes are harmful (too small under-expresses hierar-

chy, too large degrades accuracy). Varying the *number of stored patterns* causes only small overall fluctuations: over-sized memories slightly reduce top-level accuracy, moderate increases help mid-level, and fine-level performance peaks at higher counts. In practice, we recommend *moderate curvature* and a *modest memory size*. (See Appendix E.7)

## 5. Related Work

**Associative memories and modern Hopfield networks**
Classical Hopfield networks were first introduced by Hopfield (1982), who viewed the network as a recurrent dynamical system whose energy function stores binary patterns as attractors and retrieves them via energy minimization. Continuous variants later extended the state space to real-valued vectors (Hopfield, 1984). Dense associative memory and modern Hopfield-type networks further generalized the original Hopfield energy (Krotov & Hopfield, 2016). Krotov and Hopfield (Krotov & Hopfield, 2021) provide a systematic analysis of the modern Hopfield-network family and its large associative-memory capacity.

In contemporary machine learning, *modern Hopfield networks* (MHNs) are used as differentiable associative-memory layers inside deep architectures. For example, Ramsauer et al. (2021) derive one-step-convergent energies, analyze memory capacity, and clarify the correspondence between MHNs and attention. Application-oriented Hopfield layers have also been used for large-scale biological sequence and immune repertoire classification (Widrich et al., 2020). Subsequent work extends this framework along several axes, including kernelized or learnable query–memory similarities that improve capacity and robustness (Wu et al., 2024), sparse and structured variants based on sparse entropic regularizers or Fenchel–Young losses and SparseMAP (Hu et al., 2023; Santos et al., 2024), and latent-structured Hopfield networks that embed continuous attractor dynamics into autoencoder latent spaces for semantic association and episodic-style memory (Li et al., 2026).

In this paper we adopt this *machine-learning perspective*: we treat MHNs as pluggable energy-based retrieval layers and focus on moving the Hopfield energy and dynamics to negatively curved manifolds, in order to study how geometry and optimization jointly affect hierarchical retrieval, rather than proposing new similarity kernels or a biologically realistic circuit.

**Hyperbolic Geometry** (Nickel & Kiela, 2017) first proposed using hyperbolic space to learn hierarchical representations of symbolic data, such as text and graphs, by embedding them into the Poincaré ball model. Since then, the application of hyperbolic geometry has been explored in various domains. (Ganea et al., 2018) introduced hyperbolic neural network layers, which have enabled the development of hybrid architectures such as hyperbolic graph convolutional networks (Chami et al., 2019), hyperbolic variational autoencoders (Ovinnikov, 2020), and hyperbolic attention networks (Gülçehre et al., 2019). In addition, Hyperbolic Neural Networks++ (Shimizu et al., 2021) provides a unified construction of fundamental neural components in the Poincaré ball, including fully-connected, convolutional, and attention mechanisms. These architectures have been successfully applied to tasks such as deep metric learning, object detection , and natural language processing. Beyond practical applications, theoretical investigations into hyperbolic spaces and their models have also deepened, demonstrating properties such as lower representation distortion (De Sa et al., 2018), better generalization ability (Suzuki et al., 2021), and stronger representation power in low-dimensional spaces (De Sa et al., 2018). Unlike prior implicit uses of hyperbolic geometry, energy-based Hopfield retrieval is carried out directly in hyperbolic space, broadening applicability to hierarchical representation learning.

## 6. Discussion

This work presents Hyperbolic Associative Memory Networks (HAMNs), extending modern Hopfield networks from Euclidean to hyperbolic geometry with retrieval formulated as geodesic-based energy minimization. Our results validate the hierarchy-sensitivity hypothesis: HAMNs outperform Euclidean MHNs on deeply hierarchical data but perform comparably on flat/shallow data, and remain competitive with other hyperbolic decoders on deep label spaces. Notably, ontology embeddings act as a label prior, while the hyperbolic memory layer modulates the exploitation of hierarchical geometry, forming complementary components. HAMNs thus serve as a generic module for tasks requiring hierarchical pattern storage/retrieval, enabling consistent gains where multi-level structures exist. This study has several limitations. It focuses on differentiable associative memory modules for machine learning, excluding neurobiological memory models. Methodologically, we only implement HAMNs for constant negative curvature (via the Poincaré ball model), with other non-Euclidean geometries unexplored. Practically, despite lower theoretical FLOPs and parameters than Euclidean MHNs, hyperbolic GPU operations introduce significant overhead, increasing wall-clock time and peak memory usage. Our experiments assess hierarchical geometry exploitation, with scalability to larger models and modalities reserved for future work.

## Acknowledgments

This work was supported by the Zhejiang Province Leading Earth Goose Program-Key R&D Program of Zhejiang Province 2026LDC01028(XC)

## Impact Statement

This paper presents work whose goal is to advance the field of Machine Learning. There are many potential societal consequences of our work, none which we feel must be specifically highlighted here.

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

# A. Hyperbolic Energy-based Optimization Framework

## A.1. Minkowski inner product and hyperbolic distance

We briefly recall how the Minkowski inner product controls hyperbolic distance in the Lorentz model, and how this justifies our design choice of using $-\langle \cdot, \cdot \rangle_L$ as the association term in the HAMNs energy. Formally, the $d$-dimensional hyperbolic space in Lorentz form is

$$\mathbb{H}^d = \left\{ u \in \mathbb{R}^{d+1} : \langle u, u \rangle_L = -1, \ u_0 > 0 \right\}, \tag{12}$$

where the Minkowski inner product is defined as

$$\langle u, v \rangle_L = -u_0 v_0 + \sum_{i=1}^{d} u_i v_i, \qquad u, v \in \mathbb{R}^{d+1}. \tag{13}$$

It is well known that the geodesic distance on $\mathbb{H}^d$ can be expressed as

$$d_{\mathbb{H}}(u, v) = \operatorname{arcosh}\left(-\langle u, v \rangle_L\right), \qquad u, v \in \mathbb{H}^d. \tag{14}$$

(Nickel & Kiela, 2018; Ratcliffe, 2006)

Since $\operatorname{arcosh}(\cdot)$ is strictly increasing on $[1, +\infty)$, Eq. (14) implies a simple monotonic relationship: reducing the hyperbolic distance $d_{\mathbb{H}}(u, v)$ is equivalent to reducing its argument $-\langle u, v \rangle_L$, or, equivalently, *increasing* the Minkowski inner product $\langle u, v \rangle_L$. In other words, the negative Minkowski inner product $-\langle u, v \rangle_L$ behaves as a curvature-aware dissimilarity measure that is strictly order-equivalent to the hyperbolic distance.

**Implication for the HAMNs energy**     In our associative memory module we use an energy of the form

$$E(z, \Xi) = -\frac{1}{\theta} \log \left( \sum_{m \in \Xi} \exp(\theta \langle z, m \rangle_L) \right) + \lambda \, \Phi(z), \tag{15}$$

where $\Xi$ denotes the set of memory vectors, $\Phi(z)$ is an intrinsic regularizer, $\lambda \geq 0$ is its weight, and $\theta > 0$ is an inverse-temperature (temperature-scaling) parameter. Because of (14), increasing $\langle z, m \rangle_L$ is equivalent to decreasing the hyperbolic distance $d_{\mathbb{H}}(z, m)$. The log-sum-exp term $-\frac{1}{\theta} \log \sum_{m \in \Xi} \exp(\theta \langle z, m \rangle_L)$ is a smooth approximation of $-\max_{m \in \Xi} \langle z, m \rangle_L$, thus minimizing it encourages $z$ to move closer (in the hyperbolic metric) to its most relevant memories.

**Time-like vs space-like components**     Any point $u \in \mathbb{H}^d$ in the Lorentz model can be written as $u = (u_0, \bar{u}) \in \mathbb{R}^{1+d}$ with $\langle u, u \rangle_L = -1$, which implies $u_0 = \cosh r_u$ and $\|\bar{u}\|_2 = \sinh r_u$ for some radial coordinate $r_u = d_{\mathbb{H}}(u, o)$. For two points $u, v \in \mathbb{H}^d$ with radii $r_u, r_v$ and angular separation $\varphi$ between their spatial parts $\bar{u}, \bar{v}$, the Minkowski inner product decomposes as

$$\langle u, v \rangle_L = -\cosh r_u \cosh r_v + \sinh r_u \sinh r_v \cos \varphi.$$

Thus the "time-like" component $-\cosh r_u \cosh r_v$ and the "space-like" component $\sinh r_u \sinh r_v \cos \varphi$ contribute with opposite signs. For fixed angular term $\cos \varphi$, increasing the radii $r_u, r_v$ makes $\langle u, v \rangle_L$ more negative, so that the surrogate dissimilarity $-\langle u, v \rangle_L$ grows roughly with the sum $r_u + r_v$. Consequently, the Minkowski-based energy is more sensitive to radial separation than to purely angular differences, which naturally encourages the model to encode hierarchical depth along the radial direction.

**Depth–radius correlation**     A further geometric intuition comes from the radial coordinate $\|z\|_{\mathbb{H}}$, defined as the hyperbolic distance from a reference origin $o \in \mathbb{H}^d$, i.e., $\|z\|_{\mathbb{H}} = d_{\mathbb{H}}(z, o)$. On the hierarchical CIFAR-100 experiment, we can directly probe this geometry: for each stored class memory we record its tree depth (top / super / coarse / fine) and its hyperbolic radius. As detailed in Appendix E.1.2, the average radius increases monotonically from top to fine classes, and the Spearman correlation between discrete depth and $\|z\|_{\mathbb{H}}$ is $\rho = 0.57$ (with $p \approx 1.6 \times 10^{-12}$). This provides empirical evidence that optimizing the Minkowski-based hyperbolic energy naturally organizes concepts according to hierarchical depth in the learned representation.

### A.2. Illustrative hyperbolic energy landscape

To complement the analytic discussion above, Fig. 1 visualizes the retrieval energy $E(\xi)$ of a toy HAMNs on a 2D Poincaré disk. We embed a three-level hierarchy with *top*, *coarse*, and *fine* attractors placed at increasing radii. Red, orange, and cyan dots indicate the centers of the top-, coarse-, and fine-level patterns, respectively. The color map shows the value of $E(\xi)$ on a dense grid inside the disk.

We observe that (i) top-level patterns near the center give rise to broad and smooth basins, corresponding to coarse semantic groups; (ii) coarse-level patterns at intermediate radii carve out more localized wells while remaining separated; and (iii) fine-level patterns near the boundary induce sharp basins that densely tile the outer region without collapsing into each other. Visually, the energy field therefore changes systematically across hierarchical levels: as we move radially outward, basins become progressively sharper and more numerous. This illustrates how negative curvature allows us to allocate increasingly fine-grained attractors along the radial direction while preserving a smooth global energy landscape, in line with our capacity and margin discussion in App. B.

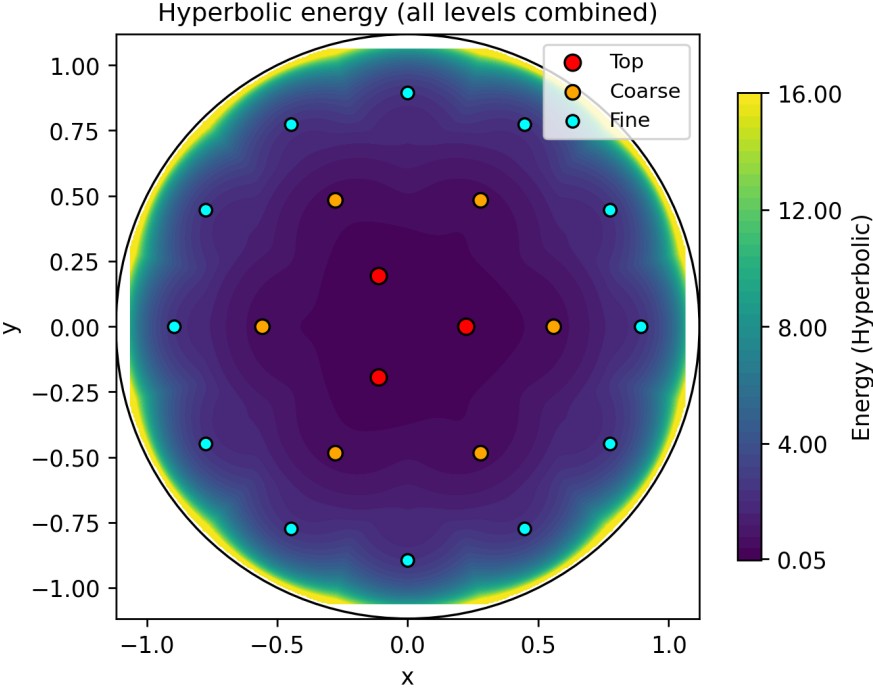

*Figure 1.* Hyperbolic retrieval energy on a 2D Poincaré disk for a toy three-level hierarchy. Red, orange, and cyan dots denote top-, coarse-, and fine-level attractors, respectively. Darker regions correspond to lower energy.

### A.3. Bounding the Energy Function

We consider the energy

$$E(\xi) \;=\; -\frac{1}{\theta}\log\left(\sum_{i=1}^{N}\exp\big(\theta\,\langle x_i, \xi\rangle_M\big)\right) \;+\; \frac{1}{2}\,d_{\mathcal{M}}(\xi,p)^2, \tag{16}$$

where the (hyperbolic) similarity is $\langle x,y\rangle_M := -\cosh\big(d_{\mathcal{M}}(x,y)\big)$ on a complete, simply connected Riemannian manifold $(\mathcal{M}, g)$ of constant negative curvature $-c$.

**Setup and notation**  Fix a base point $p \in \mathcal{M}$ and define

$$r_i := d_{\mathcal{M}}(x_i, p), \qquad r := d_{\mathcal{M}}(\xi, p).$$

Let $M_r := \max_i r_i$. We assume optimization is restricted (by standard clipping/projection) to a geodesic ball around $p$, i.e., $r \le R_r$. Here $N$ is the number of stored patterns and $\theta > 0$ the inverse temperature.

### A.3.1. BOUNDING THE SIMILARITY

By the triangle inequality,

$$|r_i - r| \ \leq \ d_{\mathcal{M}}(x_i, \xi) \ \leq \ r_i + r.$$

Since $\cosh(\cdot)$ is strictly increasing on $[0, \infty)$ and $\langle x_i, \xi \rangle_M = -\cosh d_{\mathcal{M}}(x_i, \xi)$, we obtain for each $i$

$$-\cosh(r_i + r) \ \leq \ \langle x_i, \xi \rangle_M \ \leq \ -\cosh(|r_i - r|). \tag{17}$$

Consequently, using $r_i \leq M_r$ and $r \leq R_r$,

$$\boxed{-\cosh(M_r + R_r) \ \leq \ \langle x_i, \xi \rangle_M \ \leq \ -1} \qquad (\forall i), \tag{18}$$

because $\cosh(0) = 1$ and $|r_i - r|$ can be as small as $0$.

### A.3.2. BOUNDING THE ENERGY

Write $E(\xi) = E_{\text{cave}}(\xi) + E_{\text{cvx}}(\xi)$ with

$$E_{\text{cave}}(\xi) = -\frac{1}{\theta} \log \sum_{i=1}^{N} e^{\theta z_i}, \quad z_i := \langle x_i, \xi \rangle_M, \qquad E_{\text{cvx}}(\xi) = \tfrac{1}{2} d_{\mathcal{M}}(\xi, p)^2.$$

For any $\theta > 0$, the log-sum-exp bounds yield

$$\max_i z_i \ \leq \ \frac{1}{\theta} \log \sum_i e^{\theta z_i} \ \leq \ \max_i z_i + \frac{\log N}{\theta} \quad \implies \quad -\max_i z_i - \frac{\log N}{\theta} \ \leq \ E_{\text{cave}}(\xi) \ \leq \ -\max_i z_i.$$

From (18) we have $\max_i z_i \in \big[-\cosh(M_r + R_r), \ -1\big]$. Hence

$$\boxed{1 - \frac{\log N}{\theta} \ \leq \ E_{\text{cave}}(\xi) \ \leq \ \cosh(M_r + R_r)}. \tag{19}$$

For the convex part (squared distance to $p$),

$$0 \ \leq \ E_{\text{cvx}}(\xi) = \tfrac{1}{2} d_{\mathcal{M}}(\xi, p)^2 \ \leq \ \tfrac{1}{2} R_r^2. \tag{20}$$

### A.3.3. FINAL BOUNDS

Combining (19) and (20) yields

$$\boxed{1 - \frac{\log N}{\theta} \ \leq \ E(\xi) \ \leq \ \cosh(M_r + R_r) \ + \ \tfrac{1}{2} R_r^2}. \tag{21}$$

The constants depend only on the maximal radial extents of memories and states $(M_r, R_r)$ and the inverse temperature $\theta$, but *not* on the specific hyperbolic model. Thus the boundedness of $E$—and hence the numerical stability of CCCP or Riemannian-gradient iterations—holds uniformly across all constant-curvature hyperbolic realizations.

## A.4. Optimization of the Energy Function via CCCP

### A.4.1. CONCAVITY/CONVEXITY ON HADAMARD MANIFOLDS

We work on a Hadamard manifold $(\mathcal{M}, g)$ with constant negative curvature. Write

$$E(\xi) \ = \ E_{\text{cvx}}(\xi) \ + \ E_{\text{cave}}(\xi), \qquad E_{\text{cvx}}(\xi) = \tfrac{1}{2} d_{\mathcal{M}}(\xi, p)^2, \quad E_{\text{cave}}(\xi) = -\tfrac{1}{\theta} \log \sum_{i=1}^{N} e^{\theta\, s_i(\xi)},$$

where $s_i(\xi) := \langle x_i, \xi \rangle_M = -\cosh(d_{\mathcal{M}}(x_i, \xi))$. It is known that $d_{\mathcal{M}}(\cdot, \cdot)$ is geodesically convex on Hadamard manifolds; since $\cosh$ is convex and strictly increasing on $[0, \infty)$, the composition $\cosh \circ d_{\mathcal{M}}$ is geodesically convex, hence $s_i(\xi) =$

$-\cosh(d_{\mathcal{M}}(x_i, \xi))$ is *geodesically concave*. For the concavity of $E_{\text{cave}}(\xi) = -\operatorname{lse}_\theta(\{s_i(\xi)\}_i)$, let $F(\xi) = \operatorname{lse}_\theta(\{s_i(\xi)\}_i)$. For any unit tangent vector $u \in T_\xi\mathcal{M}$, the Riemannian Hessian follows the standard second-derivative formula for log-sum-exp composites:

$$\operatorname{Hess}_\xi F[u, u] = \sum_{i=1}^N w_i(\xi) \operatorname{Hess}_\xi s_i[u, u] + \theta \operatorname{Var}_{w(\xi)}(\langle \operatorname{grad} s_i(\xi), u \rangle_g),$$

where $w_i(\xi) = \frac{e^{\theta s_i(\xi)}}{\sum_j e^{\theta s_j(\xi)}}$. Each $s_i$ is geodesically concave, so $\operatorname{Hess}_\xi s_i[\cdot, \cdot] \preceq 0$; the second term is a nonnegative variance term. Hence

$$\operatorname{Hess}_\xi(-F)[u, u] = -\sum_{i=1}^N w_i(\xi) \operatorname{Hess}_\xi s_i[u, u] - \theta \operatorname{Var}_{w(\xi)}(\langle \operatorname{grad} s_i(\xi), u \rangle_g)$$

is the sum of a positive semidefinite term and a negative semidefinite term.

**Regional nature of the concavity condition.**    The above decomposition also clarifies why the concavity of $E_{\text{cave}} = -F$ should be understood as a local and regional property, rather than as a global guarantee. In multi-memory competition regions, the softmax distribution is not collapsed to a single memory, and the variance term in the Hessian can dominate the positive semidefinite contribution induced by $-\sum_i w_i \operatorname{Hess} s_i$. In such regions, $E_{\text{cave}}$ can be locally geodesically concave. By contrast, near a dominant attractor where $w_k(\xi) \to 1$ for some memory $x_k$, the variance term tends to zero and the log-sum-exp behaves like a single similarity term. In this limiting regime, $-F$ may become locally geodesically convex rather than concave. Therefore, the CCCP/MM interpretation used in this paper should be read as a regional majorization argument, most appropriate in multi-memory competition regions, and not as an unconditional global concavity claim over the whole optimization trajectory.

On a bounded geodesic ball, assume there exist constants $0 < \underline{\kappa} \le \overline{\kappa}$ such that $\underline{\kappa}I \preceq -\operatorname{Hess}_\xi s_i \preceq \overline{\kappa}I$ for all $i$ and all $\xi$ in the ball (i.e., $-s_i$ is uniformly strongly geodesically convex on this bounded region), and let $L := \max_{i,\xi} \|\operatorname{grad} s_i(\xi)\|_g < \infty$. Then for any unit $u \in T_\xi\mathcal{M}$,

$$\underline{\kappa} \le -\sum_i w_i(\xi) \operatorname{Hess}_\xi s_i[u, u] \le \overline{\kappa}, \qquad 0 \le \operatorname{Var}_{w(\xi)}(\cdot) \le L^2,$$

and therefore

$$\underline{\kappa} - \theta L^2 \le \operatorname{Hess}_\xi(-F)[u, u] \le \overline{\kappa}.$$

In particular, when $0 < \theta \le \underline{\kappa}/L^2$, the lower bound is nonnegative, so $E_{\text{cave}}(\xi) = -F(\xi)$ is (at least locally) geodesically *convex* rather than concave.

For the standard CCCP/MM monotonicity guarantee, a sufficient condition is geodesic concavity of $E_{\text{cave}}$ on the region visited by the iterates. Such concavity can be guaranteed in regions where the variance term is uniformly non-degenerate: if there exists $v_0 > 0$ such that for all $\xi$ in the region and all unit $u$,

$$\operatorname{Var}_{w(\xi)}(\langle \operatorname{grad} s_i(\xi), u \rangle_g) \ge v_0,$$

then whenever

$$\theta \ge \overline{\kappa}/v_0,$$

we obtain $\operatorname{Hess}_\xi(-F)[u, u] \le 0$, i.e., $E_{\text{cave}}$ is geodesically concave on that region and $E = E_{\text{cvx}} + E_{\text{cave}}$ satisfies the local "convex + concave" requirement for CCCP/MM. This condition is not expected to hold uniformly near all final attractors, because the variance term may vanish when the softmax distribution becomes sharply peaked.

**Riemannian DC / CCCP viewpoint**    Classical DC programming and CCCP are formulated in Euclidean spaces for functions $f = g - h$ with $g$ convex and $h$ convex; in our setting the ambient space is a Hadamard manifold and convexity is understood in the *geodesic* sense. The role of the Euclidean linearization $h(x) \approx h(x^{(t)}) + \langle \nabla h(x^{(t)}), x - x^{(t)} \rangle$ is played by the Riemannian first–order expansion

$$h(\xi) \approx h(\xi^{(t)}) + \langle \operatorname{grad} h(\xi^{(t)}), \log_{\xi^{(t)}}(\xi) \rangle_{\xi^{(t)}},$$

where $\log_{\xi^{(t)}}(\cdot)$ is the Riemannian logarithm map. Replacing Euclidean convexity and linearization by geodesic convexity and the logarithm map is the only change; the surrogate $Q(\cdot \mid \xi^{(t)})$ in (22) is therefore the natural Riemannian analogue of the CCCP surrogate on Hadamard manifolds (see also standard references on Riemannian optimization, e.g. (Bonnabel, 2013)).

A.4.2. EMPIRICAL LOCAL CONCAVITY PROBE

Because the concavity of $E_{\mathrm{cave}}$ is only a local and regional property, we additionally probe whether the iterates visited in training lie in regions where this local concavity approximation is empirically valid. At each pre-update state $q$ along the retrieval trajectory, we sample several unit tangent directions $u \in T_q\mathcal{M}$ and evaluate the local second-order difference of the concave term:

$$\Delta_\varepsilon^2 E_{\mathrm{cave}}(q; u) = \frac{E_{\mathrm{cave}}(\exp_q(\varepsilon u)) - 2E_{\mathrm{cave}}(q) + E_{\mathrm{cave}}(\exp_q(-\varepsilon u))}{\varepsilon^2}.$$

We regard local concavity as satisfied at $q$ if the worst sampled second difference is no larger than a threshold $\tau_{\mathrm{conc}}$, namely

$$\max_{u \in \mathcal{U}(q)} \Delta_\varepsilon^2 E_{\mathrm{cave}}(q; u) \leq \tau_{\mathrm{conc}}.$$

where $\mathcal{U}(q)$ denotes the sampled set of unit tangent directions.

We apply this probe in the CIFAR-100 hierarchical classification setting, where the retrieval module is repeatedly invoked during end-to-end training. To avoid masking the phenomenon by using too few retrieval updates, we run the probe with 8 retrieval steps and collect local tests throughout training. In total, we obtain approximately $3.2 \times 10^7$ local second-difference probes. We do not observe any violation under this criterion, i.e., the empirical nonconcave percentage is $0\%$. This result does not constitute a global proof of concavity. Rather, it indicates that, along the optimization trajectories visited by our implementation, the iterates typically remain in regions where the local concavity condition required by the CCCP/MM interpretation is empirically stable.

A.4.3. CCCP LINEARIZATION AND SURROGATE

Let $\xi^{(t)}$ be the current iterate. When $E_{\mathrm{cave}}$ is geodesically concave on a geodesically convex neighborhood $\mathcal{R}$ containing $\xi^{(t)}$ (e.g., under the non-degenerate variance condition in Sec. A.4.1), the Riemannian first-order expansion gives an upper bound: for all $\xi \in \mathcal{R}$,

$$E_{\mathrm{cave}}(\xi) \leq E_{\mathrm{cave}}(\xi^{(t)}) + \left\langle a^{(t)}, \log_{\xi^{(t)}}(\xi) \right\rangle_{\xi^{(t)}}, \qquad a^{(t)} := \mathrm{grad}\, E_{\mathrm{cave}}(\xi^{(t)}).$$

(Outside such a concavity region, the inequality direction may reverse.)

Thus the "bound-minimization" surrogate for CCCP is

$$Q\left(\xi \mid \xi^{(t)}\right) = \tfrac{1}{2} d_\mathcal{M}(\xi, p)^2 + \left\langle a^{(t)}, \log_{\xi^{(t)}}(\xi) \right\rangle_{\xi^{(t)}} + \mathrm{const}. \tag{22}$$

A.4.4. EXPLICIT UPDATE VIA A TANGENT-SPACE SURROGATE (WITH PARALLEL TRANSPORT)

On a Hadamard manifold, the surrogate (22) is strongly geodesically convex due to the term $\tfrac{1}{2} d_\mathcal{M}(\xi, p)^2$, but its exact minimizer generally does not admit a simple closed form because $\xi \mapsto \log_{\xi^{(t)}}(\xi)$ is nonlinear on curved manifolds. We therefore use an explicit update obtained by minimizing a first-order tangent-space surrogate at the reference point $p$.

Let $b^{(t)} := \mathrm{PT}_{\xi^{(t)} \to p}\left(a^{(t)}\right) \in T_p\mathcal{M}$ and consider the proxy objective on $T_p\mathcal{M}$:

$$\widetilde{Q}(v \mid \xi^{(t)}) = \tfrac{1}{2}\|v\|_g^2 + \langle b^{(t)}, v \rangle_p + \mathrm{const}, \qquad v \in T_p\mathcal{M},$$

whose minimizer is $v^\star = -b^{(t)}$. We then update by the exponential map:

$$\xi^{(t+1)} = \exp_p\left(-\eta\, \mathrm{PT}_{\xi^{(t)} \to p}\left(a^{(t)}\right)\right), \qquad \eta \in (0, 1]. \tag{23}$$

When $\eta = 1$, this is the exact minimizer of $\widetilde{Q}(\cdot \mid \xi^{(t)})$, and it coincides with the classical Euclidean CCCP step in the flat case.

A.4.5. SOFTMAX WEIGHTS AND RIEMANNIAN GRADIENT OF THE CONCAVE TERM

Let $w_i^{(t)} = \frac{\exp(\theta\, s_i(\xi^{(t)}))}{\sum_j \exp(\theta\, s_j(\xi^{(t)}))}$. By the chain rule together with §A.5, (26), we obtain

$$a^{(t)} = \operatorname{grad} E_{\text{cave}}(\xi^{(t)}) = -\sum_{i=1}^N w_i^{(t)} \operatorname{grad}_\xi s_i(\xi)\Big|_{\xi=\xi^{(t)}}. \tag{24}$$

**Convergence note**  The classical CCCP/MM monotonicity argument requires a decomposition $E = E_{\text{cvx}} + E_{\text{cave}}$ where $E_{\text{cvx}}$ is geodesically convex and $E_{\text{cave}}$ is geodesically concave on a geodesically convex region $\mathcal{R} \subset \mathcal{M}$ containing the iterates. In our setting, $E_{\text{cvx}}(\xi) = \frac{1}{2}d_{\mathcal{M}}(\xi, p)^2$ is geodesically convex on any Hadamard manifold, while geodesic concavity of $E_{\text{cave}}(\xi) = -\frac{1}{\theta}\log\sum_i e^{\theta s_i(\xi)}$ generally holds only locally/regionally (e.g., under the non-degenerate variance condition in Sec. A.4.1 together with a sufficiently large $\theta$). When $\xi^{(t)} \in \mathcal{R}$ and $E_{\text{cave}}$ is geodesically concave on $\mathcal{R}$, the first-order Riemannian expansion yields a valid upper bound $E_{\text{cave}}(\xi) \leq E_{\text{cave}}(\xi^{(t)}) + \langle \operatorname{grad} E_{\text{cave}}(\xi^{(t)}), \log_{\xi^{(t)}}(\xi)\rangle$ for all $\xi \in \mathcal{R}$, and hence the surrogate $Q(\cdot \mid \xi^{(t)})$ majorizes $E$ on $\mathcal{R}$. If, in addition, $E$ is bounded below and the sublevel set $\{\xi \in \mathcal{R} : E(\xi) \leq E(\xi^{(0)})\}$ is compact, then exact minimization of $Q(\cdot \mid \xi^{(t)})$ produces a sequence $\{\xi^{(t)}\}$ with monotonically nonincreasing energies $E(\xi^{(t)})$ that converge to a finite limit, and every accumulation point $\xi^\star$ is a Riemannian stationary point $\operatorname{grad} E(\xi^\star) = 0$. Outside such a concavity/majorization region, the surrogate may fail to be a global upper bound, so monotonic decrease is not guaranteed; nevertheless, we typically observe stable behavior in practice. As in Euclidean CCCP (Yuille & Rangarajan, 2001), we do not claim that the whole sequence $\{\xi^{(t)}\}$ converges, only energy monotonicity (when the above conditions hold) and stationary accumulation points. The empirical probe in Appendix A.4.2 further examines whether the iterates visited in training lie in such local concavity regions

## A.5. Riemannian Gradient of the Concave Term

Consider the hyperbolic similarity

$$s_M(x, y) := \langle x, y\rangle_M = -\cosh\bigl(d_{\mathcal{M}}(x, y)\bigr).$$

Let $d_{\mathcal{M}}$ denote the geodesic distance and $\log_x : \mathcal{M} \to T_x\mathcal{M}$ the Riemannian logarithm at $x$. On a Hadamard manifold, for any $x \neq y$,

$$\operatorname{grad}_x d_{\mathcal{M}}(x, y) = -\frac{\log_x(y)}{\|\log_x(y)\|_g}, \tag{25}$$

where $\|\cdot\|_g$ is the norm induced by $g$ on $T_x\mathcal{M}$.

**Chain rule for the similarity gradient**  By the chain rule,

$$\operatorname{grad}_x s_M(x, y) = -\sinh\bigl(d_{\mathcal{M}}(x, y)\bigr)\operatorname{grad}_x d_{\mathcal{M}}(x, y),$$

and substituting (25) yields

$$\boxed{\operatorname{grad}_x s_M(x, y) = \sinh\bigl(d_{\mathcal{M}}(x, y)\bigr)\frac{\log_x(y)}{\|\log_x(y)\|_g}.} \tag{26}$$

The gradient points along the unit tangent from $x$ to $y$ with magnitude $\sinh(d_{\mathcal{M}}(x, y))$.

**Riemannian gradient of $E_{\text{cave}}$**  Let $s_i(\xi) = \langle x_i, \xi\rangle_M$ and $w_i(\xi) = \frac{e^{\theta s_i(\xi)}}{\sum_j e^{\theta s_j(\xi)}}$. Then

$$\boxed{\operatorname{grad} E_{\text{cave}}(\xi) = -\sum_{i=1}^N w_i(\xi)\operatorname{grad}_\xi s_i(\xi) = -\sum_{i=1}^N w_i(\xi)\sinh\bigl(d_{\mathcal{M}}(x_i, \xi)\bigr)\frac{\log_\xi(x_i)}{\|\log_\xi(x_i)\|_g}.} \tag{27}$$

**Coordinate gradient (Poincaré ball example)**  If the chosen coordinates are conformal (e.g., the Poincaré ball), then $g(\xi) = \lambda(\xi)^2 I$ with $\lambda(\xi) = \frac{2}{1-c\|\xi\|^2}$. The Euclidean (coordinate) gradient $\nabla_\xi$ and the Riemannian gradient $\operatorname{grad}_\xi$ satisfy

$$\operatorname{grad}_\xi f = G(\xi)^{-1}\nabla_\xi f = \lambda(\xi)^{-2}\nabla_\xi f, \qquad \text{equivalently} \quad \nabla_\xi f = G(\xi)\operatorname{grad}_\xi f = \lambda(\xi)^2\operatorname{grad}_\xi f. \tag{28}$$

Plugging (27) into (28) yields an implementation-ready Euclidean gradient expression.

### A.6. Optional Neural Computation Analogies

**(i) About** (7) **(softmax weights).** Equation (7) represents the Gibbs/Luce choice rule, which can be neurally implemented similarly to *competition-suppression* circuits with *divisive normalization* or *finite-temperature winner-takes-all* (WTA): excitatory convergence generates a "similarity-driven" effect, while global fast suppression implements normalization. $w_i^{(t)}$ can be interpreted as sampling/confidence coding of memory items at finite temperature (in line with normalization and probability sampling perspectives in the cortex, as discussed in (Carandini & Heeger, 2012; Luce et al., 1959)).

**(ii) About** (11) **(Riemannian gradient).** Equation (11) shows that the Riemannian gradient of the concave term is a weighted sum of local quantities $\mathrm{grad}_\xi \langle x_i, \xi \rangle_M$. Each term depends solely on the "presynaptic pattern $x_i$" and the "postsynaptic state $\xi$," which aligns with *local (anti-)Hebbian driving + gating*: the probability $w_i^{(t)}$ serves as a gating coefficient and component. This is functionally similar to the *contrastive Hebb/equilibrium propagation* in energy models (Movellan, 1991; Scellier & Bengio, 2017). From an information geometry perspective, the Riemannian gradient corresponds to the *natural gradient* (Amari, 1998) on the manifold, which is compatible with the neural efficiency perspective that "intrinsic metrics determine the update direction."

**(iii) About** (9) **(closed-form CCCP step).** Equation (9) describes the "velocity" $v^{(t)}$ obtained at $\xi^{(t)}$ being parallel transported to the origin and then updated by one step through the exponential map. In the Poincaré ball, $\exp_0(\cdot)$ degenerates to *radial* `tanh` *saturation* (see equation (3)), which is equivalent to "*leaky integration + gain control*": integrating along the velocity direction while maintaining the state within a bounded manifold using a saturation nonlinearity/normalization. This echoes the observations of gain modulation and stable activity ranges in the cortex (Dayan & Abbott, 2005; Carandini & Heeger, 2012). Parallel transport can be understood as mapping the "local encoding direction" to a shared reference frame, consistent with the empirical observation that "population neural activity resides on a low-dimensional manifold and migrates within it" (e.g. Cunningham & Yu, 2014).

**Disclaimer.** The above analogies are purely *interpretable analogies* from the perspective of neural computation and are not intended to simulate the biological implementation of neural networks. This work does not pursue biological interpretability, and the analogies are provided solely for an intuitive understanding of the neural network computation process.

## B. Supplementary Notes on Storage Capacity

We analyze the storage capacity of HAMNs on a Hadamard manifold $(\mathcal{M}, g)$ of constant negative curvature $-c < 0$. To make the dependence on curvature explicit, we write

$$D_c(x, y) := \sqrt{c}\, d_\mathcal{M}(x, y),$$

and use the hyperbolic similarity

$$\langle x, y \rangle_M := -\cosh\big(D_c(x, y)\big).$$

When $c = 1$, this reduces to the notation used in the main text. The intrinsically regularized HAMN energy is

$$E(\xi) = -\frac{1}{\theta} \log \sum_{i=1}^{N} \exp\big(\theta \langle x_i, \xi \rangle_M\big) + \frac{1}{2} d_\mathcal{M}(\xi, p)^2,$$

where $p \in \mathcal{M}$ is a fixed reference point and $\theta > 0$ is the inverse temperature.

### B.1. Energy-well separation and recallability

We first make explicit how geodesic separation controls the softmax mass on the correct pattern.

**Proposition B.1** (Geodesic margin and softmax mass). *Let the stored patterns be $\{x_i\}_{i=1}^{N} \subset \mathcal{M}$, and define the minimum pairwise geodesic separation*

$$\delta := \min_{i \neq j} d_\mathcal{M}(x_i, x_j).$$

*Fix any radius $0 < \rho < \delta/2$ and set*

$$\varepsilon := \frac{\delta}{2} - \rho > 0.$$

*If the query $\xi$ lies in the geodesic ball $B_{\mathcal{M}}(x_k, \rho)$ around some memory $x_k$, then for the softmax weights*

$$w_i(\xi) := \frac{\exp\big(\theta\langle x_i, \xi\rangle_M\big)}{\sum_{\ell=1}^{N} \exp\big(\theta\langle x_\ell, \xi\rangle_M\big)}, \qquad \langle x_i, \xi\rangle_M = -\cosh\big(D_c(x_i, \xi)\big),$$

*we have*

$$w_k(\xi) \geq \frac{1}{1 + (N-1)\exp\big(-\theta\Delta_c(\delta, \varepsilon)\big)}, \tag{29}$$

*where*

$$\Delta_c(\delta, \varepsilon) := 2\sinh\Big(\frac{\sqrt{c}\,\delta}{2}\Big)\sinh(\sqrt{c}\,\varepsilon) > 0. \tag{30}$$

*In particular, for fixed $N, \theta, \rho$, the lower bound in (29) is strictly increasing in the geodesic margin $\delta$.*

*Proof.* Since $\xi \in B_{\mathcal{M}}(x_k, \rho)$, we have

$$d_{\mathcal{M}}(x_k, \xi) \leq \rho.$$

For any $j \neq k$, the triangle inequality gives

$$d_{\mathcal{M}}(x_j, \xi) \geq d_{\mathcal{M}}(x_j, x_k) - d_{\mathcal{M}}(x_k, \xi) \geq \delta - \rho = \frac{\delta}{2} + \varepsilon.$$

Equivalently,

$$D_c(x_k, \xi) \leq \sqrt{c}\,\rho = \frac{\sqrt{c}\,\delta}{2} - \sqrt{c}\,\varepsilon,$$

and

$$D_c(x_j, \xi) \geq \sqrt{c}(\delta - \rho) = \frac{\sqrt{c}\,\delta}{2} + \sqrt{c}\,\varepsilon.$$

Because $\cosh(\cdot)$ is increasing on $[0, \infty)$ and $\langle x, \xi\rangle_M = -\cosh(D_c(x, \xi))$, we obtain

$$s_k(\xi) := \langle x_k, \xi\rangle_M \geq -\cosh\Big(\frac{\sqrt{c}\,\delta}{2} - \sqrt{c}\,\varepsilon\Big),$$

and for every $j \neq k$,

$$s_j(\xi) := \langle x_j, \xi\rangle_M \leq -\cosh\Big(\frac{\sqrt{c}\,\delta}{2} + \sqrt{c}\,\varepsilon\Big).$$

Therefore

$$s_k(\xi) - s_j(\xi) \geq \cosh\Big(\frac{\sqrt{c}\,\delta}{2} + \sqrt{c}\,\varepsilon\Big) - \cosh\Big(\frac{\sqrt{c}\,\delta}{2} - \sqrt{c}\,\varepsilon\Big).$$

Using the identity

$$\cosh(a+b) - \cosh(a-b) = 2\sinh(a)\sinh(b),$$

with

$$a = \frac{\sqrt{c}\,\delta}{2}, \qquad b = \sqrt{c}\,\varepsilon,$$

we get

$$s_k(\xi) - s_j(\xi) \geq 2\sinh\Big(\frac{\sqrt{c}\,\delta}{2}\Big)\sinh(\sqrt{c}\,\varepsilon) = \Delta_c(\delta, \varepsilon).$$

Hence, for every $j \neq k$,

$$\exp\big(\theta(s_j(\xi) - s_k(\xi))\big) \leq \exp\big(-\theta\Delta_c(\delta, \varepsilon)\big).$$

Now rewrite the correct softmax mass as

$$w_k(\xi) = \frac{1}{1 + \sum_{j \neq k} \exp\big(\theta(s_j(\xi) - s_k(\xi))\big)}.$$

Applying the previous inequality to all $j \neq k$ yields

$$w_k(\xi) \geq \frac{1}{1 + (N-1)\exp\big(-\theta\Delta_c(\delta, \varepsilon)\big)},$$

which proves (29).

Finally, if $\rho$ is fixed, then

$$\Delta_c(\delta, \delta/2 - \rho) = \cosh\big(\sqrt{c}(\delta - \rho)\big) - \cosh(\sqrt{c}\,\rho),$$

whose derivative with respect to $\delta$ is

$$\sqrt{c}\,\sinh\big(\sqrt{c}(\delta - \rho)\big) > 0$$

whenever $\delta > \rho$. Thus the bound is strictly increasing in $\delta$. $\qquad\square$

*Remark* B.2. Proposition B.1 gives a deterministic separation-to-mass principle: once a query lies closer to $x_k$ than to all competing memories by a geodesic margin, the log-odds of the correct memory grow at least as $\theta\Delta_c(\delta, \varepsilon)$. Thus increasing hyperbolic separation between stored memories strengthens the correct softmax mass, rather than hurting recall.

## B.2. A one-step attraction basin

The previous proposition controls the softmax mass. To turn this into an invariant basin statement, one needs a mild inward-pointing condition on the actual retrieval map. This is the standard way to avoid the informal statement that $w_k(\xi) \geq 1/2$ alone necessarily makes the full Riemannian update point toward $x_k$; the latter can fail if non-target forces or the regularizer are large.

Let $\widehat{T}$ denote the undamped HAMN retrieval update, and define its damped version by

$$T_\eta(\xi) := \exp_\xi\big(\eta V(\xi)\big), \qquad V(\xi) := \log_\xi \widehat{T}(\xi), \qquad \eta \in (0, 1].$$

This covers both a damped CCCP update and a damped Riemannian-gradient update.

**Proposition B.3** (A deterministic one-step basin criterion). *Let $x_k \in \mathcal{M}$ and $0 < \rho < \delta/2$. Suppose that the vector field $V$ is $C^1$ on a neighborhood of the closed ball $\overline{B}_{\mathcal{M}}(x_k, \rho)$. If there exists $\mu > 0$ such that for all $\xi \in \partial B_{\mathcal{M}}(x_k, \rho)$,*

$$\big\langle \nabla_\xi d_{\mathcal{M}}(\xi, x_k),\, V(\xi) \big\rangle_\xi \leq -\mu, \tag{31}$$

*then there exists $\eta_0 \in (0, 1]$ such that for every $0 < \eta \leq \eta_0$,*

$$T_\eta\big(\overline{B}_{\mathcal{M}}(x_k, \rho)\big) \subset B_{\mathcal{M}}(x_k, \rho).$$

*In other words, $B_{\mathcal{M}}(x_k, \rho)$ is a one-step invariant attraction basin for the damped HAMN retrieval map.*

*Proof.* Define

$$r_k(\xi) := d_{\mathcal{M}}(\xi, x_k).$$

On $\mathcal{M} \setminus \{x_k\}$, $r_k$ is smooth in a Hadamard manifold because there is a unique minimizing geodesic from $x_k$ to every point and there is no cut locus. By assumption, on the boundary $r_k(\xi) = \rho$,

$$\mathrm{D}r_k(\xi)[V(\xi)] = \big\langle \nabla r_k(\xi), V(\xi) \big\rangle_\xi \leq -\mu.$$

By continuity and compactness of the boundary sphere, there exists $\alpha \in (0, \rho)$ such that the same derivative is still strictly negative on the outer annulus

$$A_\alpha := \{\xi \in \overline{B}_{\mathcal{M}}(x_k, \rho) : \rho - \alpha \leq r_k(\xi) \leq \rho\},$$

namely

$$\mathrm{D}r_k(\xi)[V(\xi)] \leq -\frac{\mu}{2}, \qquad \xi \in A_\alpha.$$

Since $V$ is $C^1$ and $A_\alpha$ is compact, both $\|V(\xi)\|$ and the second derivative of $r_k$ along the geodesic $t \mapsto \exp_\xi(tV(\xi))$ are uniformly bounded on $A_\alpha$. Hence there exists $C > 0$ such that, for all sufficiently small $\eta$ and all $\xi \in A_\alpha$,

$$r_k\big(\exp_\xi(\eta V(\xi))\big) \leq r_k(\xi) + \eta\,\mathrm{D}r_k(\xi)[V(\xi)] + C\eta^2 \leq r_k(\xi) - \frac{\mu}{2}\eta + C\eta^2.$$

Choosing $\eta \leq \mu/(4C)$ gives

$$r_k(T_\eta(\xi)) \leq r_k(\xi) - \frac{\mu}{4}\eta \leq \rho, \qquad \xi \in A_\alpha.$$

Thus points in the outer annulus are mapped strictly inward.

It remains to control the inner part

$$I_\alpha := \{\xi \in \overline{B}_\mathcal{M}(x_k, \rho) : r_k(\xi) \leq \rho - \alpha\}.$$

Since $V$ is bounded on $\overline{B}_\mathcal{M}(x_k, \rho)$, let

$$M := \sup_{\xi \in \overline{B}_\mathcal{M}(x_k, \rho)} \|V(\xi)\| < \infty.$$

The curve $t \mapsto \exp_\xi(tV(\xi))$ has speed $\|V(\xi)\|$, so

$$d_\mathcal{M}(T_\eta(\xi), \xi) \leq \eta M.$$

Choosing $\eta \leq \alpha/(2M)$ yields

$$r_k(T_\eta(\xi)) \leq r_k(\xi) + d_\mathcal{M}(T_\eta(\xi), \xi) \leq \rho - \alpha + \frac{\alpha}{2} < \rho.$$

Combining the two regions and taking

$$\eta_0 := \min\left\{1, \frac{\mu}{4C}, \frac{\alpha}{2M}\right\}$$

proves the claim. $\qquad\square$

**A checkable sufficient condition.** For the damped negative-gradient retrieval direction

$$V(\xi) := -\operatorname{grad} E(\xi),$$

let

$$u_i(\xi) := \frac{\log_\xi(x_i)}{d_\mathcal{M}(\xi, x_i)}$$

be the unit tangent vector at $\xi$ pointing toward $x_i$.

We use the identities

$$\nabla_\xi d_\mathcal{M}(\xi, x_i) = -u_i(\xi), \qquad \nabla_\xi \frac{1}{2} d_\mathcal{M}(\xi, p)^2 = -\log_\xi(p).$$

Since

$$\langle x_i, \xi \rangle_M = -\cosh(D_c(x_i, \xi)),$$

we have

$$\nabla_\xi \langle x_i, \xi \rangle_M = \sqrt{c}\, \sinh(D_c(x_i, \xi))\, u_i(\xi).$$

Therefore,

$$V(\xi) = -\operatorname{grad} E(\xi) = \sum_{i=1}^N w_i(\xi)\sqrt{c}\, \sinh(D_c(x_i, \xi)) u_i(\xi) + \log_\xi(p).$$

On the boundary $\partial B_\mathcal{M}(x_k, \rho)$, the inward direction is $u_k(\xi)$ and

$$\nabla_\xi d_\mathcal{M}(\xi, x_k) = -u_k(\xi).$$

Therefore the inward condition (31) is implied by

$$\langle V(\xi), u_k(\xi) \rangle_\xi \geq \mu \qquad \forall \xi \in \partial B_\mathcal{M}(x_k, \rho). \tag{32}$$

A conservative sufficient bound is

$$w_k(\xi)\sqrt{c}\, \sinh(\sqrt{c}\rho) - \sum_{j \neq k} w_j(\xi)\sqrt{c}\, \sinh(D_c(x_j, \xi)) - d_\mathcal{M}(\xi, p) \geq \mu. \tag{33}$$

Thus Proposition B.1 supplies the softmax domination term, while (33) is the explicit condition ensuring that the full retrieval vector field points inward after accounting for non-target memories and the reference-point regularizer.

## B.3. Hyperbolic volume and a sphere-packing capacity bound

Assume all patterns lie in a geodesic ball $B_{\mathcal{M}}(p, R - \delta/2)$.

**Proposition B.4** (Sphere-packing style capacity bound). *Assume $\{x_i\}_{i=1}^N \subset B_{\mathcal{M}}(p, R - \delta/2)$. If the recall basins $\{B_{\mathcal{M}}(x_i, \delta/2)\}_{i=1}^N$ are pairwise disjoint, then*

$$N \leq \frac{\mathrm{Vol}\big(B_{\mathcal{M}}(p, R)\big)}{\mathrm{Vol}\big(B_{\mathcal{M}}(\cdot, \delta/2)\big)}.$$

*In a $d$-dimensional hyperbolic space of curvature $-c$, the volume of a ball of radius $r$ satisfies*

$$\mathrm{Vol}_c(B(r)) = \omega_{d-1} \int_0^r \left( \frac{\sinh(\sqrt{c}\, t)}{\sqrt{c}} \right)^{d-1} dt.$$

*Moreover, for $r \gg 1/\sqrt{c}$,*

$$\mathrm{Vol}_c(B(r)) \asymp \kappa_{d,c} \exp\big((d-1)\sqrt{c}\, r\big),$$

*see, e.g., (Ratcliffe, 2006). Hence, for fixed $\delta$,*

$$N \leq C_{d,c,\delta} \, \exp\big((d-1)\sqrt{c}\, R\big),$$

*where $C_{d,c,\delta} > 0$ depends on $d, c, \delta$ but not on $R$. When both $R$ and $\delta/2$ are in the large-radius regime, the volume ratio further scales as*

$$N \lesssim \exp\Big((d-1)\sqrt{c}\,(R - \delta/2)\Big). \tag{34}$$

*Thus the number of non-overlapping recall wells grows exponentially in the effective hyperbolic radius, with a rate controlled jointly by the dimension $d$ and the curvature $c$.*

*Proof.* Since $x_i \in B_{\mathcal{M}}(p, R - \delta/2)$, every point $y \in B_{\mathcal{M}}(x_i, \delta/2)$ satisfies, by the triangle inequality,

$$d_{\mathcal{M}}(p, y) \leq d_{\mathcal{M}}(p, x_i) + d_{\mathcal{M}}(x_i, y) < R - \frac{\delta}{2} + \frac{\delta}{2} = R.$$

Hence

$$B_{\mathcal{M}}(x_i, \delta/2) \subset B_{\mathcal{M}}(p, R) \qquad \forall i.$$

If these balls are pairwise disjoint, then

$$\sum_{i=1}^N \mathrm{Vol}\big(B_{\mathcal{M}}(x_i, \delta/2)\big) \leq \mathrm{Vol}\big(B_{\mathcal{M}}(p, R)\big).$$

Hyperbolic space is homogeneous, so

$$\mathrm{Vol}\big(B_{\mathcal{M}}(x_i, \delta/2)\big) = \mathrm{Vol}\big(B_{\mathcal{M}}(\cdot, \delta/2)\big)$$

for all $i$. Dividing by this common basin volume gives the stated packing bound.

The volume formula follows from geodesic polar coordinates in constant negative curvature. The large-radius asymptotic follows from

$$\sinh(\sqrt{c}\, t) \asymp \frac{1}{2} e^{\sqrt{c}\, t} \qquad (t \gg 1/\sqrt{c}).$$

For fixed $\delta$, the denominator is a constant independent of $R$, yielding the bound with $C_{d,c,\delta}$. If both radii are in the large-radius regime, applying the same exponential asymptotic to the denominator gives (34). $\square$

### B.4. Random-memory capacity on a hyperbolic shell

The previous subsection gives a deterministic result. We now present a random-memory capacity theorem: random shell memories induce angular separation with high probability; angular separation further implies geodesic separation with high probability; and, whenever the deterministic softmax and basin conditions above are satisfied, geodesic separation further yields retrieval-stable basins.

Let $(\mathbb{H}_c^d, g)$ be the $d$-dimensional hyperbolic space of curvature $-c < 0$, and fix $p \in \mathbb{H}_c^d$. Choose

$$R_d := K\sqrt{d-1}, \qquad K > 0.$$

Let

$$u_1, \ldots, u_N \overset{\text{i.i.d.}}{\sim} \mathrm{Unif}(S^{d-1})$$

be independent random directions in $T_p\mathbb{H}_c^d$, and define

$$x_i := \exp_p(R_d u_i), \qquad i = 1, \ldots, N.$$

Thus all memories lie on the geodesic sphere of radius $R_d$ centered at $p$.

**Lemma B.5** (Tail bound for random directions on the sphere). *Let $d \geq 3$ and let $u, v \sim \mathrm{Unif}(S^{d-1})$ be independent. Set*

$$Z := \langle u, v \rangle.$$

*Then $Z$ has density*

$$f_d(z) = C_d(1 - z^2)^{(d-3)/2}, \qquad z \in [-1, 1],$$

*where*

$$C_d = \frac{\Gamma(d/2)}{\sqrt{\pi}\,\Gamma((d-1)/2)}.$$

*Consequently, for every $t \in (0, 1)$,*

$$\mathbb{P}(Z \geq t) \leq A\sqrt{d}\,\exp\left(-\frac{d-3}{2}t^2\right),$$

*where $A > 0$ is an absolute constant.*

*Proof.* By rotational invariance, we may fix $u = e_1$. Then $Z = \langle e_1, v \rangle = v_1$, the first coordinate of a uniform random point on $S^{d-1}$. Its density is

$$f_d(z) = C_d(1 - z^2)^{(d-3)/2}, \qquad z \in [-1, 1].$$

Therefore,

$$\mathbb{P}(Z \geq t) = C_d \int_t^1 (1 - z^2)^{(d-3)/2}\, dz.$$

Since $d \geq 3$, the function $(1 - z^2)^{(d-3)/2}$ is non-increasing on $[0, 1]$. Hence

$$\int_t^1 (1 - z^2)^{(d-3)/2}\, dz \leq (1 - t)(1 - t^2)^{(d-3)/2}.$$

Using $\log(1 - x) \leq -x$ for $x \in (0, 1)$, we obtain

$$(1 - t^2)^{(d-3)/2} \leq \exp\left(-\frac{d-3}{2}t^2\right).$$

Thus

$$\mathbb{P}(Z \geq t) \leq C_d \exp\left(-\frac{d-3}{2}t^2\right).$$

Finally, the standard Gamma-ratio estimate gives

$$C_d = \frac{\Gamma(d/2)}{\sqrt{\pi}\,\Gamma((d-1)/2)} \leq A\sqrt{d}$$

for a universal constant $A > 0$. This proves the claim. $\square$

**Lemma B.6** (High-probability pairwise angular separation). *Let $d \geq 3$. For fixed $t \in (0,1)$,*

$$\mathbb{P}\left(\max_{1 \leq i < j \leq N} \langle u_i, u_j \rangle > t\right) \leq \binom{N}{2} A\sqrt{d}\exp\left(-\frac{d-3}{2}t^2\right).$$

*In particular, if*

$$\binom{N}{2} A\sqrt{d}\exp\left(-\frac{d-3}{2}t^2\right) \leq \zeta,$$

*then with probability at least $1 - \zeta$,*

$$\langle u_i, u_j \rangle \leq t \qquad \forall i \neq j.$$

*Proof.* Apply Lemma B.5 to each pair $(u_i, u_j)$ and take a union bound over the $\binom{N}{2}$ unordered pairs. $\qquad\square$

**Lemma B.7** (Angular separation implies hyperbolic separation). *Let*

$$x_i = \exp_p(R_d u_i), \qquad x_j = \exp_p(R_d u_j),$$

*with $\|u_i\| = \|u_j\| = 1$. If*

$$\langle u_i, u_j \rangle \leq t,$$

*then*

$$d_{\mathbb{H}_c^d}(x_i, x_j) \geq \delta_{d,c}(t),$$

*where*

$$\delta_{d,c}(t) := \frac{1}{\sqrt{c}} \operatorname{arcosh}\left(\cosh^2(\sqrt{c}\,R_d) - \sinh^2(\sqrt{c}\,R_d)t\right). \tag{35}$$

*Proof.* For two points at the same radius $R_d$ from $p$, the hyperbolic law of cosines in curvature $-c$ gives

$$\cosh\left(\sqrt{c}\,d_{\mathbb{H}_c^d}(x_i, x_j)\right) = \cosh^2(\sqrt{c}\,R_d) - \sinh^2(\sqrt{c}\,R_d)\cos\varphi_{ij},$$

where $\varphi_{ij}$ is the angle between the two radial geodesics at $p$. Since

$$\cos\varphi_{ij} = \langle u_i, u_j \rangle,$$

the assumption $\langle u_i, u_j \rangle \leq t$ implies

$$\cosh\left(\sqrt{c}\,d_{\mathbb{H}_c^d}(x_i, x_j)\right) \geq \cosh^2(\sqrt{c}\,R_d) - \sinh^2(\sqrt{c}\,R_d)t.$$

Because $\operatorname{arcosh}$ is increasing on $[1, \infty)$, the desired lower bound follows. $\qquad\square$

**Corollary B.8** (High-probability geodesic separation). *Let $d \geq 3$. If*

$$\binom{N}{2} A\sqrt{d}\exp\left(-\frac{d-3}{2}t^2\right) \leq \zeta,$$

*then with probability at least $1 - \zeta$,*

$$d_{\mathbb{H}_c^d}(x_i, x_j) \geq \delta_{d,c}(t) \qquad \forall i \neq j.$$

**Theorem B.9** (Random retrieval-stable memory capacity). *Let $d \geq 3$. Fix $t \in (0,1)$ and define*

$$\delta_d := \delta_{d,c}(t) = \frac{1}{\sqrt{c}} \operatorname{arcosh}\left(\cosh^2(\sqrt{c}\,R_d) - \sinh^2(\sqrt{c}\,R_d)t\right).$$

*Let $0 < \rho_d < \delta_d/2$ and set*

$$\varepsilon_d := \frac{\delta_d}{2} - \rho_d.$$

*Assume that the softmax domination condition*

$$N \leq 1 + \exp\left(\theta\Delta_c(\delta_d, \varepsilon_d)\right) \tag{36}$$

*holds, where*

$$\Delta_c(\delta, \varepsilon) = 2 \sinh\left(\frac{\sqrt{c}\,\delta}{2}\right) \sinh(\sqrt{c}\,\varepsilon).$$

*Assume additionally that the damped HAMN retrieval vector field satisfies the inward condition in Proposition B.3 on each boundary $\partial B_{\mathbb{H}_c^d}(x_i, \rho_d)$. If*

$$\binom{N}{2} A \sqrt{d} \exp\left(-\frac{d-3}{2} t^2\right) \leq \zeta,$$

*then with probability at least $1 - \zeta$, all random memories $\{x_i\}_{i=1}^N$ admit pairwise disjoint one-step invariant retrieval basins*

$$B_{\mathbb{H}_c^d}(x_i, \rho_d), \qquad i = 1, \ldots, N.$$

*Proof.* By Corollary B.8, with probability at least $1 - \zeta$,

$$d_{\mathbb{H}_c^d}(x_i, x_j) \geq \delta_d \qquad \forall i \neq j.$$

Since $\rho_d < \delta_d/2$, the geodesic balls

$$B_{\mathbb{H}_c^d}(x_i, \rho_d)$$

are pairwise disjoint.

On the same high-probability event, Proposition B.1 applies to every memory $x_i$ with $\delta = \delta_d$ and $\varepsilon = \varepsilon_d$. Therefore, for every $q \in B_{\mathbb{H}_c^d}(x_i, \rho_d)$,

$$w_i(q) \geq \frac{1}{1 + (N-1) \exp\left(-\theta \Delta_c(\delta_d, \varepsilon_d)\right)}.$$

The softmax domination condition (36) implies

$$w_i(q) \geq \frac{1}{2}.$$

Together with the assumed inward condition of Proposition B.3, this yields one-step invariance of each ball $B_{\mathbb{H}_c^d}(x_i, \rho_d)$ under a sufficiently small damping stepsize. Hence all memories are retrieval-stable with pairwise disjoint basins. $\square$

**Theorem B.10** (Exponential random capacity in the ambient dimension)**.** *Fix $K > 0$, $c > 0$, $\theta > 0$, and $t \in (0, 1)$. Let*

$$R_d = K\sqrt{d-1}.$$

*For every*

$$0 < \gamma < \frac{t^2}{4},$$

*define*

$$N_d := \left\lfloor e^{\gamma(d-1)} \right\rfloor.$$

*Then, with probability tending to $1$ as $d \to \infty$, the random shell memories*

$$x_i = \exp_p(R_d u_i), \qquad u_i \overset{\text{i.i.d.}}{\sim} \text{Unif}(S^{d-1}),$$

*are pairwise separated by at least $\delta_{d,c}(t)$.*

*Consequently, whenever for a chosen radius $\rho_d < \delta_{d,c}(t)/2$ both the softmax domination condition (36) with $N = N_d$ and the deterministic inward-basin condition in Proposition B.3 are satisfied, HAMNs can store*

$$N_d = e^{\Omega(d)}$$

*random memories with pairwise disjoint one-step invariant retrieval basins.*

*Proof.* For all sufficiently large $d$, we have $d \geq 3$. By Lemma B.6,

$$\mathbb{P}(\text{failure}) \leq \binom{N_d}{2} A\sqrt{d}\exp\left(-\frac{d-3}{2}t^2\right).$$

Since $N_d \leq e^{\gamma(d-1)}$,

$$\binom{N_d}{2} \leq \frac{1}{2}e^{2\gamma(d-1)}.$$

Therefore,

$$\mathbb{P}(\text{failure}) \leq \frac{A}{2}\sqrt{d}\exp\left(2\gamma(d-1) - \frac{d-3}{2}t^2\right).$$

Because $\gamma < t^2/4$, the exponent is negative for all sufficiently large $d$, and hence the failure probability tends to $0$. Thus, with probability tending to $1$,

$$d_{\mathbb{H}_c^d}(x_i, x_j) \geq \delta_{d,c}(t) \qquad \forall i \neq j.$$

It remains to understand the scale of this separation. From (35),

$$\cosh(\sqrt{c}\,\delta_{d,c}(t)) = \cosh^2(\sqrt{c}\,R_d) - \sinh^2(\sqrt{c}\,R_d)t = 1 + (1-t)\sinh^2(\sqrt{c}\,R_d).$$

Since $R_d = K\sqrt{d-1} \to \infty$,

$$\sinh^2(\sqrt{c}\,R_d) \sim \frac{1}{4}e^{2\sqrt{c}\,R_d},$$

and therefore

$$\delta_{d,c}(t) = 2R_d + O(1) = 2K\sqrt{d-1} + O(1).$$

Thus the random memories are separated at a hyperbolic scale determined by the shell radius.

If, in addition, for some $\rho_d < \delta_{d,c}(t)/2$, both the softmax domination condition (36) with $N = N_d$ and the deterministic inward-basin condition of Proposition B.3 hold, then Theorem B.9 turns this high-probability separation into pairwise disjoint one-step invariant retrieval basins. Therefore the number of random retrieval-stable memories scales as $e^{\Omega(d)}$. $\qquad\square$

**Corollary B.11** (Explicit lower bound on random memory capacity)**.** *Under the setting of Theorem B.10, fix any $t \in (0, 1)$. Suppose that, for the chosen radius $\rho_d < \delta_{d,c}(t)/2$, both the softmax domination condition (36) with $N = N_d$ and the deterministic inward-basin condition hold for all sufficiently large $d$. Then, for any fixed constant*

$$0 < \kappa < \frac{t^2}{4},$$

*take*

$$N_d = \left\lfloor \exp\bigl(\kappa(d-1)\bigr) \right\rfloor.$$

*As $d \to \infty$, these $N_d$ random shell memories admit pairwise disjoint one-step invariant retrieval basins with probability tending to $1$.*

*Equivalently, if the high-probability random-memory capacity of HAMNs under this shell model is denoted by $\mathrm{Cap}_{\mathrm{HAMN}}^{\mathrm{rand}}(d)$, then for any fixed $\kappa < t^2/4$ and all sufficiently large $d$,*

$$\mathrm{Cap}_{\mathrm{HAMN}}^{\mathrm{rand}}(d) \geq \left\lfloor \exp\bigl(\kappa(d-1)\bigr) \right\rfloor.$$

*Therefore, the random-memory capacity is at least exponential in $d$:*

$$\mathrm{Cap}_{\mathrm{HAMN}}^{\mathrm{rand}}(d) \geq \exp(\Omega(d)).$$

*Proof.* Let

$$N_d = \left\lfloor \exp\bigl(\kappa(d-1)\bigr) \right\rfloor, \qquad 0 < \kappa < \frac{t^2}{4}.$$

For all sufficiently large $d$, we have $d \geq 3$. By Lemma B.6, the failure probability satisfies

$$\mathbb{P}(\text{failure}) \leq \binom{N_d}{2} A \sqrt{d} \exp\left(-\frac{d-3}{2} t^2\right).$$

Since

$$N_d \leq \exp\big(\kappa(d-1)\big),$$

we have

$$\binom{N_d}{2} \leq \frac{1}{2} \exp\big(2\kappa(d-1)\big).$$

Therefore,

$$\mathbb{P}(\text{failure}) \leq \frac{A}{2} \sqrt{d} \exp\left(2\kappa(d-1) - \frac{d-3}{2} t^2\right).$$

Because

$$\kappa < \frac{t^2}{4},$$

the exponent tends to $-\infty$ for all sufficiently large $d$, and hence the failure probability tends to $0$. Thus, with probability tending to $1$, all random memories are pairwise geodesically separated by at least $\delta_{d,c}(t)$.

Combining this high-probability separation with the softmax domination condition (36) and the deterministic inward-basin condition, Theorem B.9 implies that these random memories admit pairwise disjoint one-step invariant retrieval basins. This proves the stated capacity lower bound. $\qquad\square$

*Remark* B.12 (When the softmax and inward conditions are automatically satisfied). Theorem B.10 separates the probabilistic part from the dynamical part. The probabilistic part proves high-probability geodesic separation. The dynamical part is supplied by the softmax domination condition (36) and the inward-basin criterion in Proposition B.3.

For the damped negative-gradient HAMN update, one can verify these conditions asymptotically under the random shell model by choosing

$$\rho_d = \beta \delta_{d,c}(t), \qquad 0 < \beta < \frac{1}{2}.$$

Then

$$\varepsilon_d = \frac{\delta_{d,c}(t)}{2} - \rho_d = \left(\frac{1}{2} - \beta\right) \delta_{d,c}(t) = \Theta(\sqrt{d}),$$

and hence

$$\Delta_c(\delta_d, \varepsilon_d) = 2 \sinh\left(\frac{\sqrt{c}\,\delta_d}{2}\right) \sinh(\sqrt{c}\,\varepsilon_d) = \exp(\Theta(\sqrt{d})).$$

For $N_d = e^{\gamma(d-1)}$, this gives

$$N_d \leq 1 + \exp\big(\theta \Delta_c(\delta_d, \varepsilon_d)\big)$$

for all sufficiently large $d$, since the right-hand side grows as $\exp(\exp(\Theta(\sqrt{d})))$, which dominates $e^{\gamma(d-1)}$.

Moreover, the incorrect softmax mass satisfies the conservative bound

$$1 - w_k(q) \lesssim N_d \exp\big(-\theta \Delta_c(\delta_d, \varepsilon_d)\big) \to 0.$$

Meanwhile, the correct attractive force on the boundary scales as

$$\sqrt{c} \sinh(\sqrt{c}\,\rho_d) = \exp(\Theta(\sqrt{d})),$$

which dominates the $O(\sqrt{d})$ radial regularizer and the vanishing weighted contribution of non-target memories. Therefore, under this random shell model, the softmax domination and inward-basin conditions are satisfied for all sufficiently large $d$.

### B.5. Comparison with Euclidean MHNs

Modern Euclidean Hopfield networks can also achieve exponential capacity in the ambient dimension for random patterns; for example, under the log-sum-exp energy, they can reach $N = 2^{\Omega(d)}$ (Ramsauer et al., 2021). Our result is complementary to this. Our analysis highlights an additional geometry-controlled advantage in negatively curved spaces: since hyperbolic volume grows exponentially with radius, the number of non-overlapping attraction basins also grows exponentially with the effective hyperbolic radius. Meanwhile, under the random shell model, HAMNs also admit an $e^{\Omega(d)}$ high-probability lower bound on random-memory capacity.

**Main takeaway.** Error-free recall is jointly guaranteed by the geodesic margin, softmax domination, and the inward retrieval condition. The deterministic sphere-packing result shows that the number of recall wells grows exponentially with the effective hyperbolic radius. The random shell theorem further shows that HAMNs can store at least $e^{\Omega(d)}$ random retrieval-stable memories with high probability. These results explain why HAMNs are especially suitable for deeply hierarchical data: in such data, semantic depth can naturally correspond to outward radial organization in hyperbolic space.

## C. Hyperbolic Geometry Background and Common Model Formulas

### C.1. Isometry invariance

If $\phi : (\mathcal{M}, g) \to (\mathcal{M}', g')$ is an isometry, then

$$d_{\mathcal{M}}(x, y) = d_{\mathcal{M}'}(\phi(x), \phi(y)).$$

Therefore,

$$\langle x, y \rangle_M = -\cosh\big(d_{\mathcal{M}}(x, y)\big) = -\cosh\big(d_{\mathcal{M}'}(\phi(x), \phi(y))\big) = \langle \phi(x), \phi(y) \rangle_{M'}.$$

Hence, any energy and update constructed from $d_{\mathcal{M}}$ and $-\cosh d_{\mathcal{M}}$ are model-equivalent across hyperbolic realizations such as the Poincaré ball, Lorentz hyperboloid, Klein model, and upper-half plane. This justifies writing the main derivations in a model-agnostic form, while instantiating the implementation in the Poincaré ball.

### C.2. Hierarchy intuition: a toy example

**Hierarchical data.** We call a dataset $\mathcal{D} = \{(x_i, y_i)\}_{i=1}^{N}$ *hierarchical* if its label space $\mathcal{Y}$ is equipped with a directed acyclic graph (DAG) $(\mathcal{Y}, \mathcal{E})$ or a rooted tree, where each node $y \in \mathcal{Y}$ has a depth $\mathrm{depth}(y)$ and ancestors $\mathrm{Anc}(y)$. Evaluation is required to be *hierarchy-aware*, in the sense that it depends on the graph distance between predicted and true labels, e.g. hierarchical accuracy at multiple levels (coarse/fine), shortest-path distance in the label DAG, or correlation between tree distance and representation distance such as cophenetic correlation (Sokal & Rohlf, 1962).

*For simplicity in this toy example we take curvature $-c = -1$, i.e., $c = 1$, on the Poincaré ball.*

**Euclidean space "flattens" hierarchies.** Embed a tree of depth $L$ into the plane: all nodes at level $\ell$ lie on the same-radius circle. As $L$ grows, the outermost nodes crowd the same ring and leaf–leaf distances are governed almost only by the angular gap and become very similar, so leaves from different major branches appear "about equally far" and hierarchical information is weakened.

**Hyperbolic space "pulls apart" hierarchies.** Keep angles uniform, but encode depth by hyperbolic radius:

$$\rho_\ell = \tanh(\alpha\ell/2), \qquad \alpha > 0.$$

Since $d(0, x) = 2\,\mathrm{artanh}\,\|x\|$, any level-$\ell$ node satisfies

$$d_{\mathbb{D}}(0, x_\ell) = \alpha\,\ell.$$

That is, each additional level increases the radial hyperbolic distance by approximately a fixed amount, so different levels separate naturally. Moreover, because the metric is magnified near the boundary, two points on the same level but from different major branches acquire a much larger hyperbolic distance even for a tiny angular gap, whereas points within the same subtree are closer.

**Rule of thumb.** If two leaves have lowest common ancestor depth $a$, then the dominant term of their distance is

$$d_{\mathbb{D}}(x_i, x_j) \approx 2\alpha(L - a) + \text{lower-order terms},$$

which increases with tree distance and is monotone in the LCA depth. Hence hyperbolic space simultaneously preserves two signals—depth through the radial direction and branching relation through the angular direction—and avoids the hierarchical flattening of Euclidean embeddings.

### C.3. Key Formulas for Common Hyperbolic Models (Constant Curvature $-c < 0$)

**Notation & convention.** Let the curvature be $-c$ with $c > 0$ and write $\text{arcosh}(\cdot)$ for the inverse hyperbolic cosine. All standard hyperbolic models are *isometric*; hence any model-agnostic derivation in the paper becomes an implementation by choosing

$$d_{\mathcal{M}}, \qquad \exp_p^c, \qquad \log_p^c$$

from a specific model. We use the hyperbolic similarity

$$\langle x, y \rangle_M := -\cosh\big(d_{\mathcal{M}}(x, y)\big),$$

with the universal chain rule

$$\nabla_x \langle x, y \rangle_M = -\sinh\big(d_{\mathcal{M}}(x, y)\big) \nabla_x d_{\mathcal{M}}(x, y),$$

equivalently

$$\text{grad}_x \langle x, y \rangle_M = \sinh(d_{\mathcal{M}}(x, y)) \frac{\log_x(y)}{\|\log_x(y)\|_g}$$

see App. §A.5.

**Model-agnostic CCCP closed form used in this work.** Let

$$a(\xi) := \text{grad}\, E_{\text{cave}}(\xi) = -\sum_{i=1}^N w_i(\xi)\, \text{grad}_\xi \langle x_i, \xi \rangle_M, \qquad w_i(\xi) = \frac{e^{\theta \langle x_i, \xi \rangle_M}}{\sum_j e^{\theta \langle x_j, \xi \rangle_M}}.$$

Our CCCP step reads

$$v := -\text{PT}_{\xi \to p}\big(a(\xi)\big), \qquad \xi^+ = \exp_p^c(\eta v), \quad \eta \in (0, 1]. \tag{37}$$

Thus each model only needs $\exp_p^c, \log_p^c, d_{\mathcal{M}}$ and, if desired, a convenient form of parallel transport.

### C.3.1. POINCARÉ BALL $\mathbb{D}_c^d = \{x \in \mathbb{R}^d : \|x\| < 1/\sqrt{c}\}$

**Exponential/logarithmic maps at the origin.**

$$\exp_0^c(v) = \tanh(\sqrt{c}\,\|v\|) \frac{v}{\sqrt{c}\,\|v\|}, \qquad \log_0^c(x) = \text{artanh}(\sqrt{c}\,\|x\|) \frac{x}{\sqrt{c}\,\|x\|}.$$

**Base-point maps.** With Möbius addition $\oplus_c$ and conformal factor

$$\lambda_x^c = \frac{2}{1 - c\|x\|^2},$$

we have

$$\exp_p^c(v) = p \oplus_c \left( \tanh\left( \frac{\sqrt{c}\,\lambda_p^c \|v\|}{2} \right) \frac{v}{\sqrt{c}\,\|v\|} \right),$$

and

$$\log_p^c(x) = \frac{2}{\sqrt{c}\,\lambda_p^c} \text{artanh}\big(\sqrt{c}\,\|(-p) \oplus_c x\|\big) \frac{(-p) \oplus_c x}{\|(-p) \oplus_c x\|}.$$

**Geodesic distance.**

$$d_{\mathbb{D}_c}(x, y) = \frac{1}{\sqrt{c}} \text{arcosh}\left( 1 + \frac{2c\|x - y\|^2}{(1 - c\|x\|^2)(1 - c\|y\|^2)} \right).$$

**Parallel transport & implementation note.** The ball is *conformal*, $g(x) = \lambda(x)^2 I$ with

$$\lambda(x) = \frac{2}{1 - c\|x\|^2}.$$

If autodiff provides the Euclidean gradient $\nabla_x f$, the Riemannian gradient is

$$\operatorname{grad}_x f = \lambda(x)^{-2} \nabla_x f.$$

Choosing $p = 0$, the transport along the unique geodesic to the origin can be implemented as a scalar rescaling

$$\mathrm{PT}_{\xi \to 0}(u) = \frac{\lambda(\xi)}{\lambda(0)} u.$$

This scaling preserves the Riemannian norm because the ball is conformal: letting $\|\cdot\|_E$ denote the Euclidean norm,

$$\|u'\|_E = \frac{\lambda(\xi)}{\lambda(0)} \|u\|_E$$

ensures

$$\lambda(0)^2 \|u'\|_E^2 = \lambda(\xi)^2 \|u\|_E^2.$$

Then the explicit step with $p = 0$ amounts to

$$v_0 := -\frac{\lambda(\xi)}{\lambda(0)} a(\xi)$$

followed by $\exp_0^c$, where $a(\xi) = \operatorname{grad} E_{\text{cave}}(\xi)$. Equivalently, using the Euclidean gradient $g_E = \nabla_\xi E_{\text{cave}}(\xi)$,

$$v_0 = -\frac{\lambda(\xi)}{\lambda(0)} \lambda(\xi)^{-2} g_E.$$

**Möbius addition / scalar multiplication.** Möbius addition provides the gyrovector-space analogue of Euclidean vector addition in the Poincaré ball. It is commonly used to express hyperbolic translations, bias addition, scalar multiplication, and neural operations in a closed form while keeping all points inside the ball. This formalism goes back to the theory of Möbius gyrovector spaces (Ungar, 2005; 2008) and has been widely adopted in hyperbolic representation learning and hyperbolic neural networks (Nickel & Kiela, 2017; Ganea et al., 2018). For $u, v \in \mathbb{D}_c^d$, the Möbius addition and scalar multiplication are given by

$$u \oplus_c v = \frac{\left(1 + 2c\langle u, v\rangle + c\|v\|^2\right) u + \left(1 - c\|u\|^2\right) v}{1 + 2c\langle u, v\rangle + c^2 \|u\|^2 \|v\|^2},$$

$$r \otimes_c u = \frac{1}{\sqrt{c}} \tanh\left(r \operatorname{artanh}(\sqrt{c}\,\|u\|)\right) \frac{u}{\|u\|}.$$

Here $u \oplus_c v$ can be viewed as the hyperbolic counterpart of translating $u$ by $v$, while $r \otimes_c u$ moves $u$ along the geodesic ray from the origin through $u$ by a factor $r$ in hyperbolic distance. In particular, when $c \to 0$, these operations recover the Euclidean limits $u \oplus_c v \to u + v$ and $r \otimes_c u \to ru$, which explains why they are often used as drop-in hyperbolic analogues of Euclidean affine operations. In this paper, they are mainly used to instantiate $\exp_p^c$, $\log_p^c$ and other Poincaré-ball operations needed by the model-agnostic HAMN retrieval update.

C.3.2. UPPER HALF-PLANE $\mathbb{H}_c = \{(u, y) \in \mathbb{R}^{d-1} \times \mathbb{R}_{>0}\}$

**Geodesic distance.**

$$d_{\mathbb{H}_c}\big((u_1, y_1), (u_2, y_2)\big) = \frac{1}{\sqrt{c}} \operatorname{arcosh}\left(1 + \frac{c\big(\|u_1 - u_2\|^2 + (y_1 - y_2)^2\big)}{2 y_1 y_2}\right).$$

**Implementation note.** Rather than using explicit closed forms for $\exp / \log$ and parallel transport in the half-plane, we implement them via an isometry to the Poincaré ball or Lorentz model: map points to the chosen model, perform $\exp / \log$, $d$, and PT there, and map back. This avoids subtle formula errors and improves reproducibility.

### C.3.3. KLEIN MODEL $\mathbb{K}_c^d = \{x \in \mathbb{R}^d : \|x\| < 1/\sqrt{c}\}$

**Exponential/logarithmic maps at the origin.** They coincide with the ball at 0:

$$\exp_0^c, \qquad \log_0^c$$

as above.

**Geodesic distance.**

$$d_{\mathbb{K}_c}(p, q) = \frac{1}{\sqrt{c}} \operatorname{arcosh}\left( \frac{1 - c\, p^\top q}{\sqrt{(1 - c\|p\|^2)(1 - c\|q\|^2)}} \right).$$

**Parallel transport: recommendation.** Since the Klein model is not conformal, closed-form PT is more involved. In practice, use an isometry to Lorentz or Poincaré, perform PT and the exponential step there, and map back to Klein.

### C.3.4. HEMISPHERE $J_c = \{u \in \mathbb{S}^n : u_{n+1} > 0\}$

**Implementation note.** We recommend using the standard isometry to the Lorentz model to compute $\exp / \log$, $d$, and PT, and then map back to the hemisphere. We omit redundant explicit formulas here to avoid confusion, since our experiments instantiate Poincaré/Lorentz directly.

### C.3.5. LORENTZ HYPERBOLOID $L_c = \{X \in \mathbb{R}^{n+1} : X_0^2 - \sum_{i=1}^n X_i^2 = \frac{1}{c}, \; X_0 > 0\}$

**Minkowski bilinear form.**

$$(X, Y)_M = -X_0 Y_0 + \sum_{i=1}^n X_i Y_i.$$

**Distance and similarity.**

$$d_{L_c}(X, Y) = \frac{1}{\sqrt{c}} \operatorname{arcosh}\big( -c\, (X, Y)_M \big), \qquad \langle X, Y \rangle_M = -\cosh\big( d_{L_c}(X, Y) \big).$$

**Exponential/logarithm at $e_0 = (\frac{1}{\sqrt{c}}, 0, \ldots, 0)$.**

$$\exp_{e_0}^c(W) = \left( \frac{1}{\sqrt{c}} \cosh(\sqrt{c}\, \|W\|), \frac{1}{\sqrt{c}} \sinh(\sqrt{c}\, \|W\|) \frac{W}{\|W\|} \right),$$

$$\log_{e_0}^c(X) = \frac{1}{\sqrt{c}} \operatorname{arcosh}(\sqrt{c}\, X_0) \frac{X_{1:n}}{\|X_{1:n}\|}.$$

**Parallel transport.** It can be implemented by a Lorentz boost: let $B_{X \to e_0} \in SO^+(1, n)$ map $X$ to $e_0$, then

$$\mathrm{PT}_{X \to e_0}(V) = B_{X \to e_0} V.$$

**Where to plug in the main text.**

- **Memory encoding (Sec. 3, Eq. (3)).** Pick $\exp_p^c, \log_p^c$ from any model. If tangent clipping is used, clip in $T_p \mathcal{M}$ and map back via $\exp_p^c$.

- **Energy (Eq. (5)).** Substitute the chosen model's $d_\mathcal{M}$, or equivalently $\langle \cdot, \cdot \rangle_M$; nothing else changes.

- **Retrieval/optimization (CCCP step Eq. (37)).** Use the model-agnostic gradient of the concave term via $\log_\xi(\cdot)$ and $\sinh d$, then apply the model's PT and $\exp_p^c$ to update.

- **Energy bounds (Appendix A.3).** Plug the model's $d_\mathcal{M}$ into the same bounding argument.

# D. Additional Details of Hyperbolic Hopfield Layers

The main text introduces the computational pipeline of HAMN modules and summarizes the Poincaré-ball retrieval procedure in Alg. 1. This appendix provides additional implementation details for the three modules used in our experiments:

**Hyperbolic Hopfield**, **Hyperbolic Hopfield Pooling**, **Hyperbolic Hopfield Layer**.

All three modules share the same curvature parameter $c$, which can be fixed or made learnable, and follow the same CCCP closed-form step derived in the main text. In our implementation, the reference point is set to $p = 0$, i.e., the center of the Poincaré ball $\mathbb{D}_c^d$.

## D.1. Hyperbolic Hopfield

**HypHopfield** takes queries $R \in \mathbb{R}^{B \times d}$ and memories $Y \in \mathbb{R}^{N \times d}$ as input, and outputs retrieved states $Z \in \mathbb{R}^{B \times d}$. The main text summarizes the retrieval procedure in Alg. 1; here we spell out the implementation details on the Poincaré ball.

**Similarity and soft weights.** For each query $R_b$ and memory $Y_i$, we compute the hyperbolic similarity and the corresponding soft weight as

$$S_{b,i} = \langle Y_i, R_b \rangle_M = -\cosh\big(d_{\mathbb{D}_c}(Y_i, R_b)\big),$$

$$W_{b,i} = \frac{e^{\theta S_{b,i}}}{\sum_{j=1}^{N} e^{\theta S_{b,j}}}, \qquad W \in \mathbb{R}^{B \times N}.$$

**Intrinsic gradient and parallel transport.** Let the concave term be

$$E_{\text{cave}}(\xi) = -\frac{1}{\theta} \log \sum_i e^{\theta \langle x_i, \xi \rangle_M}.$$

For each batch element $R_b$, the Riemannian gradient of the concave term is

$$a_b = \text{grad}\, E_{\text{cave}}(R_b) = -\sum_{i=1}^{N} W_{b,i} \, \text{grad}_\xi \langle Y_i, \xi \rangle_M \Big|_{\xi = R_b}.$$

Using the gradient expression in Appendix A.5, we have

$$\text{grad}_\xi \langle Y_i, \xi \rangle_M = \sinh\big(d_{\mathbb{D}_c}(Y_i, \xi)\big) \frac{\log_\xi(Y_i)}{\|\log_\xi(Y_i)\|_g}.$$

We then transport this direction to the reference point $p = 0$:

$$v_b := -\text{PT}_{R_b \to 0}(a_b).$$

In the conformal Poincaré ball, the metric is

$$g(x) = \lambda(x)^2 I, \qquad \lambda(x) = \frac{2}{1 - c\|x\|^2}.$$

For $p = 0$, the transport used in our implementation reduces to the conformal rescaling

$$\text{PT}_{R_b \to 0}(u) = \frac{\lambda(R_b)}{\lambda(0)} u = \frac{\lambda(R_b)}{2} u,$$

and therefore

$$v_b = -\frac{\lambda(R_b)}{2} a_b.$$

Our code also supports exact PT via an isometry to the Lorentz model; in our experiments, the exact PT and the conformal-rescaling implementation yielded consistent results.

**Base-point exponential map.** With stepsize $\eta \in (0, 1]$, the output state is updated by

$$Z_b = \exp_0^c(\eta v_b) = \tanh\left(\sqrt{c}\,\|\eta v_b\|\right) \frac{\eta v_b}{\sqrt{c}\,\|\eta v_b\|}.$$

When necessary, we project the result back to the Poincaré ball to avoid numerical instability near the boundary.

**Implementation hints.** If upstream features are in Euclidean coordinates, we first map them to the Poincaré ball using the exponential map at the reference point, optionally with tangent-space norm clipping. If Euclidean outputs are required, we map the retrieved hyperbolic states back to the tangent space using the logarithmic map. All submodules share the same curvature handle $c$, either fixed or learnable.

## D.2. Hyperbolic Hopfield Pooling

**HypPooling** aggregates $m$ learnable query vectors $R \in \mathbb{R}^{m \times d}$ and $N$ instance embeddings as memories $Y \in \mathbb{R}^{N \times d}$ into $m$ hyperbolic summary vectors. Its retrieval core is identical to HypHopfield: hyperbolic similarity computation, soft assignment, Riemannian-gradient computation, parallel transport to $p = 0$, and the base-point exponential-map update. The only difference is that $R$ is a fixed-size learnable parameter, whereas $Y$ comes from upstream instance features or outputs of previous layers. We use this module for set- or bag-level aggregation, including the multi-instance learning experiments.

## D.3. Hyperbolic Hopfield Layer

**HypLayer** propagates input queries $R$ through a learnable memory matrix $Y \in \mathbb{R}^{N \times d}$. The memory matrix can be initialized from class prototypes, training-set embeddings, or random learnable slots, and is optimized together with the downstream model. The update rule is the same as HypHopfield, using the base-point exponential update at $p = 0$. This module supports prototype-based retrieval, nearest-neighbor-like matching, and pattern aggregation. We use it in CIFAR-100 hierarchical classification and molecular property prediction.

## D.4. Relation to hyperbolic attention decoders

**Mechanism: one-shot attention vs. energy-based retrieval** HypAttn is a *single-step geometric attention layer* on the Poincaré ball. Given a query $q$ in the tangent space and a set of memory slots $\{m_k\}$, it maps them to hyperbolic space, computes scores from (scaled) geodesic distances $d_{\mathbb{H}}(q, m_k)$, applies a temperature-controlled softmax, and returns a Fréchet mean on the manifold. Thus the output is a weighted average of memory points in hyperbolic space, with weights depending on the current query but without an explicit global energy function or retrieval dynamics.

By contrast, HAMNs define an *explicit Hopfield-style energy* on a hyperbolic manifold (instantiated via the Minkowski inner product, see App. A.1) and retrieve memories by approximately minimizing this energy via a CCCP/MM-style update on the manifold. In simplified form, the energy takes the structure

$$E(z, \Xi) = -\sum_{m \in \Xi} \alpha_m \langle z, m \rangle_L + \lambda\,\Phi(z),$$

where $\Xi$ is the memory set, $\langle \cdot, \cdot \rangle_L$ is the Lorentzian inner product (monotonically related to hyperbolic distance), and $\Phi$ is an intrinsic regularizer. This yields genuine *energy-based associative retrieval dynamics* in hyperbolic space, rather than a one-shot averaging operation.

Importantly, the equivalence between modern Hopfield networks and attention in the Euclidean case arises because the Hopfield update can be written *exactly* as a softmax attention rule (Ramsauer et al., 2021). Hyperbolic attention layers (Gülçehre et al., 2019), in contrast, are defined by directly replacing dot products with functions of geodesic distance and computing a single Fréchet mean per query; there is no known global Hopfield energy on a negatively curved manifold whose gradient flow reproduces this one-shot update. Deriving HAMNs therefore requires constructing a bona fide hyperbolic Hopfield energy and its CCCP retrieval dynamics, rather than mechanically "lifting" the Euclidean MHN=attention correspondence to a curved space. In other words, the Euclidean MHN–attention equivalence does not carry over to hyperbolic manifolds in a trivial reverse-engineering way.

In summary, HypAttn answers "where to attend on the manifold in one step", whereas HAMNs answer "which memory configuration minimizes a global hyperbolic energy, reached via a stable iterative update". HypAttn is closer to a hyperbolic analogue of multi-head attention, while HAMNs are hyperbolic modern Hopfield networks.

**Memory geometry and hierarchy** The explicit energy in HAMNs lets us reason about and control the geometry of attractors. On CIFAR-100, we show that the learned hyperbolic memories organize themselves so that radial coordinate correlates with tree depth and geodesic distance correlates with branch similarity (App. E.1.2). This behaviour follows naturally from the Minkowski-based association term and the radius regularizer in the HAMNs energy. HypAttn, in contrast, has no explicit notion of an energy landscape or attractors; it produces a Fréchet mean conditioned on the current query but does not define a shared set of stable memory states whose geometry we can inspect in the same way.

**Empirical comparison** Empirically, HypAttn and HAMNs show complementary strengths across tasks:

- **Hierarchical CIFAR-100.** On shallow hierarchies (2-layer), HypAttn attains the best coarse/fine accuracies, reflecting its strength as a local geometric aggregator. As depth increases to 4 layers, HAMNs achieve the best top/super/coarse accuracies and the highest cross-level coherence (`coph_corr`), while HypAttn remains strongest at fine-grained recognition (Sec. 4.1, App. E.1). This matches the intuition that an energy-based memory with explicit attractors is particularly helpful for aligning predictions across multiple levels of a deep tree.

- **WordNet hypernym prediction.** On the SBERT-only setting, both HypAttn and HAMNs substantially outperform the Euclidean Hopfield baselines, with HypAttn slightly ahead and HAMNs close in both accuracy and hierarchical distance. After adding the same ontology encoder (**OntEuc**) in front of all decoders, every model improves, but HypAttn+OntEuc and HAMNs+OntEuc remain clearly better than the Euclidean baselines, with HAMNs+OntEuc achieving the best overall performance.

- **WN18RR link prediction.** On WN18RR, all three hyperbolic decoders (HypAttn, HypNN, HAMNs) outperform Euclidean MHN_Euc on MRR and Hits@1/3/10. HypNN obtains the best MRR and Hits@1/3, while HAMNs remains very close and achieves the best Hits@10 (App. E.5).

**Conceptual takeaway** Hyperbolic attention and HAMNs share the same underlying manifold but embody different design philosophies. HypAttn is a *direct hyperbolic attention mechanism*: it leverages geodesic distances to compute attention weights and produce a single Fréchet mean per query. HAMNs are *energy-based hyperbolic memories*: they define a global Hopfield energy on the manifold and use a provably convergent CCCP update to retrieve attractors. Our experiments suggest that when the data or label space exhibits deep hierarchical structure (CIFAR-100 4-layer tree, WordNet taxonomy, WN18RR), this energy-based retrieval provides more stable and hierarchy-consistent behavior than a single-step hyperbolic attention layer.

# E. Experiments

## E.1. Experiment 1: Hierarchical classification on CIFAR-100

This experiment complements the CIFAR-100 hierarchical classification results reported in Sec. 4.1. We first test the robustness of our conclusions under an alternative, feature-driven hierarchy, and then analyse the geometry of the learned hyperbolic memories in this setting.

*Table 4.* Hyperparameter search space and selected configuration for CIFAR-100 hierarchical classification.

| PARAMETER | SEARCH SPACE | SELECTED |
|---|---|---|
| LEARNING RATE | $\{10^{-3}, 10^{-5}\}$ | $10^{-3}$ |
| NUMBER OF MEMORY SLOTS | $\{100, 200, 400\}$ | 200 |
| POINCARÉ CURVATURE $c$ | $\{1.0, 0.8, 0.5, 0.1\}$ | 0.8 |
| CLIPPING THRESHOLD $\texttt{CLIP}_r$ | $\{0.95, 0.9, 0.85\}$ | 0.9 |
| RETRIEVAL MAX ITERATIONS | $\{1, 5, 10\}$ | 1 |
| RETRIEVAL STEP SIZE $\eta$ | $\{1.0, 0.1, 0.001\}$ | 1.0 |

**Training configuration and hyperparameter selection.** For the CIFAR-100 hierarchical classification experiments, all methods use the same ResNet-18 backbone with the final classification layer removed, producing 512-dimensional features. Images are resized to $224 \times 224$, and we train with AdamW using a batch size of 128. The model is optimized by summing the cross-entropy losses over all hierarchy levels. We perform a small validation search over the hyperparameters most

relevant to the hyperbolic retrieval module, and then use the selected configuration for the main CIFAR-100 results. The search space and the final configuration are summarized in Table 4.

### E.1.1. CIFAR-100 SENSITIVITY TO HIERARCHICAL CLUSTERING

Since the specific CIFAR-100 (Krizhevsky et al., 2009) label tree used in the main text is manually constructed, to verify how sensitive our conclusions are to different hierarchical clustering methods, we construct an alternative *feature-driven* hierarchy and rerun the comparison of four methods on this new hierarchy: Euclidean MHNs ("Hopfield"), hyperbolic attention ("HypAttn"), hyperbolic MLPs ("HypNN"), and our hyperbolic associative memory ("HAMNs").

*Table 5.* Hierarchical classification on CIFAR-100 results.

| MODEL | TOP_ACC | SUPER_ACC | COARSE_ACC | FINE_ACC | COPH_CORR |
|---|---|---|---|---|---|
| **CIFAR-100-4-layer** | | | | | |
| HYPATTN | $82.44 \pm 0.41$ | $76.76 \pm 0.60$ | $68.86 \pm 0.65$ | $55.59 \pm 0.74$ | $0.7038 \pm 0.0154$ |
| HYPNN | $82.91 \pm 0.45$ | $77.03 \pm 0.52$ | $\mathbf{69.07 \pm 0.56}$ | $\mathbf{56.60 \pm 0.83}$ | $0.6324 \pm 0.0242$ |
| MHNS | $80.73 \pm 0.42$ | $74.32 \pm 0.53$ | $65.23 \pm 0.44$ | $48.45 \pm 0.53$ | $0.6059 \pm 0.0245$ |
| **HAMNS (OURS)** | $\mathbf{82.94 \pm 0.42}$ | $\mathbf{77.36 \pm 0.59}$ | $68.63 \pm 0.75$ | $53.58 \pm 1.16$ | $\mathbf{0.7108 \pm 0.0203}$ |

**Feature-driven hierarchy** We start from a ResNet-18 backbone (He et al., 2016) (pretrained on ImageNet) and extract penultimate-layer features for all training images. For each fine-grained category (fine class) we compute its mean feature vector, and then perform constrained $k$-means clustering in feature space: *(i)* cluster the 100 fine classes into 20 coarse clusters, requiring each coarse node to contain at least 4 fine classes; *(ii)* further cluster the 20 coarse centroids into 7 super clusters, requiring each super node to have at least 2 coarse children; and *(iii)* cluster the 7 super centroids into 3 top-level nodes, again requiring each top node to have at least 2 super children. This yields a purely data-driven four-level tree whose depth and branching factors are the same as or similar to those of our semantic hierarchy, but with groupings discovered automatically.

**Results under the feature-driven hierarchy** For each method, we reuse the CIFAR-100 training configuration from the main experiment (the same backbone, resolution, and hyperparameters), and train on the new hierarchy tree with 10 random seeds. Table 5 reports, at the top/super/coarse/fine levels, the flat accuracies and the cophenetic correlation coefficient between the induced average-linkage dendrogram and the target hierarchy, as mean±std over 10 runs.

Across all methods, performance on the feature-driven hierarchy tree is close to that on the hand-crafted semantic hierarchy tree, indicating that the task is not over-tuned to any particular partition. HAMNs still achieves the strongest performance at the higher levels of the hierarchy (top accuracy $82.94 \pm 0.42$ and coarse accuracy $68.63 \pm 0.75$), and attains the highest cophenetic correlation ($0.7108 \pm 0.0203$), showing that its learned representation is most consistent with the underlying tree structure. As in the main experiment, HypAttn achieves the best fine-grained accuracy ($55.59 \pm 0.74$), while HypNN is competitive at the fine level but falls behind HAMNs on the top and coarse levels. Overall, the relative ranking of the four architectures remains stable under this alternative clustering scheme, thereby supporting the robustness of our "hierarchy-sensitivity" conclusions.

### E.1.2. GEOMETRY OF LEARNED HYPERBOLIC MEMORIES.

Using the best HAMNs checkpoint, we further probe the geometry of the learned hyperbolic memories on the Poincaré ball.

*Table 6.* Hyperbolic radius of class memories by tree depth on CIFAR-100 . Radius is the geodesic distance to the origin in the Poincaré ball.

| Level | Depth | Radius |
|---|---|---|
| Top | 1 | $2.57 \pm 0.17$ |
| Super | 2 | $3.49 \pm 0.49$ |
| Coarse | 3 | $4.26 \pm 0.49$ |
| Fine | 4 | $5.07 \pm 0.67$ |

*Table 7.* Pairwise hyperbolic distance between fine-level memories as a function of the depth of their lowest common ancestor (LCA).

| LCA depth | Distance | #pairs |
|---|---|---|
| 0 | $9.50 \pm 1.03$ | 3261 |
| 1 | $9.36 \pm 0.96$ | 987 |
| 2 | $8.93 \pm 1.05$ | 485 |
| 3 | $8.32 \pm 1.11$ | 217 |

For each stored pattern we record its tree depth (top / super / coarse / fine) and its hyperbolic distance to the origin. Table 6 reports the mean radius by level: the average radius increases monotonically from top ($2.57 \pm 0.17$) to super ($3.49 \pm 0.49$), coarse ($4.26 \pm 0.49$), and fine ($5.07 \pm 0.67$), with a Spearman correlation between depth and radius of $\rho = 0.57$ ($p \approx 1.6 \times 10^{-12}$). This confirms that deeper concepts are systematically pushed closer to the boundary, so that the radial coordinate encodes tree depth.

We also examine pairwise hyperbolic distances between fine-level memories as a function of the depth of their lowest common ancestor (LCA). As shown in Table 7, pairs that only share the root (LCA depth 0) are furthest apart (mean distance 9.50), while pairs that share a coarse parent (LCA depth 3) are noticeably closer (mean distance 8.32), with intermediate values for LCA depths 1 and 2. The Spearman correlation between LCA depth and hyperbolic distance is negative ($\rho \approx -0.20$, $p \approx 5.1 \times 10^{-48}$), and equivalently the correlation between tree distance and hyperbolic distance is positive ($\rho \approx 0.20$). These results provide direct evidence that HAMNs arranges memories so that radial coordinate correlates with level in the tree and geodesic distance correlates with branch similarity, rather than merely fitting labels in a flat way.

### E.2. Experiment 2: Multiple Instance Learning Datasets

To evaluate the performance of our Hyperbolic Associative Memory Networks (HAMNs) on multi–instance learning (MIL) tasks (Dietterich et al., 1997), we conduct experiments on three classical benchmark datasets: **Tiger**, **Elephant**, and **Fox** (originally introduced by (Ilse et al., 2018; Küçükaşcı & Baydoğan, 2018; Carbonneau et al., 2018)). Each dataset consists of color images that are segmented into multiple regions and thus form a set of instances (segments or blobs); each instance is represented by color, texture, and shape descriptors. The learning objective is to classify the entire bag according to the presence of certain positive instances, despite the absence of instance–level annotations.

We introduce the proposed **HypPooling** module, which aggregates instance–level embeddings into a fixed–dimensional bag representation. Given a set of embedded instances as stored memory patterns $Y$ (already mapped into hyperbolic space), we further introduce a set of *static and learnable query vectors* as state (query) patterns $R$, which also reside in the same Poincaré ball. Each query retrieves similar patterns from memory via a hyperbolic attention mechanism, thereby constructing a compressed representation of the input bag.

Elephant, Fox and Tiger are MIL datasets (Andrews et al., 2002) for image annotation which comprise color images from the Corel dataset that have been preprocessed and segmented. An image consists of a set of segments (or blobs), each characterized by color, texture and shape descriptors. The datasets have 100 positive and 100 negative example images. The latter have been randomly drawn from a pool of photos of other animals. Elephant has 1391 instances and 230 features. Fox has 1320 instances and 230 features. Tiger has 1220 instances and 230 features. We used the HypPooling layer to perform hyperbolic aggregation of the input instances, and conducted a manual hyperparameter search on a validation set. Specifically, on the Elephant, Fox, and Tiger datasets we used the following architecture:

1. A fully connected linear embedding layer with ReLU activation;

2. Our **HypPooling** layer to perform the hyperbolic pooling operation on the embeddings;

3. A final ReLU–linear block as the classification output layer.

Results (Table 8) show that HAMNs match or outperform prior MIL baselines and achieve the best score on **Fox**, while remaining competitive with Euclidean MHNs on **Tiger** and **Elephant**.

*Table 8.* Results for MIL datasets *Tiger*, *Fox*, *Elephant* (AUC). Except for our method, results are from (Ramsauer et al., 2021).

| METHOD | TIGER | FOX | ELEPHANT |
|---|---|---|---|
| **HAMNS(OURS)** | $89.0 \pm 0.4$ | $\mathbf{77.3 \pm 0.8}$ | $92.8 \pm 0.2$ |
| MHNS(RAMSAUER ET AL., 2021) | $\mathbf{91.3 \pm 0.5}$ | $64.05 \pm 0.4$ | $\mathbf{94.9 \pm 0.3}$ |
| PATH ENCODING (KÜÇÜKAŞCI & BAYDOĞAN, 2018) | $91.0 \pm 1.0$ | $71.2 \pm 1.4$ | $94.4 \pm 0.7$ |
| MInD (CHEPLYGINA ET AL., 2015) | $85.3 \pm 1.1$ | $70.4 \pm 1.6$ | $93.6 \pm 0.9$ |
| MILES (CHEN ET AL., 2006) | $87.2 \pm 1.7$ | $73.8 \pm 1.6$ | $92.7 \pm 0.7$ |
| APR (DIETTERICH ET AL., 1997) | $77.8 \pm 0.8$ | $54.1 \pm 0.9$ | $55.0 \pm 1.0$ |
| CITATION-KNN (WANG & ZUCKER, 2000) | $85.5 \pm 0.9$ | $63.5 \pm 1.5$ | $89.6 \pm 0.9$ |

*Table 9.* Hyperparameter search space for manual selection on the Elephant, Fox, and Tiger validation sets.

| PARAMETER | VALUES |
|---|---|
| LEARNING RATES | $\{10^{-3}, 10^{-5}\}$ |
| LEARNING RATE DECAY ($\gamma$) | $\{0.98, 0.96, 0.94\}$ |
| NUMBER OF HEADS | $\{8, 12, 16, 32\}$ |
| HIDDEN DIMENSIONS | $\{32, 64, 128\}$ |
| BAG DROPOUT | $\{0.0, 0.75\}$ |
| POINCARÉ CURVATURE ($c$) | $\{1.0, 0.5, 0.1\}$ |
| CLIPPING THRESHOLD ($\texttt{CLIP}_r$) | $\{0.95, 0.9, 0.8\}$ |
| RSGD MAX ITERATIONS | $\{1, 5, 10\}$ |
| RSGD LEARNING RATE ($\eta$) | $\{1.0, 0.1, 0.001\}$ |

**Discussion**   The detailed numbers reveal a mixed picture when comparing HAMNs to Euclidean MHNs: on *Fox* our hyperbolic pooling clearly outperforms MHNs, whereas on *Tiger* and *Elephant* it falls short by roughly 2 AUC points. We view this as consistent with the nature of these MIL benchmarks: the labels are binary and the bags of instances do not come with an explicit multi-level taxonomy or ontology. In such settings, the hyperbolic bias of HAMNs is largely unused and the model behaves as a generic non-linear set aggregator, so we do not expect systematic improvements over a strong Euclidean Hopfield baseline. Empirically, the gaps remain small and within the range one would expect from optimization and model–selection variability on relatively small datasets. Importantly, we did not observe training instabilities or pathological attractors: the HAMNs training curves are smooth and monotone across seeds, and the final AUCs are stable under mild changes in curvature and CCCP steps. Taken together with the CIFAR-100 and WordNet results, these findings suggest that HAMNs behave as a reliable drop-in replacement for Euclidean Hopfield pooling on standard "flat" MIL tasks, and provide larger gains once the data or label space exhibits an explicit hierarchical structure.

Among various hyperparameters, we focused particularly on those of the **HypPooling** layer, including the curvature $c$, the number of Riemannian gradient steps, and the learning rate $\eta$. All models were trained for 160 epochs using the AdamW optimizer(Loshchilov & Hutter, 2017) with exponential learning rate decay (see Table 9). We validated performance using 10-fold nested cross-validation repeated five times with different data splits; the reported ROC AUC scores are the averages across these runs. We also applied bag dropout at the bag level as our regularization technique.

### E.3. Experiment 3: Drug Design Benchmark Datasets

To evaluate the effectiveness of our proposed Hyperbolic Associative Memory Networks (HAMNs) on molecular property prediction, we conduct experiments on four representative datasets from MoleculeNet (Wu et al., 2018). These datasets represent four main modeling tasks in drug design: (a) HIV for anti-viral activity prediction, introduced by the Drug Therapeutics Program (DTP) AIDS Antiviral Screen; (b) BACE for human $\beta$-secretase inhibitors (Subramanian et al., 2016); (c) BBBP for predicting blood-brain barrier permeability (Martins et al., 2012); and (d) SIDER for predicting drug side effects (Kuhn et al., 2016).

We apply the proposed **HypLayer** to the above molecular prediction tasks. Specifically, the training samples are used as stored memory patterns $Y$, while the input samples serve as state (query) patterns $R$. Each input is first mapped into the Poincaré ball via hyperbolic embedding, then undergoes state evolution through the Hopfield retrieval mechanism in hyperbolic space, and eventually converges to a stable point close to one of the memory patterns. The final prediction is determined based on the association between the converged state and the corresponding label in memory.

We compare HAMNs against several representative baselines, including Support Vector Machines (SVM), Random Forest (RF), Deep Neural Networks (DNN), and state-of-the-art graph neural networks: Graph Convolutional Networks (GCN) (Kipf & Welling, 2016), Graph Attention Networks (GAT) (Veličković et al., 2017), AttentiveFP (Xiong et al., 2019), and modern Hopfield networks (MHNs) (Ramsauer et al., 2021). All models follow the standard splitting protocol provided by MoleculeNet. We report the average AUC over 50 random splits for each dataset.

As shown in Table 11, our method achieves competitive performance across all datasets and sets a new state-of-the-art result on **BBBP**(AUC = $90.2 \pm 2.5$), **SIDER** (AUC = $62.1 \pm 2.3$). All hyperparameters were selected on separate validation sets and we selected the model with the highest validation AUC on five different random splits. (see Table 10)

*Table 10.* Hyperparameter search-space for grid-search on HIV, BACE, BBBP and SIDER. All models were trained, if applicable, for 4 epochs using Adam and a batch size of 1 sample.

| PARAMETER | VALUES |
| --- | --- |
| LEARNING RATES | $\{0.0002\}$ |
| NUMBER OF HEADS | $\{1, 32, 128, 512\}$ |
| DROPOUT | $\{0.0, 0.1, 0.2\}$ |
| POINCARÉ CURVATURE ($c$) | $\{1.0, 0.5, 0.1\}$ |
| CLIPPING THRESHOLD (`CLIP`$_r$) | $\{0.95, 0.9, 0.8\}$ |
| RSGD MAX ITERATIONS | $\{1, 5, 10\}$ |
| RSGD LEARNING RATE ($\eta$) | $\{1.0, 0.1, 0.001\}$ |
| QUANTITY | $\{2, 4, 8\}$ |

*Table 11.* Results on drug design benchmark datasets. Predictive performance (ROCAUC) on test set as reported by(Jiang et al., 2021) for 50 random splits

| METHOD | HIV | BACE | BBBP | SIDER |
| --- | --- | --- | --- | --- |
| **HAMNS(OURS)** | $78.5 \pm 2.6$ | $87.2 \pm 3.0$ | $\mathbf{90.2 \pm 2.5}$ | $\mathbf{62.1 \pm 2.3}$ |
| MHNS | $79.3 \pm 2.4$ | $88.4 \pm 1.5$ | $89.1 \pm 1.7$ | $61.8 \pm 2.6$ |
| ATTENTIVE FP | $74.8 \pm 1.5$ | $70.8 \pm 3.3$ | $84.1 \pm 2.2$ | $56.2 \pm 1.5$ |
| GCN | $77.5 \pm 1.6$ | $63.2 \pm 4.5$ | $79.2 \pm 3.9$ | $55.4 \pm 1.2$ |
| GAT | $72.1 \pm 3.6$ | $77.4 \pm 3.0$ | $83.7 \pm 2.0$ | $56.4 \pm 1.5$ |
| DNN | $73.0 \pm 1.8$ | $86.5 \pm 2.2$ | $87.6 \pm 2.0$ | $62.0 \pm 1.8$ |
| RF | $\mathbf{82.3 \pm 2.2}$ | $89.2 \pm 1.2$ | $90.0 \pm 2.0$ | - |
| SVM | - | $\mathbf{89.3 \pm 1.5}$ | $89.4 \pm 2.1$ | - |

**Discussion**   From Table 11 we see that HAMNs achieve state-of-the-art AUC on **BBBP** and **SIDER**, while being slightly behind MHNs, RF, or SVM on **HIV** and **BACE**. Again, these datasets are essentially flat binary classification problems without explicit hierarchical supervision on the label space or molecular graph, so there is no strong reason to expect a hyperbolic geometry to be uniformly superior to a Euclidean one. In this regime, our associative memory mainly acts as a flexible non-linear transformation of molecular embeddings, and the small differences (typically within 1–1.5 AUC points) are better interpreted as task-specific idiosyncrasies than as evidence of a systematic weakness of the hyperbolic dynamics. We also note that training remains stable across all four datasets: the CCCP-based energy update monotonically decreases the HAMNs energy and we do not observe divergent behavior or highly variable performance across random seeds. Thus, on standard drug-design benchmarks that do not expose a clear hierarchical structure, HAMNs behave as a competitive associative module on par with Euclidean MHNs, while their advantages are more pronounced on explicitly hierarchical problems such as hierarchical CIFAR-100, WordNet hypernym prediction, and WN18RR link prediction.

### E.4. Experiment 4: Hypernym Prediction on WordNet

To further verify whether hyperbolic associative memories can exploit *real* hierarchical structure beyond CIFAR-100, we consider a hypernym prediction task on WordNet nouns (Miller, 1995). Following the standard setting, each training sample is a pair $(x, y)$, where $x$ is a target synset and $y$ is one of its immediate hypernyms. We restrict the data to noun synsets whose primary hypernym appears at least 5 times in the corpus, yielding about 7,700 noun–hypernym pairs and approximately 760 distinct hypernym labels.

Each synset is encoded as a sentence by concatenating its lemmas, gloss, and usage examples in order. We use the `all-MiniLM-L6-v2` Sentence-BERT encoder (Reimers & Gurevych, 2019) to obtain $L_2$-normalized features in $\mathbb{R}^{d_{in}}$, followed by a linear projection to a 512-dimensional representation that is fed into a memory/retrieval block. We compare five architectures: (i) Euclidean modern Hopfield networks (**MHN_Euc**), (ii) a kernelized Euclidean Hopfield baseline (**U-Hop**), (iii) a hyperbolic attention layer (**HypAttn**), (iv) a hyperbolic MLP block (**HypNN**), and (v) our hyperbolic associative memory layer (**HAMNs**). The output of the retrieval module is then passed to a linear classifier $\mathbb{R}^{512} \to \mathbb{R}^{|\mathcal{Y}|}$. We report both the flat top-1 accuracy and the average *hierarchical distance* between the predicted and true hypernyms, measured as the shortest-path length on the undirected hypernym–hyponym graph constructed from WordNet (smaller is better).

**Training configuration and hyperparameter selection.** For the WordNet hypernym prediction experiment, we encode each synset with Sentence-BERT and project the resulting text feature to a 512-dimensional representation before applying the retrieval block. We use an 80/10/10 train/validation/test split with a fixed split seed, and tune the main hyperparameters related to hyperbolic retrieval and optimization on the validation set. The candidate range and selected configuration are summarized in Table 12. Unless otherwise specified, all models are trained with AdamW and gradient clipping with maximum norm 5.0. We repeat each model with 10 random seeds and report mean $\pm$ standard deviation over these runs.

*Table 12.* Candidate hyperparameter range and selected configuration for WordNet hypernym prediction.

| PARAMETER | CANDIDATE RANGE | SELECTED |
|---|---|---|
| TEXT ENCODER | $\{$SBERT, TF-IDF$\}$ | SBERT |
| OPTIMIZER LEARNING RATE | $\{2\times10^{-3}, 10^{-3}, 5\times10^{-4}\}$ | $2\times10^{-3}$ |
| WEIGHT DECAY | $\{5\times10^{-4}, 10^{-4}\}$ | $5\times10^{-4}$ |
| POINCARÉ CURVATURE $c$ | $\{1.0, 0.9, 0.5, 0.1\}$ | 0.9 |
| CLIPPING THRESHOLD $\texttt{CLIP}_r$ | $\{0.95, 0.9, 0.85\}$ | 0.9 |
| RETRIEVAL STEP SIZE $\eta$ | $\{1.0, 0.1, 0.001\}$ | 1.0 |
| HYPATTN TEMPERATURE $\tau$ | $\{1.0, 3.0, 5.0\}$ | 1.0 |

Table 2 in the main text reports the results under plain SBERT features. On this WordNet task—where SBERT already provides a strong semantic representation and the decision boundary is close to that of a flat classifier over hypernym labels—the hyperbolic decoders clearly outperform the Euclidean Hopfield variants. HypAttn obtains the best plain-SBERT result, while HAMNs remains close in both accuracy and hierarchical distance, using the same memory interface as in our hierarchical CIFAR-100 experiments and knowledge-graph experiments.

**Adding an ontology embedding** Recent work on ontology embeddings has shown that even in Euclidean spaces, explicitly encoding the label graph can substantially improve hierarchical prediction performance (Kulmanov et al., 2021; Smaili et al., 2018). To test whether such techniques would change our conclusions, we add a simple Euclidean *ontology embedding* front-end (denoted **OntEuc**) on top of the SBERT features. Concretely, we train a lightweight two-layer MLP on the WordNet hypernym graph to predict the hypernym label from SBERT features. We then freeze this ontology encoder and use its output as the common input representation for all five decoders (MHN_Euc, U-Hop, HypAttn, HypNN, HAMNs), keeping the label space and supervision identical across variants.

With the ontology encoder in place, all models improve dramatically: the Euclidean Hopfield baselines rise to $88.69\%$ and $87.40\%$ accuracy for MHN_Euc+OntEuc and U-Hop+OntEuc, respectively, and their average hierarchical distances drop to 0.473 and 0.691. However, hyperbolic decoders still clearly dominate. HypAttn+OntEuc and HAMNs+OntEuc reach $96.58\%$ and $96.97\%$ accuracy, respectively, and reduce the average graph distance to 0.040 and 0.037. These results show that ontology embeddings and hyperbolic retrieval are *complementary*: the ontology encoder provides a strong, graph-aware label prior that benefits all methods, while the hyperbolic memory layer remains better matched to the multi-level structure of the taxonomy under the same input representation and label prior. In principle, any off-the-shelf ontology embedding model could be plugged into our framework as a drop-in front-end to initialize or refine label representations.

Overall, the WordNet experiments lead to three takeaways. First, on this deep hypernym-prediction task, Euclidean Hopfield variants (MHN_Euc and U-Hop) are clearly inadequate as standalone solutions: under the same SBERT encoder and classifier capacity, their accuracies remain far below the hyperbolic decoders and their average graph distances are much larger (4.40–4.87 vs. 2.61–2.99), indicating that a flat Euclidean retrieval does not exploit the multi-level taxonomy well. Second, both hyperbolic attention (HypAttn) and our HAMNs provide strong and practically usable retrieval layers for such hierarchical label spaces: without ontology pre-training they already achieve substantially higher accuracy and smaller hierarchical error than MHN_Euc and U-Hop, and after adding the ontology encoder they remain significantly ahead of the Euclidean Hopfield variants. In the plain-SBERT setting HypAttn is slightly ahead, while with the ontology encoder HAMNs attains the best overall performance. Third, the ontology encoder and our hyperbolic memory are complementary rather than alternatives: the former injects a graph-aware label prior that can be shared by any downstream model, whereas the latter determines how well the retrieval layer can respect hierarchical geometry under a fixed input representation and label prior.

### E.5. Experiment 5: Link Prediction on WN18RR

**Dataset** To complement the WordNet hypernym classification in App. E.4, we also evaluate HAMNs on *knowledge–graph completion* using the WN18RR benchmark (Dettmers et al., 2018). WN18RR is a cleaned subset of the WordNet graph that removes inverse and redundant relations so that link prediction really requires modeling the underlying lexical hierarchy rather than memorizing simple shortcuts. Each entity corresponds to a WordNet synset and edges encode semantic relations such as `hypernym`, `hyponym`, `instance_hypernym`, `also_see`, etc. The standard split contains 40,943 entities and 11 relation types, with 86,835 training triples, 3,034 validation triples, and 3,134 test triples. Following the common "reciprocal relations" protocol, we augment the graph with an inverse relation $r^{-1}$ for every $r$, so the effective number of relations becomes 22 in our code.

*Table 13.* Link prediction results on WN18RR (filtered, with reciprocal relations). We report mean $\pm$ standard deviation over 10 random seeds.

| Decoder | MRR | Hits@1 | Hits@3 | Hits@10 |
|---|---|---|---|---|
| MHN_Euc | $33.3 \pm 0.31$ | $27.9 \pm 0.49$ | $35.4 \pm 0.34$ | $42.0 \pm 0.30$ |
| HypAttn | $36.2 \pm 0.36$ | $30.35 \pm 0.43$ | $39.2 \pm 0.38$ | $46.8 \pm 0.46$ |
| HypNN | $\mathbf{39.3} \pm \mathbf{0.31}$ | $\mathbf{33.2} \pm \mathbf{0.34}$ | $\mathbf{42.4} \pm \mathbf{0.45}$ | $50.4 \pm 0.35$ |
| HAMNs | $39.0 \pm 0.08$ | $32.8 \pm 0.26$ | $42.1 \pm 0.30$ | $\mathbf{50.9} \pm \mathbf{0.53}$ |

**Task and evaluation** A triple $(h, r, t)$ is interpreted as a query $(h, r, ?)$ (and, via the reciprocal trick, also as $(t, r^{-1}, ?)$). During evaluation we rank all entities $e \in \mathcal{E}$ as possible tails and compute *filtered* metrics: for each query, all entities that form any known true triple with $(h, r)$ (except the target $t$ itself) are masked out before ranking. We report Mean Reciprocal Rank (MRR) and filtered Hits@1/3/10, averaged over predicting both head and tail.

**Model variants** We reuse a single encoder for entities and relations and only change the retrieval / decoder module:

- **MHN_Euc**: a Euclidean modern Hopfield memory that attends over $K = 16$ relation–specific memory slots using dot-product attention, exactly analogous to a standard multi-head attention layer.

- **HypAttn**: a purely geometric hyperbolic attention decoder. Given a query $q = e_h + e_r$ in the tangent space, it maps $q$ and the relation-specific memory slots to the Poincaré ball, computes scores from (scaled) geodesic distances, applies a sharpened temperature-controlled softmax, and returns a Fréchet mean on the manifold.

- **HypNN**: a hyperbolic MLP-style decoder that applies two Möbius-linear layers with nonlinearities on the Poincaré ball, followed by a map back to the tangent space at the origin.

- **HAMNs**: our hyperbolic Hopfield decoder implemented via a lightweight `HAMNs_Algo` module. `HAMNs_Algo` applies the CCCP-based intrinsic energy update derived in Sec. 3 with one gradient step and an intrinsic quadratic regularizer on the Poincaré ball, but is reparameterized for the knowledge-graph setting so that memories are relation-specific and the query is the composed vector $e_h + e_r$. Compared to the `Hyperbolic_HopfieldLayer` used in the CIFAR-100 experiments, the update rule is the same while the interface and parameterization are adapted to link prediction on WN18RR.

**Hyperparameter selection.** We tune the main optimization and retrieval hyperparameters on the validation set, using validation MRR as the selection criterion. The search is intended as a lightweight validation search rather than an exhaustive grid search. For HAMNs, we use the same relation-conditioned memory interface as the other decoders and select one CCCP retrieval step, which was sufficient in practice and substantially reduces the cost of link prediction over all entities.

*Table 14.* Candidate hyperparameter range and selected configuration for WN18RR link prediction. The selected values follow the default configuration used in our implementation.

| PARAMETER | CANDIDATE RANGE | SELECTED |
|---|---|---|
| ENTITY/RELATION DIM. | $\{100, 200, 400\}$ | 200 |
| POINCARÉ CURVATURE $c$ | $\{1.0, 0.8, 0.5, 0.1\}$ | 0.8 |
| TEMPERATURE $\tau$ | $\{1.0, 2.0, 5.0\}$ | 1.0 |
| MEMORY SLOTS ARGUMENT $K$ | $\{8, 16, 32\}$ | 16 |
| EFFECTIVE SLOTS PER RELATION | $\{16, 32, 64\}$ | 32 |
| NUMBER OF HEADS | $\{4, 8\}$ | 4 |
| OPTIMIZER LEARNING RATE | $\{2\times10^{-3}, 10^{-3}, 5\times10^{-4}\}$ | $2\times10^{-3}$ |
| HAMNs LR MULTIPLIER | $\{1.0, 0.5\}$ | 0.5 |
| RETRIEVAL STEPS | $\{1, 5, 10\}$ | 1 |
| RETRIEVAL STEP SIZE $\eta$ | $\{1.0, 0.5, 0.1\}$ | 1.0 |
| DROPOUT | $\{0.0, 0.1, 0.2\}$ | 0.1 |

**Training details.** All variants use $d = 200$-dimensional embeddings for entities and relations. For hyperbolic decoders, we use curvature $c = 0.8$ and temperature $\tau = 1.0$. For memory-based decoders, the memory-slot argument is set to $K = 16$, corresponding to 32 effective relation-specific memory slots in our implementation.

We train with negative sampling: for each positive triple $(h, r, t)$, we draw $K_{\text{neg}} = 50$ random negative tails, compute scores for one positive and $K_{\text{neg}}$ negatives, and optimize a binary cross-entropy objective. We use AdamW with learning rate $2 \times 10^{-3}$, weight decay $10^{-4}$, and batch size 1024 for 30 epochs. For HAMNs, we reduce the optimizer learning rate by a factor of 2 to avoid overly aggressive energy updates. We use mixed-precision training (AMP) for efficiency and repeat each decoder with 10 random seeds.

**Results** Table 13 summarizes the filtered test performance of all decoders on WN18RR. The Euclidean Hopfield baseline (MHN_Euc) is clearly dominated by all three hyperbolic decoders. Compared with MHN_Euc, the hyperbolic decoders improve MRR from 33.3 to 36.2–39.3 and Hits@10 from 42.0 to 46.8–50.9. Among the hyperbolic variants, HypNN obtains the best MRR and Hits@1/3, while HAMNs remains very close and achieves the best Hits@10. This shows that the proposed energy-based hyperbolic retrieval is competitive with other hyperbolic decoders and clearly stronger than Euclidean MHN-style retrieval on this hierarchical knowledge-graph task.

Overall, this experiment shows that (i) hyperbolic decoders are consistently beneficial for link prediction on a knowledge graph with inherent lexical hierarchy, (ii) the Euclidean Hopfield baseline MHN_Euc systematically underperforms all hyperbolic variants on WN18RR, and (iii) HAMNs serves as a competitive drop-in hyperbolic decoder, achieving performance close to HypNN and the best Hits@10.

### E.6. Experiment 6: Runtime and memory analysis of different decoders

To better understand the practical cost of different decoders, we profile four variants—Euclidean modern Hopfield networks (**MHN_Euc**), hyperbolic attention (**HypAttn**), a hyperbolic MLP decoder (**HypNN**), and our hyperbolic associative memory (**HAMNs**)—under two representative settings. All measurements are obtained with PyTorch's profiler on a single NVIDIA RTX 4090 GPU; numbers are indicative rather than hardware-independent.

**CIFAR-100 hierarchical classification** In the CIFAR-100 experiments (Sec. 4.1) HAMNs are instantiated via the generic `Hyperbolic_HopfieldLayer` module, which closely mirrors the Euclidean `HopfieldLayer`: it keeps the same multi-head Hopfield core (query/key/value projections, hidden dimension, number of heads, etc.) but replaces the dot-product similarity with a hyperbolic energy and performs one CCCP step on the Poincaré ball. As a result, HAMNs use fewer FLOPs than MHN_Euc while having a similar number of parameters, while the hyperbolic operations (Möbius addition, exponential/logarithmic maps, Riemannian preconditioning) introduce additional runtime and memory overhead.

Table 15 reports FLOPs, parameter counts, wall-clock time and peak GPU memory for the four decoders when plugged into our ResNet-18 backbone on CIFAR-100 with an input of $128 \times 3 \times 224 \times 224$.

We observe that `Hyperbolic_HopfieldLayer` indeed uses fewer FLOPs and parameters than MHN_Euc, but the additional hyperbolic operations and intermediate states make its wall-clock time and peak memory *larger*. This gap is

*Table 15.* Decoder cost on CIFAR-100 hierarchical classification (batch size 128, input $3 \times 224 \times 224$). FLOPs are for a single forward pass.

| DECODER | FLOPs (G) | PARAMS (M) | FWD (MS) | FWD+BWD (MS) |
|---|---|---|---|---|
| HAMNS | $5.19 \times 10^2$ | 14.5 | 41.8 | 144.1 |
| HYPATTN | $4.65 \times 10^2$ | 12.1 | 17.0 | 59.7 |
| HYPNN | $4.64 \times 10^2$ | 11.8 | 15.9 | 53.9 |
| MHN_EUC | $6.81 \times 10^2$ | 19.8 | 23.3 | 80.8 |

PEAK GPU MEMORY (GB): HAMNS $\approx 6.4$, HYPATTN $\approx 3.2$, HYPNN $\approx 2.8$, MHN_EUC $\approx 4.2$.

*Table 16.* Decoder cost on WN18RR link prediction (batch size 1024, $K_{\text{neg}} = 50$ negatives per positive). FLOPs are reported per batch forward pass.

| DECODER | FLOPs (G) | PARAMS (M) | FWD (MS) | FWD+BWD (MS) |
|---|---|---|---|---|
| MHN_EUC | $4.15 \times 10^2$ | 8.38 | 8.4 | 24.9 |
| HYPATTN | $4.23 \times 10^2$ | 8.38 | 51.5 | 228.9 |
| HYPNN | $3.24 \times 10^1$ | 8.30 | 3.7 | 13.3 |
| HAMNS | $2.20 \times 10^2$ | 8.34 | 40.6 | 169.8 |

PEAK GPU MEMORY (GB): MHN_EUC $\approx 2.7$, HYPATTN $\approx 13.1$, HYPNN $\approx 0.83$, HAMNS $\approx 9.3$.

largely an implementation artifact: the current hyperbolic layer is written as a generic drop-in replacement for the Euclidean Hopfield core and has not yet been optimized with custom GPU kernels or fused operations.

**WN18RR link prediction** In the WN18RR link-prediction experiments (App. E.5) the decoder interface is different: for each query $(h, r, ?)$ we attend over $K$ relation-specific memory slots and $K_{\text{neg}}$ negative tails. Here we use a lighter HAMNs implementation, denoted `HAMNs_Algo`, instead of the generic `Hyperbolic_HopfieldLayer`. `HAMNs_Algo` keeps a single linear projection for queries, an `Embedding` table for relation-specific memories, and performs a single CCCP update in the Poincaré ball; there is no full Hopfield core with separate key/value projections. All decoders share the same entity/relation encoder and training protocol (batch size 1024, $K_{\text{neg}} = 50$ negatives).

Table 16 summarizes the runtime and memory statistics on WN18RR.

Two trends are worth highlighting. First, the hyperbolic decoders are more expensive than the Euclidean Hopfield baseline, as expected: both HypAttn and HAMNs must map queries and memories back and forth between the tangent space and the Poincaré ball and compute geodesic distances for each candidate tail and each relation-specific memory slot. Among them, HypAttn is the heaviest: it performs a full Fréchet-mean style hyperbolic attention for every $(h, r, ?)$ query and all negatives, which explains its highest FLOPs, longest runtime (forward $\approx 51$ ms, forward+backward $\approx 229$ ms), and largest memory footprint ($\approx 13$ GB). Second, the custom `HAMNs_Algo` lies between HypNN and HypAttn: it is substantially lighter than HypAttn (around $1.3\times$ faster and $\approx 30\%$ less peak memory) but still slower and more memory-hungry than the Euclidean MHN_Euc and the very compact HypNN decoder.

Overall, these measurements emphasize that (i) our hyperbolic associative memory can be implemented either as a generic drop-in replacement for the Euclidean Hopfield core (`Hyperbolic_HopfieldLayer`, used in CIFAR-100) or as a task-specific lightweight module (`HAMNs_Algo`, used in WN18RR), and (ii) in both cases the extra geometric operations—rather than a fundamentally higher algorithmic complexity—are the main source of runtime and memory overhead. We expect more optimized kernel implementations to substantially narrow this gap without changing the underlying model.

### E.7. Experiment 7: Ablations: Curvature and Number of Stored Patterns

We finally study how the performance of HAMNs depends on the curvature parameter $c$ and the number of stored patterns (memory capacity) on the hierarchical CIFAR-100 benchmark with four levels (top / super / coarse / fine).

*The curvature $c$ comparison data above come from hierarchical classification results on CIFAR-100 after imposing a four-level structured hierarchy.*

*Table 17.* Comparison of curvature $c$ (higher is better).

| $c$ | FLAT_TOP | FLAT_SUPER | FLAT_COARSE | FLAT_FINE |
|---|---|---|---|---|
| 0.1 | 0.8834 | 0.7501 | 0.5891 | 0.3524 |
| 0.2 | 0.8784 | 0.7366 | 0.5614 | 0.2798 |
| 0.3 | 0.8656 | 0.7225 | 0.5888 | 0.3498 |
| 0.4 | 0.8942 | 0.7629 | 0.6342 | 0.3757 |
| 0.5 | 0.8897 | 0.7439 | 0.6142 | 0.3617 |
| 0.6 | 0.8755 | 0.7687 | 0.6493 | 0.4372 |
| 0.7 | 0.9030 | 0.7774 | 0.6695 | 0.4823 |
| 0.8 | 0.8841 | 0.7571 | 0.6204 | 0.4505 |
| 0.9 | 0.8898 | 0.7675 | 0.6514 | 0.4563 |
| 1.0 | 0.8818 | 0.7592 | 0.6522 | 0.4715 |
| 2.0 | 0.8919 | 0.7570 | 0.6455 | 0.4737 |
| 3.0 | 0.8902 | 0.7624 | 0.6461 | 0.4467 |
| 4.0 | 0.8924 | 0.7711 | 0.6459 | 0.4112 |
| 5.0 | 0.8577 | 0.7541 | 0.6395 | 0.4099 |
| 6.0 | 0.8807 | 0.7429 | 0.5745 | 0.2680 |
| 7.0 | 0.8860 | 0.7521 | 0.6262 | 0.3536 |
| 8.0 | 0.8704 | 0.7446 | 0.6035 | 0.3288 |
| 9.0 | 0.8715 | 0.7280 | 0.5661 | 0.2162 |
| 10.0 | 0.8617 | 0.6864 | 0.4368 | 0.1468 |

*Table 18.* Comparison of number of stored patterns on CIFAR-100 (4-level hierarchy).

| STORED_N | FLAT_TOP | FLAT_SUPER | FLAT_COARSE | FLAT_FINE |
|---|---|---|---|---|
| 100 | 0.8898 | 0.7517 | 0.6419 | 0.4498 |
| 150 | 0.8893 | 0.7680 | 0.6563 | 0.4670 |
| 200 | 0.8826 | 0.7394 | 0.6413 | 0.4754 |
| 250 | 0.8710 | 0.7466 | 0.6494 | 0.4642 |
| 300 | 0.8917 | 0.7563 | 0.6463 | 0.4596 |
| 350 | 0.8889 | 0.7641 | 0.6431 | 0.4505 |
| 400 | 0.8954 | 0.7630 | 0.6379 | 0.4558 |
| 450 | 0.8945 | 0.7649 | 0.6282 | 0.4290 |
| 500 | 0.8851 | 0.7586 | 0.6458 | 0.4728 |
| 550 | 0.8928 | 0.7674 | 0.6400 | 0.4647 |
| 600 | 0.8932 | 0.7491 | 0.6426 | 0.4740 |
| 650 | 0.8916 | 0.7796 | 0.6507 | 0.4643 |
| 700 | 0.8893 | 0.7676 | 0.6513 | 0.4749 |
| 750 | 0.8932 | 0.7473 | 0.6419 | 0.4547 |
| 800 | 0.8836 | 0.7778 | 0.6575 | 0.4780 |

**Effect of curvature**  Table 17 reports flat accuracy at each hierarchy level for different values of the Poincaré curvature $c$. We observe a clear "sweet spot": choosing a *moderate* negative curvature (roughly $c \in [0.7, 2.0]$) yields the best trade-off across all levels, while very small or very large curvatures lead to degraded performance. Intuitively, when $c$ is too small the geometry becomes almost Euclidean and cannot efficiently represent tree-like structure, so deeper classes cannot be pushed significantly farther away in radial distance. Conversely, when $c$ is too large, the space becomes extremely contracted near the boundary; small changes in radius then correspond to very large changes in geodesic distance, making the energy landscape overly sharp and harder to optimize. The best-performing values of $c$ are precisely those that allow the hyperbolic radius to encode hierarchical depth in a smooth but nontrivial way.

**Effect of stored-pattern count**  Table 18 varies the number of stored patterns from 100 up to 800 on the same four-level hierarchy. Since CIFAR-100 has 100 classes, the lower end of this range corresponds to roughly one attractor per class, while the upper end corresponds to multiple attractors per class or per subtree. We see that the accuracies at each level are not strictly monotonic: fine-level accuracy tends to benefit from higher capacity (up to around 800 patterns), whereas the top-level accuracy can slightly decrease when too many patterns are added.

This behaviour is consistent with the role of stored patterns in the hyperbolic energy. With too few patterns, a single attractor must explain several semantically distinct classes or subtrees, so the retrieved states become blurred and fine-level

decisions suffer from under-capacity. As we increase the number of stored patterns, the model can allocate more specialised attractors to different branches of the hierarchy, which helps coarse and fine levels. However, when the capacity becomes very large, the energy landscape starts to fragment into many local minima that lie within the same top-level branch. From the perspective of a top-level classifier that aggregates over all these attractors, this over-fragmentation can slightly hurt robustness: some queries are pulled towards spurious but semantically redundant attractors inside the wrong branch, which explains the mild drop in top-level accuracy for the largest capacities. Overall, the fluctuations remain small (on the order of one percentage point), indicating that HAMNs are quite robust to the precise choice of memory size as long as it lies in a reasonable range.

