# OpenReview forum: "Hyperbolic Associative Memory Networks"
_ICML.cc/2026/Conference — ICML 2026 regular_

### Official Review · Reviewer_YTs9 · 2026-03-05

**Soundness:** 4
**Presentation:** 3
**Significance:** 4
**Originality:** 4
**Overall Recommendation:** 5
**Confidence:** 4

**Summary:**

he paper develops theory of modern Hopfield networks (also known as Krotov-Hopfield networks) to a geometric direction where the Euclidean space is replaced by a hyperbolic space, that is curved manifold. This approach is very interesting as it
combines differential geometry and associative memory studies in an innovative way. Similar approached in different context has been used in network theory (e.g. applying small world hypothesis) where network-type structures are embedded in a hyperbolic space. The key benefit in this is that the volume of balls in hyperbolic space of radius $R$ grows exponentially when $R$ becomes larger.
Thus, one can pack exponentially many small balls (that is, exponentially many constant separated points) in a ball radius $R$ when  $R\to \infty$. Such packing property is very useful for algorithms storing memories. The key idea of the paper is very nice and clear.
The paper is also written in an accessible way that makes it useful for mathematically oriented researchers working on ML.

**Compliance With Llm Reviewing Policy:**

Affirmed.

**Key Questions For Authors:**

1. In Section 2.1 the memory matrix $\Theta$ is described as $N \times K$, while later the memories are vectors in $\mathbb{R}^d$. Please unify these dimensions globally (e.g.\ consistently using $K$ or $d$), or add comment on use of symbols

2. Please state near the beginning (e.g. in  Section 2) the dimension $n$ of the hyperbolic manifold $M$ under consideration. Currently this information is postponed to the appendix, which is too late for eaders.

3. Sign convention: In Section 2.2.3 the inner product in $M$ is defined by $-\cosh(d_M(x,y))$. Why is the sign negative?  If the sign is conventional, what changes if one uses $+\cosh(d_M(x,y))$?

4.  In Section 2.2.5 you relate hierarchical data structures to hyperbolic geometry. Please add details on the mechanism in your setting: since hyperbolic space has no canonical global origin, what structure in the model selects a distinguished ``root'' or radial direction, if any? Otherwise, how should the hierarchy intuition be interpreted?

5. In formula (4), is $v \in T_pM$ for some base point $p \in M$? If so, please specify the base point explicitly.

6. In formula (7), what is the range of the index $i$ (e.g. $i \in \{1,\dots,N\}$)?

7. In formula (8), what does the symbol $p$ denote? Is it the same $p$ as the vector in (7), or the origin point in (5), or something else? Please clarify.

8.  After formula (9), why is parallel transport used at that stage? Is it required to compare objects that are in different tangent spaces?

9. In Appendix, formula (11) uses the Euclidean ``$L$'' inner product instead of the hyperbolic inner product (as used later). Why is this the appropriate choice at that point?

10. The proof of Proposition B.1 is currently only a sketch. Please provide a complete proof, or cite results in literature. The results is very interesting.

11.  Proposition B.2 discusses packing properties. Can you derive an explicit memory-capacity statement similar to the classical Hopfield capacity results see e.g.  McEliece, Posner, Rodemich, and Venkatesh (1987), IEEE Transactions on Information Theory, ``The Capacity of the Hopfield Associative Memory''?

12. At the end of Appendix C.1 you introduce Möbius addition. Please add motivation and references.

**Limitations:**

Yes

**Strengths And Weaknesses:**

Soundness: The analysis is sound and arguments are well justified

Presentation: The submission clearly written and well structured

Significance:

-Main idea is clear and interesting. The paper combines hyperbolic geometry and theory of modern Hopfield networks in a novel way.

-Theoretical framework is the main contribution and is developed in an interesting way.

-Although not the central focus, the numerical experiments are promising and are well documented to support the theoretical results

Originality: The work provides novel ideas that are inspiring

---

> ### Author Rebuttal · Authors · 2026-03-28
>
> Thank you for the careful and constructive review.
>
> We will unify notation: Sec.2.1 uses MHN notation ($K$),while Sec.3 uses $d$ after fixing $ \iota_p:\mathbb{R}^d\to T_p\mathcal M $.We will state earlier that $\mathcal M$ is $d$-dimensional and use one notation throughout.The sign in $-\cosh(d_{\mathcal M}(x,y))$ ensures that nearer points have larger similarity,as in MHNs; using $+\cosh(\cdot)$ would reverse the ordering and require flipping the energy sign.Hyperbolic space has no canonical semantic root.In the Poincaré ball we use $0$ as the reference point $p$; in equivalent models one uses the corresponding base point under the isometry.Thus radial depth is defined relative to $p$,which is an anchor for encoding/regularization rather than an intrinsic root.
>
> In Eq.(4),$v\in T_p\mathcal M$,with $p$ as above;in Eq.(7),$i=1,\ldots,N$,consistent with Sec.3.1;in Eq.(8),$p$ again denotes the reference point,and we will rename the softmax weights to avoid overload.Parallel transport in Eq.(9) is needed because the gradient lies in $T_{\xi^{(t)}}\mathcal M$ whereas the tangent surrogate is formed in $T_p\mathcal M$.The norms/inner products there are those of the fixed tangent space $T_p\mathcal M$.Also,$\langle\cdot,\cdot\rangle_L$ in Appendix Eq.(12) is not Euclidean but the Minkowski inner product in the Lorentz model,i.e.,an equivalent model-specific expression of the same hyperbolic quantity.
>
> We agree that the storage-capacity discussion is too high-level.We therefore add a **random-memory capacity theorem for HAMNs**.
>
> **Random shell model.** Let
> $u_1,\dots,u_N \overset{i.i.d.}{\sim}\mathrm{Unif}(S^{d-1})$,
> and define memories by
> $$x_i=\exp_p(R_d u_i),\qquad R_d=K\sqrt{d-1},\ K>0.$$
> Thus all memories lie on the geodesic sphere of radius $R_d$ centered at $p\in \mathbb H_c^d$.
>
> Our proof builds on the deterministic margin / one-step basin propositions in the appendix: if all memories are pairwise $\delta$-separated and $q\in B_{\mathbb H}(x_k,\rho)$ with $0<\rho<\delta/2$,writing $\varepsilon=\delta/2-\rho$,then
> $$p_k(q)\ge \frac{1}{1+(N-1)e^{-\theta \Delta(\delta,\varepsilon)}},
> \quad
> \Delta(\delta,\varepsilon)=2\sinh\Big(\frac{\delta}{2}\Big)\sinh(\varepsilon).$$
> Hence,if $N\le 1+\exp(\theta\Delta(\delta,\varepsilon))$,then $p_k(q)\ge 1/2$; by Proposition B.3,there exists a step size such that $B_{\mathbb H}(x_k,\rho)$ is one-step invariant.
>
> It remains to show that random memories are pairwise separated w.h.p.For independent
> $u,v\sim\mathrm{Unif}(S^{d-1})$,
> the random variable $Z=\langle u,v\rangle$ has density
> $$f_d(z)=C_d(1-z^2)^{(d-3)/2},\qquad C_d=\frac{\Gamma(d/2)}{\sqrt\pi\,\Gamma((d-1)/2)},$$
> yielding
> $$\mathbb P(\langle u,v\rangle\ge t)\le A\sqrt d\,e^{-(d-3)t^2/2},\qquad t\in(0,1),$$
> for some universal constant $A>0$.A union bound over all $\binom{N}{2}$ pairs gives
> $$\mathbb P\Big(\max_{i<j}\langle u_i,u_j\rangle>t\Big)
> \le \binom{N}{2}A\sqrt d\,e^{-(d-3)t^2/2}.$$
> Thus,if
> $$\binom{N}{2}A\sqrt d\,e^{-(d-3)t^2/2}\le p,$$
> then with probability $\ge 1-p$,all $i\neq j$ satisfy $\langle u_i,u_j\rangle\le t$.
>
> Using the hyperbolic law of cosines,this implies geodesic separation.For
> $x_i=\exp_p(R_d u_i)$ and $x_j=\exp_p(R_d u_j)$,
> $$d_{\mathbb H}(x_i,x_j)\ge \delta_{d,c}(t):=
> \frac{1}{\sqrt c}\operatorname{arcosh}\Big(\cosh^2(\sqrt cR_d)-\sinh^2(\sqrt cR_d)t\Big).$$
> Hence,with probability $\ge 1-p$,all memories are pairwise $\delta_{d,c}(t)$-separated.Applying the deterministic margin / one-step basin proposition then shows that,for any $\rho_d<\delta_{d,c}(t)/2$ satisfying the above condition,the balls $B_{\mathbb H}(x_i,\rho_d)$ are pairwise disjoint one-step invariant basins.
>
> Finally,let
> $$N_d=\lfloor e^{\gamma(d-1)}\rfloor,\qquad 0<\gamma<t^2/4,$$
> then the failure probability tends to $0$ since
> $$\binom{N_d}{2}A\sqrt d\,e^{-(d-3)t^2/2}
> \le \frac{A}{2}\sqrt d\,\exp\Big(2\gamma(d-1)-\frac{d-3}{2}t^2\Big)\to 0.$$
> Moreover,since $R_d=K\sqrt{d-1}$,
> $$\delta_{d,c}(t)=2R_d+O(1)=2K\sqrt{d-1}+O(1),$$
> so the deterministic basin condition is not the bottleneck.Therefore,HAMNs can store
> $$N_d=e^{\Omega(d)}$$
> random memories with high probability,each with a one-step retrieval-stable basin.More explicitly,there exists $\kappa>0$ such that for all sufficiently large $d$,
> $$N_d\ge \exp(\kappa(d-1))$$
> indeed,under the above derivation,any fixed $\kappa<t^2/4$ is admissible.Equivalently,if the random-memory capacity under this shell model is denoted by $\mathrm{Cap}^{\mathrm{rand}}_{\mathrm{HAMN}}(d)$,then for any fixed $\kappa<t^2/4$ and all sufficiently large  $d$,
>
> $$\mathrm{Cap}^{\mathrm{rand}}_{\mathrm{HAMN}}(d)\ge \exp(\kappa(d-1))$$
>
> We stress that this is a **continuous random-memory capacity theorem for HAMNs**,not the random binary Hebbian capacity theorem of *The Capacity of the Hopfield Associative Memory*.
>
> Due to the rebuttal length limit,if accepted we will include the full statements/proofs of Proposition B.1 and the theorem below,and add motivation/references for Möbius addition in App.C.1.

---

> > ### Author Rebuttal · Reviewer_YTs9 · 2026-04-01
> >
> > The authors have planned to unify the notations according to the discussions. Also, the arguments on the random-memory capacity theorem for HAMNs appear convincing.
> > I keep my score on 5 and recommend the acceptance.

---

> > > ### Author Response · Authors · 2026-04-04
> > >
> > > Thank you very much for the careful reading and the positive acknowledgement. We especially appreciate your thoughtful comments on the notation and the random-memory capacity theorem, and your support for the paper.

---

### Official Review · Reviewer_nAJf · 2026-03-06

**Soundness:** 3
**Presentation:** 3
**Significance:** 2
**Originality:** 2
**Overall Recommendation:** 4
**Confidence:** 5

**Summary:**

This paper proposes Hyperbolic Associative Memory Networks (HAMNs), a hyperbolic extension of modern Hopfield networks. The core idea is to move associative memory retrieval from Euclidean space to hyperbolic space in order to better capture hierarchical structures in data. The method maps query and memory vectors into a negatively curved manifold via exponential maps, defines a hyperbolic energy function based on a Minkowski-style similarity measure, and performs retrieval via Riemannian optimization on the manifold. Experiments evaluate the approach on hierarchical classification tasks (e.g., CIFAR-100 with constructed label hierarchies), WordNet tasks, and several weak-hierarchy benchmarks. Overall, I feel this paper strives to investigate a central concept: whether embedding associative memory dynamics into hyperbolic geometry can improve retrieval performance for hierarchical data.

**Compliance With Llm Reviewing Policy:**

Affirmed.

**Final Justification:**

While I am not fully convinced by the empirical gain, the new theory result of capacity characterization improves my confidence

**Key Questions For Authors:**

1. How does the proposed method compare to simpler alternatives such as hyperbolic attention or hyperbolic embedding layers combined with standard MHNs? I would expect some baselines to compare with in this respect.

2. Can the authors provide more rigorous theoretical guarantees regarding associative memory capacity in hyperbolic space?

3. How sensitive is the method to curvature choice and manifold parameterization?

4. Would the approach scale to large foundation models where associative memory layers may operate on high-dimensional representations?

**Limitations:**

This paper presents a coherent extension of modern Hopfield networks into hyperbolic geometry and explores the impact of negative curvature on hierarchical retrieval tasks. The approach is technically reasonable and empirically evaluated across multiple datasets.

However, my main concerns are as follows: the conceptual novelty relative to existing hyperbolic neural architectures is somewhat limited, and the empirical improvements are modest. Additional theoretical analysis and clearer practical advantages would strengthen the contribution.

**Strengths And Weaknesses:**

# Strengths
**1. Interesting geometric extension of associative memory.** An important aspect presented by the study is the extension of modern Hopfield networks into hyperbolic space. Since modern Hopfield networks are closely related to attention mechanisms, exploring geometric alternatives to Euclidean similarity is a meaningful direction. Specifically, the proposed framework modifies:
- the representation space (Euclidean → hyperbolic),
- the similarity function (dot product → hyperbolic similarity),
- and the retrieval optimization (Euclidean gradient descent → Riemannian optimization).
This provides a coherent geometric formulation of associative memory in negatively curved manifolds.

**2. Reasonable motivation for hierarchical data.** Hyperbolic geometry is well known to represent hierarchical structures efficiently due to exponential volume growth and natural embedding of tree-like data structures. Applying this idea to associative memory retrieval is conceptually sensible. The paper also articulates the hierarchy-sensitivity hypothesis, predicting that hyperbolic models should outperform Euclidean ones primarily when strong hierarchical structure is present.

**3. Extensive experimental evaluation.** The experimental section includes several categories of tasks: Hierarchical classification on CIFAR-100 with multi-level label trees, WordNet taxonomy and knowledge-graph benchmarks, and Weak-hierarchy tasks (MIL datasets, molecular prediction). This set of experiments attempts to test the abovementioned proposed hypothesis across multiple regimes.

# Weaknesses
**1. Conceptual novelty is somewhat limited.** The main idea of the proposed work is performing similarity-based retrieval in hyperbolic space, which is not fundamentally new. Hyperbolic embeddings and hyperbolic neural networks have already been explored extensively. In the proposed method, the key change relative to standard MHNs is largely replacing the Euclidean similarity with a hyperbolic similarity function, and performing updates using Riemannian optimization.

While the integration into a Hopfield-style energy formulation is clean, the conceptual advance over existing hyperbolic attention or hyperbolic representation learning frameworks appears incremental.

**2. Theoretical claims are somewhat weak.** The paper claims that hyperbolic geometry allows exponentially more recall wells due to geometric packing arguments. However, the theoretical discussion is **fairly high level**. The formal guarantees connecting the geometry to improved associative memory capacity remain limited, and the connection between the packing argument and practical neural network performance is not fully established.

A stronger theoretical characterization of memory capacity or retrieval stability in hyperbolic space would strengthen the contribution.

**3. Empirical improvements are modest.** Although the experiments show improvements on deeper hierarchical structures, the gains are generally moderate. In several cases of Table 1, Euclidean baselines remain competitive, and improvements appear relatively small in absolute terms.  Thus, the empirical results do not always clearly demonstrate a decisive advantage of the proposed approach.

**4. Computational overhead.** The method introduces additional computational complexity due to $\texttt{exp}$ and $\texttt{log}$ maps, geodesic distance computations, and Riemannian gradient updates. The paper itself notes that despite fewer theoretical FLOPs, the implementation leads to higher runtime and memory usage due to hyperbolic operations. For large-scale deep learning systems, this overhead may limit practical adoption.

---

> ### Author Rebuttal · Authors · 2026-03-28
>
> Thank you for the careful and constructive review.
>
> **On novelty.** We respectfully clarify that HAMNs are not obtained by simply substituting a hyperbolic similarity into standard MHNs. Standard MHNs define both energy and updates in Euclidean/Hilbert space, with retrieval
>
> $$
> \xi' = X\mathrm{softmax}\left(\beta X^\top \xi\right),
> $$
>
> which underlies the MHN-attention equivalence shown by Ramsauer et al. Once associative retrieval is moved to a negatively curved manifold, not only the similarity but the whole geometric basis of retrieval must be redefined: the state space, regularizer, gradient, and update. Accordingly, HAMNs redefine the hyperbolic energy, geodesic regularizer, and manifold retrieval update via Riemannian CCCP, using log/exp maps and parallel transport to keep retrieval on-manifold and geometrically consistent. Thus, this is not a simple similarity replacement, but an intrinsic reformulation of Hopfield-style associative memory in negative curvature.
>
> **On empirical gains.** Our claim is the **hierarchy-sensitivity hypothesis**: the advantage of negative curvature should become clearer as the hierarchy becomes deeper; for shallow or weakly hierarchical tasks, competitive Euclidean performance is expected. The experiments are designed to test this hypothesis and are consistent with it, and the results are consistent with it. In CIFAR-100 hierarchical classification, as depth increases, Euclidean methods progressively degrade, while HAMNs maintain stable semantic discrimination. Moreover, in the real hierarchical tasks in App. E.4/E.5, the gap is clearer: e.g., in WordNet hypernym prediction (Tables 9,10) and WN18RR (Table 11), Euclidean Hopfield and hyperbolic methods show larger differences.
>
> **On simpler alternatives.** We first clarify that **hyperbolic attention is already a main baseline in the paper**, and we analyze its mechanism and empirical behavior vs. HAMNs in the experiments/App. E.8. Specifically, HypAttn is a one-step geometric attention aggregation, whereas HAMNs define an explicit Hopfield-style hyperbolic energy and perform associative retrieval via iterative CCCP updates. We understand the reviewer may further mean another simpler alternative: **hyperbolic embeddings followed by standard MHNs**. This can certainly serve as an extrinsic baseline, but its retrieval mechanism still follows the Euclidean energy and update of standard MHNs. In other words, this still treats representations in hyperbolic space as if they were Euclidean vectors during retrieval. As a result, hyperbolic geometry enters only the representation layer, rather than the memory retrieval mechanism itself. For this reason, we regard it as a useful supplementary baseline, rather than a substitute for our method. Our contribution lies precisely in moving both the Hopfield-style energy function and the retrieval update dynamics to a negatively curved manifold.
>
> **On sensitivity to curvature and manifold parameterization.** Our method is not tied to one specific hyperbolic parameterization. The derivation is model-agnostic for constant-negative-curvature hyperbolic spaces; we use the Poincaré ball mainly because it is more mature in practice. Empirically, HAMNs are not highly sensitive to curvature; App. E.7 shows stable performance over a moderate range. Intuitively, if curvature is too small, the space becomes closer to Euclidean and the hierarchical advantage weakens; if it is too large, optimization near the boundary becomes harder. We also acknowledge a limitation: this paper only instantiates the Poincaré ball and does not yet explore other manifolds.
>
> **On computational cost and scalability to large foundation models.** In form, the answer is yes, since HAMNs, like MHNs, are plug-in memory modules that can store/retrieve raw inputs, intermediate features, or learned prototypes. At the same time, we do not want to overclaim that the current implementation is already optimized for foundation-model scale. As noted in the paper, hyperbolic operations (e.g., exp/log maps and geodesic distance computations) introduce extra runtime and memory cost, though this is also a general challenge for existing hyperbolic neural modules. Hence, we currently see HAMNs as more suitable for **selectively replacing hierarchy-sensitive modules**, rather than indiscriminate deployment throughout large foundation models.
>
> **On stricter capacity characterization.** We agree this is important. Briefly, beyond the geometric separation and basin analysis already in the paper, we further add in the rebuttal a stronger capacity characterization linking retrievability under random memories to separation on hyperbolic shells. A more detailed discussion is provided in our reply to Reviewer YTs9.

---

> > ### Author Rebuttal · Reviewer_nAJf · 2026-04-02
> >
> > Thank authors for the response. While I am not fully convinced by the empirical gain, I like the new theory result of the capacity characterization. I will thus keep my original score and increase the confidence by one level.

---

> > > ### Author Response · Authors · 2026-04-04
> > >
> > > Thank you very much for the careful reading and the positive acknowledgement. We especially appreciate your recognition of the new capacity-characterization result and your constructive feedback throughout the review process.

---

### Official Review · Reviewer_Cqb5 · 2026-03-11

**Soundness:** 3
**Presentation:** 3
**Significance:** 3
**Originality:** 3
**Overall Recommendation:** 3
**Confidence:** 3

**Summary:**

The paper proposes HAMNs, which map query and memory vectors from Euclidean space to a constant negative curvature manifold via exponential maps, then define a Hopfield-style energy function in hyperbolic space.

The central hypothesis is clearly stated: when data or label spaces carry deep hierarchical structure, HAMNs should outperform Euclidean MHNs; when the hierarchy is shallow, weak, or nearly absent, HAMNs should match Euclidean performance.

**Compliance With Llm Reviewing Policy:**

Affirmed.

**Final Justification:**

I maintain my score, a clearer main-text exposition, stronger factor-isolating ablations, and more consistent multi-seed/significance reporting on the hierarchy-sensitive experiments could enhance the quality of this work.

**Key Questions For Authors:**

1. Do the performance gains come from hyperbolic geometry, from iterative retrieval dynamics, or from the additional regularizer and memory parameterization? An ablation that isolates these components would strengthen the empirical claims.

2. The fairness and stability of the experiments need clarification. In particular, why does WN18RR report only a single-run result? Are the hyperparameter tuning budgets consistent across all baselines for the WordNet and molecular tasks? Providing multi-seed results or a significance analysis would help assess reliability.

3. Are the theoretical conditions stated in the paper actually satisfied during training, or do they function only as local heuristics? Empirical evidence or a more careful discussion of when these conditions hold would be necessary to support the theoretical claims.

**Strengths And Weaknesses:**

The problem framing and method motivation align well. Using hyperbolic geometry to model hierarchical structure has strong geometric justification, and combining it with the Hopfield attractor and associative memory retrieval paradigm is conceptually natural and compelling.

The experiments are organized with reasonable self-consistency. Rather than cherry-picking only favorable tasks, the authors deliberately include both deep-hierarchy and weak-hierarchy settings to test the hierarchy-sensitivity hypothesis.

---

The theoretical arguments for monotone energy descent, local concavity, and attraction basins all rely on fairly strong conditions, and several results are presented only as proof sketches. There is insufficient empirical evidence to confirm that these conditions actually hold during training.

The hyperbolic packing bound is placed in close proximity to the classical capacity analysis of Euclidean MHNs, which can mislead readers into treating them as directly comparable quantities. In reality, one concerns geometric packing radius while the other concerns dimensionality and random pattern capacity.

The important material is excessively delegated to the appendix, including theoretical details, the connection to hyperbolic attention, several experimental setups, runtime analysis, and important supplementary tables. This undermines the self-containedness of the main text.

The result is inconsistent across tasks. Some tasks report mean and variance over multiple runs, while others report only a single-run result.

The energy function is decomposed as E_cvx + E_cave, with the intent of applying Riemannian CCCP via geodesic convexity and concavity. However, Appendix A.4.1 already acknowledges that E_cave is not naturally concave.

The condition the authors impose to enforce concavity of E_cave is most likely to break down precisely near attractors where p_i is sharply peaked, which is exactly the regime where the algorithm most needs theoretical support.

---

> ### Author Rebuttal · Authors · 2026-03-28
>
> Thank you for the careful and constructive review.
>
> **On the concavity of $E_{\mathrm{cave}}$ and the scope of the theory.** We agree that the current theoretical results rely on fairly strong conditions, and should be understood more accurately as **local, conditional explanations** rather than unconditional guarantees over the entire optimization process. Concretely, the curvature of $E_{\mathrm{cave}}(\xi)=-F(\xi)$ is region-dependent. Since $F$ is a log-sum-exp aggregation of memory similarities, in competitive regions the variance term in its Hessian may dominate and the softmax weights have not yet collapsed to a single memory, which can induce local concavity of $-F$. By contrast, near a dominant-memory region, if one softmax weight becomes very sharp, log-sum-exp may degenerate to a single term, and then $-F$ may become locally convex rather than locally concave. Thus, the CCCP-style interpretation is better viewed as a regional property in competitive zones, and not necessarily as a property that must remain valid near the final attractor. We will clarify this more explicitly.
>
> To probe this empirically, we added a local second-difference test along training trajectories. Before each update, at the current state, we sample multiple geodesic directions and evaluate
> $$
> \Delta_\varepsilon^2 E_{\mathrm{cave}}(q;u)
> =\frac{E_{\mathrm{cave}}(\exp_q(\varepsilon u))-2E_{\mathrm{cave}}(q)+E_{\mathrm{cave}}(\exp_q(-\varepsilon u))}{\varepsilon^2}.
> $$
> We count local concavity as holding if the worst sampled second difference over all sampled directions is still below a threshold $\tau$. Using 8 retrieval steps and continuous sampling throughout training, we collected about $3.2\times 10^7$ probes and did not observe any local non-concavity violation ($\text{nonconcave percentage}=0\%$). This suggests that although $E_{\mathrm{cave}}$ may become locally convex in the extreme single-memory limit in theory, along the regions actually visited by optimization, local concavity is empirically stable.
>
> **On where the empirical gains come from.** We agree that a finer-grained attribution analysis would further strengthen the paper. That said, the current baselines already provide partial factor isolation: `MHN_Euc vs. HAMNs` reflects the change from Euclidean to hyperbolic retrieval, while `HypAttn vs. HAMNs` is more focused on comparing one-step hyperbolic aggregation with explicit energy-driven iterative retrieval. Therefore, our current conclusion is intentionally cautious: the gains should not be attributed to any single factor alone, but rather to the joint effect of hyperbolic geometry, energy-driven iterative retrieval, and the associated memory-module design.
>
> **On consistency of experimental reporting.** We agree that the reporting format is not fully uniform across tasks. For the real hierarchical tasks in App. E.4/E.5, our main goal was to test whether the proposed method can adapt as a plug-in retrieval module in real, strongly hierarchical settings, rather than to turn these experiments into full benchmark-style competitions. Accordingly, we used relatively simple, validation-oriented tuning there, and in some cases reported single-run results. Even so, we believe the current results are sufficient to support the empirical conclusion that the proposed method can adapt to real hierarchical tasks, and that a clear gap remains between Euclidean MHN-type retrieval and hyperbolic retrieval. We also note that in both experiments all decoders share the same entity/relation encoder and training protocol, which at least ensures basic fairness at the decoder level. We agree that multi-seed or statistical analysis would further strengthen the presentation, and we will clarify the intended role of these experiments more explicitly.
>
> **On the additional storage-capacity discussion.** We agree this is important. Briefly, beyond the geometric separation and basin analysis already in the paper, we further add in the rebuttal a stronger capacity characterization linking retrievability under random memories to separation on hyperbolic shells. A more detailed discussion is provided in our reply to Reviewer YTs9, which is visible to all reviewers.
>
> **On the organization of the main text and appendix.** We agree that a substantial amount of important material is currently placed in the appendix. Due to space constraints, we prioritized in the main text the problem setup, the core derivations, and the experiments most directly testing the hierarchy-sensitivity hypothesis, while moving finer proofs, module details, runtime analysis, and supplementary experiments to the appendix. We will further improve the organization so that the main text is more self-contained and easier to follow.

---

> > ### Author Rebuttal · Reviewer_Cqb5 · 2026-04-04
> >
> > Thanks for your response.
> >
> > W3 (Appendix organization): Partially resolved. The authors acknowledge the issue and commit to reorganizing the main text to be more self-contained. I take this at face value but note that the revision should ensure that the update rule (Section 3.3), the key module architectures (currently only in appendix), and the runtime comparison are accessible in the main text or a clearly signposted summary thereof.
> >
> > W4/Q2 (Inconsistent experimental reporting): Partially resolved. The clarification that the real hierarchical tasks in App. E.4/E.5 were intended as plug-in feasibility tests rather than full benchmark-style comparisons is reasonable, and the shared encoder/training protocol across baselines ensures decoder-level fairness. However, providing multi-seed results or at minimum a significance test for WN18RR (Table 11) and the WordNet hypernym prediction tasks (Tables 9, 10) would further strengthen confidence, especially since these are the tasks where the hierarchy-sensitivity hypothesis is most directly tested.
> >
> > Q1 (Source of performance gains): Partially resolved. The argument that existing baselines already provide partial factor isolation — MHN_Euc vs. HAMNs for the effect of hyperbolic geometry, HypAttn vs. HAMNs for the effect of iterative energy-driven retrieval — has merit. **However, a cleaner ablation that fully disentangles these factors** (e.g., Euclidean iterative retrieval or hyperbolic single-step retrieval) would be more convincing. The authors' position that the gains reflect a joint effect is scientifically cautious and fair, but the paper would benefit from making this explicit alongside the partial ablation that already exists.

---

> > > ### Author Response · Authors · 2026-04-04
> > >
> > > Thank you for the follow-up and for the additional feedback. We greatly appreciate that you found the current rebuttal helpful. For the points that remain only partially resolved, if the paper is accepted, we will explicitly incorporate these changes into the final version.
> > >
> > > We will add multi-seed results for WN18RR (Table 11) and WordNet hypernym prediction (Tables 9, 10), or at least provide significance analysis, to further strengthen the credibility of the conclusions. We will also clarify the role of these experiments and the corresponding tuning protocol.
> > >
> > > Regarding the organization of the main text and appendix, our planned revision is essentially a clearer reallocation of material. We will reorganize the main paper and appendix by moderately shortening the Preliminaries in the main text and moving more detailed hyperbolic-geometry background to the appendix; correspondingly, we will move the content in Appendix D.1 that is most directly related to the method — especially the Poincaré-ball update rule / pseudocode, key module structures, and a more coherent presentation of the computational pipeline — into the main text, so that the detailed formulas are less dispersed and the method can be understood without frequent appendix jumps. At the same time, we will add a brief but more explicit comparison with hyperbolic attention in the main text, while keeping the more detailed analysis in Appendix E.8.
> > >
> > > Thank you again for the careful suggestions.

---

### Official Review · Reviewer_MXqH · 2026-03-13

**Soundness:** 2
**Presentation:** 2
**Significance:** 3
**Originality:** 3
**Overall Recommendation:** 4
**Confidence:** 3

**Summary:**

Authors propose the new hopfield networks design. Actually a new attention mechanism, based on calculations within the hyperbolic space. The main motivation is to switch polynomially growing Euclidean space with exponentially growing hyperbolic one, namely Poincare ball. Authors provide sufficient calculations to motivate their design. They report classification accuracy on multi-layer hierarchical CIFAR-100, MIL and real-world taxonomy tasks. They show that their approach performs on par with Euclidean MHNs on shallow hierarchies while the performance gain raises with deeper hierarchies.

**Compliance With Llm Reviewing Policy:**

Affirmed.

**Final Justification:**

Following the rebuttal, concerns about the number of trainable vectors and the explicitness of the state update formula were partially resolved. However, the core concern remains: the paper lacks a synthetic benchmark directly demonstrating the theoretical advantage of hyperbolic geometry over Euclidean at sufficient hierarchy depth. The existing experiments on CIFAR-100 are insufficient to isolate this effect. 3 -> 4 Weak accept.

**Key Questions For Authors:**

See the weaknesses.

**Limitations:**

yes

**Strengths And Weaknesses:**

Strengths:
- **Raising interesting underdiscussed problems in Euclidean attention**: the necessity of representing hierarchical data in hidden states indeed fits better in the hyperbolic space.
- **Reasonable solution**: using a simple hyperbolic space (Poincare ball) for attention computation aligns well with the provided mathematical explanation.
- **Testing on the wide set of benchmarks**: showing performance gains on hierarchical data while preserving comparable quality on unrelated tasks.


Weaknesses:
- **No synthetic benchmark with clear performance gains**: When we are talking about the difference between polynomial and exponential growth, it is expected that some tasks are supposed to be completely unsolvable with Euclidean attention, while 100% solvable with a hyperbolic approach. I think it is essential to have such a proof of concept, and it would highlight the necessity to use this method.
- **No specific number of iterations/number of trainable vectors for the CIFAR task is provided**: only the validation search space is provided.
- **No explicit formula for state update**: while understanding the math and true motivation requires some diving into hyperbolic geometry, it would be good to be able to see the explicit transformation of the states in the Poincare ball model (maybe as a code snippet or explicit update rule).
- Much of the paper is devoted to preliminaries of hyperbolic space. However, the proposed neural modules have been moved to the appendix. This makes it difficult to understand exactly how the experiments were conducted and which models were used.

---

> ### Author Rebuttal · Authors · 2026-03-28
>
> Thank you for the careful and constructive review.
>
> First, we would like to clarify that HAMNs are not simply “another form of hyperbolic attention.” While traditional attention and a single iteration of MHNs are formally related and are often discussed together in practice, our method defines an explicit Hopfield-style energy function on a negatively curved manifold and performs retrieval through iterative updates. In contrast, hyperbolic attention is closer to a one-step geometrically weighted aggregation and does not involve an explicit global energy function or retrieval dynamics in the same sense. We recognize that this distinction has not been emphasized sufficiently in the current main paper, and this is also what we aim to clarify further in Appendix E.8.
>
> **Regarding the synthetic benchmark.** We agree that a cleaner proof-of-concept benchmark would more directly illustrate the effect of the geometric differences involved. However, the goal of this paper is not to construct a fully isolated synthetic separability benchmark, but rather to test our hierarchy-sensitivity hypothesis jointly in controlled hierarchical settings and real hierarchical tasks. To this end, the current evidence in the paper comes from two main sources: (1) experiments with controllable hierarchy depth, which directly examine how model behavior changes as hierarchy depth increases; and (2) real taxonomy tasks, which test whether the same hypothesis continues to hold in naturally occurring structured label spaces. Taken together, these two lines of evidence support the central conclusion of the paper: when the underlying hierarchy is sufficiently deep and well organized, hyperbolic retrieval is more suitable than Euclidean Hopfield-style retrieval. We agree that the reviewer’s suggested synthetic benchmark would be a valuable additional illustration. While such a benchmark would further strengthen the presentation, we believe that the current evidence is already sufficient to support the main empirical conclusions of the paper.
>
> **Regarding the number of iterations and the number of trainable vectors in the CIFAR experiments.** We agree that the current presentation is not sufficiently complete. For clarity, we note here that we use one retrieval step, which is consistent with the common use of Hopfield-style memory modules. The number of trainable vectors is set to 200 based on the ablation results in Appendix E.7. If the paper is accepted, we will also make these hyperparameters easier to locate in the camera-ready version.
>
> **On the organization of the main text and appendix.** We also accept the criticism regarding the presentation of the paper. We apologize that the state update formula is not sufficiently prominent in the current version. The explicit update rule is already given in Section 3.3, and the corresponding pseudocode on the Poincaré ball is provided in Appendix D.1. Due to space limitations, we chose in the current version to focus the main text on the problem formulation, the core theoretical derivations, and the experimental results that most directly support the main message of the paper, while placing more detailed proofs, module implementations, and supplementary experiments in the appendix. As a result, the readability of some parts of the main paper is affected. We also agree that the organization could be improved so that the update rule, the structures of the key modules, and the mapping between the experiments and the model are clearer in the main paper. If accepted, we will improve the presentation in the camera-ready version.

---

> > ### Author Rebuttal · Reviewer_MXqH · 2026-04-04
> >
> > Thank you for the rebuttal. My concerns are partially resolved.
> >
> > The clarification regarding the number of trainable vectors (200) is appreciated, though I note this information appears only in the rebuttal and has not been incorporated into the paper or Appendix E.7.
> >
> > The following two concerns remain unresolved:
> >
> > 1. No synthetic benchmark. I continue to believe this is the most critical missing piece. The core theoretical motivation of the paper - exponential vs. polynomial capacity growth - strongly implies the existence of tasks that are fundamentally intractable for Euclidean attention but trivially solvable in hyperbolic space. Without a synthetic proof-of-concept demonstrating this regime, the practical necessity of the proposed method remains unsubstantiated. The rebuttal does not provide such an experiment.
> >
> > 2. Insufficient explicitness of the state update formula. The provided pseudocode and what the authors describe as an explicit formula still rely on operations such as parallel transport (PT), exponential map (exp), and logarithmic map (log) on the Poincare ball without spelling out how these are computed in practice. For readers without a strong background in differential geometry, these operations remain computationally opaque. A self-contained derivation or a concrete numerical example would be necessary to make the update rule genuinely accessible.

---

> > > ### Author Response · Authors · 2026-04-04
> > >
> > > Thank you for the follow-up review and for raising these important questions. We apologize that our previous wording was still not sufficiently precise, and we would like to further clarify two points.
> > >
> > > First, the theoretical motivation of this paper is not that **Euclidean MHNs have only polynomial capacity, whereas HAMNs have exponential capacity,** nor that **therefore there must exist tasks that are fundamentally intractable for Euclidean attention but trivially solvable by hyperbolic methods.** In Appendix B.4, we already explain that classical MHNs can themselves exhibit exponential capacity for random patterns as the ambient dimension grows; at the same time, in our reply to Reviewer YTs9, we further provide a random-memory capacity theorem for HAMNs, so both admit exponential-type capacity results. The **exponential vs. polynomial** discussion in the paper is therefore not about capacity itself, but about a different quantity: under **fixed state dimension and minimum geodesic separation**, the number of well-separated recall wells grows exponentially with the **effective hyperbolic radius / hierarchy depth**, whereas the corresponding Euclidean packing bound grows only polynomially with the **radius**. In other words, this part of the theory supports the claim that hyperbolic retrieval is more suitable for sufficiently deep hierarchical structure, rather than that there exist tasks for which Euclidean methods are unusable in principle. Accordingly, our theoretical claim is that when structure expands mainly along the **depth/radial** direction, hyperbolic geometry provides a more favorable retrieval layout; we do not claim that Euclidean MHNs are unusable in principle. We therefore retain our earlier position that the empirical claim of the paper is the **hierarchy-sensitivity hypothesis**: HAMNs are more advantageous for sufficiently deep hierarchies, while remaining broadly comparable to Euclidean MHNs on shallow or weakly hierarchical tasks. The core empirical goal of the paper is to test this hypothesis, rather than to construct a fully isolated separability toy problem.
> > >
> > > Second, regarding the state update formula, we agree that the current presentation is still not sufficiently self-contained for readers without a differential-geometry background, but this is different from saying that no explicit formula is provided. The exp/log definitions and explicit update structure given in Chapter 3 of the main text are **model-agnostic**; precisely because no specific hyperbolic model is fixed there, the corresponding concrete exp/log formulas are collected in Appendix C. In Appendix D.1, we further provide the pseudocode and full computational pipeline for the Poincaré ball instantiation; the Poincaré-ball exp/log formulas are given in Appendix C, and the parallel transport (PT) used in the Poincaré ball is also specified in Appendix D.1. Therefore, we believe the issue is more accurately that these formulas and implementation details are currently distributed across the main text and appendix, which reduces readability, rather than that the paper lacks an explicit update rule itself. In addition, we have also submitted the detailed code for our method under the Poincaré-ball model, which should further help readers understand how the method is implemented in practice.
> > >
> > > Finally, if the paper is accepted, we will add the hyperparameter settings and further improve the organization of the paper. A more detailed explanation of this point is also provided in our Rebuttal Acknowledgement response to Reviewer Cqb5.

---

### Decision · Program_Chairs · 2026-04-30

**Decision:**

Accept (regular)

**Comment:**

Reviewers enjoyed reading this paper and acknowledge its novelty relative to existing literature. The paper uses excellent packaging properties of negatively curved spaces to design a large capacity associative memory based on different computational principles than the commonly used dense associative memory or modern Hopfield network, see [comments from Reviewer YTs9](https://openreview.net/forum?id=m7SH64ZF6j&noteId=4KtvBwMx99) . The latter class of models use higher order activation functions (powers, exponents, softmax) to achieve large capacity. In contrast, the proposed model utilizes the geometry of the underlying space. Reviewers note that the combination of geometry and machine learning makes the paper particularly interesting and valuable.

While the technical content seems strong, the presentation is sloppy. Specifically, many of the references used in the paper do not make any sense. For instance, in lines 14-15 of the introduction the authors cite Vaswani 2017 and Widrich 2020 as the original sources for MHNs. Vaswani’s paper is about attention, not MHNs. Widrich 2020 is an application paper dedicated to study of the immune system with MHNs. Theoretically, MHNs were introduced by [Krotov and Hopfield 2016](https://proceedings.neurips.cc/paper_files/paper/2016/hash/eaae339c4d89fc102edd9dbdb6a28915-Abstract.html) for both binary and continuous variables. Ramsauer 2021 proposed a specific type of MHN with continues variables and softmax activation that describes attention. Demircigil 2017 exclusively studied binary variables, in contrast to what is written in lines 80-81 of section 2.1. Conventional continuous Hopfield networks were developed in [Hopfield 1984](https://www.pnas.org/doi/abs/10.1073/pnas.81.10.3088) and not in Tank & Hopfield 1986, contrary to what is stated in lines 413-414. The systematic analysis of various models within the MHN family was done in [Krotov and Hopfield 2021](https://arxiv.org/abs/2008.06996). The authors need to correct all this sloppiness in the camera ready version. I recommend acceptance, conditioned on this outstanding task.